



Plateaus and jumps in the atmospheric radiocarbon record – Potential origin and value as
global age markers for glacial-to-deglacial paleoceanography, a synthesis
Michael Sarnthein[1], Kevin Küssner[2], Pieter M. Grootes[3], Blanca Ausin[4], Timothy
Eglinton[4], Juan Muglia[5], Raimund Muscheler[6], Gordon Schlolaut[7]
1) Institute of Geosciences, University of Kiel, Olshausenstr. 40, 24098 Kiel, Germany,
michael.sarnthein@ifg.uni-kiel.de,  (corresponding author)
2) Alfred-Wegener-Institut Helmholtz-Zentrum für Polar- und Meeresforschung,
Department for Marine Geology, 27570 Bremerhaven, Germany, kevin.kuessner@awi.de
3) Institute of Ecosystem Research, University of Kiel, Olshausenstr. 40, 24098 Kiel,
Germany, pgrootes@ecology.uni-kiel.de
4) Geological Institute, ETH Zürich, Sonneggstr. 5, 8092 Zuerich, Switzerland,
blanca.ausin@erdw.ethz.ch
5) College of Earth, Ocean and Atmospheric Sciences, Oregon State University, 104
CEOAS Administration Building, 101 SW 26th St, Corvallis, OR 97331, USA,
juanmuglia@gmail.com
6) Quaternary Sciences, Department of Geology Lund University, Sölvegatan 12, S-
223 62 Lund, Sweden, raimund.muscheler@geol.lu.se
7) Climate Dynamics and Landscape Evolution, GFZ German Centre for Geosciences,
Telegrafenberg, 14473 Potsdam, Germany, SchlolautG@gmail.com





ABSTRACT
Changes in the geometry of ocean Meridional Overturning Circulation (MOC) are crucial in
controlling changes of climate and the carbon inventory of the atmosphere. However, the precise
timing and global correlation of short-term glacial-to-deglacial changes of MOC in different ocean
basins still present a major challenge. A possible solution is offered by the fine structure of jumps
and plateaus in the record of radiocarbon ($^{14}$C) concentration of the atmosphere and surface ocean
that reflects changes in atmospheric $^{14}$C production as well as in the $^{14}$C exchange between air
and sea and within the ocean. Boundaries of atmospheric $^{14}$C plateaus in the $^{14}$C record of Lake
Suigetsu, now tied to Hulu U/Th model-ages instead of optical varve counts, provide a
stratigraphic 'rung ladder' of ~30 age tie points from 29 to 10 ka for correlation with and dating of
planktic oceanic $^{14}$C records. The age difference between contemporary planktic and atmospheric
$^{14}$C plateaus gives an estimate of the global distribution of $^{14}$C reservoir ages for surface waters of
the Last Glacial Maximum (LGM) and deglacial Heinrich Stadial 1 (HS-1), as shown by 19 planktic
$^{14}$C records. Clearly elevated and variable reservoir ages mark both high-latitude sites covered by
sea ice and/or meltwater and upwelling regions. $^{14}$C ventilation ages of LGM deep waters reveal
opposed geometries of Atlantic and Pacific MOC. Similar to today, Atlantic deep-water formation
went along with an estuarine inflow of old abyssal waters from the Southern Ocean up to the
northern North Pacific and an outflow of upper deep waters. Vice versa, $^{14}$C ventilation ages
suggest a reversed MOC during early HS-1 and a ~1500 year long flushing of the deep North
Pacific up to the South China Sea, when estuarine circulation geometry marked the North Atlantic,
gradually starting near 19 ka. Elevated $^{14}$C ventilation ages of LGM deep waters reflect a major
drawdown of carbon from the atmosphere. Inversely, the subsequent massive age drop and
change in MOC induced two major events of carbon release to the atmosphere as recorded in
Antarctic ice cores, shifts that highlight the significance of ocean MOC for atmospheric $CO_2$ and its
$^{14}$C inventory. These new features of MOC and the carbon cycle offer a challenge to model
simulations that, in part because of insufficient spatial model resolution and reference data for
testing the model results, still poorly reproduce them.





LIST of CONTENTS
1. INTRODUCTION – A THEORETICAL FRAMEWORK
*1.1– A variety of terms linked to the notion ʼ$^{14}$C ageʼ*
*1.2– Review of tie points used to fix calendar and reservoir ages in marine $^{14}$C records*
*1.3– Items to be addressed in this synthesis*
2. AGE TIE POINTS BASED ON $^{14}$C PLATEAU BOUNDARIES
*2.1– A slight revision of absolute age control of the Suigetsu $^{14}$C record*
*2.2– Uncertainties of age control   (A chapter to be shifted to Suppl. Materials #1)*
3. DISCUSSION
*3.1– Origin of short-term structures in the atmospheric $^{14}$C record: Changes in cosmogenic $^{14}$C production*
*        versus changes in ocean dynamics*
*3.2– $^{14}$C plateau boundaries – A suite of narrow spaced age tie points (a 'rung ladder') to rate short-term*
*        changes in marine sediment budgets, chemical inventories, and climate 29–10 cal. ka*
*3.3– Definition and origin of Zoophycos burrows: A key foe of high-resolution stratigraphy in Pleistocene*
*        sediment records turned into an ally?*
*3.4– Empiric vs. model-based $^{14}$C reservoir ages acting as tracer of past changes in surface ocean*
*        dynamics*
*3.5– Plankton-based $^{14}$C reservoir ages – A prime database to estimate past changes in the $^{14}$C*
*        ventilation age of deep waters, ocean MOC, and DIC for past states of the ocean*
*        3.5.1 • Major features of meridional overturning circulation during LGM*
*        3.5.2 • Major features of meridional overturning circulation during early HS-1*
*        3.5.3 • Deep-Ocean DIC inventory*
4. SOME CONCLUSIONS
ACKNOWLEDGMENTS
REFERENCES
TABLE and FIGURE CAPTIONS
Supplementary Figure Captions and Figures





# 1. INTRODUCTION

## 1.1 A variety of terms linked to the notion '$^{14}$C age'

The $^{14}$C concentration in the troposphere is mainly determined by $^{14}$C production, atmospheric mixing, moreover, air-sea gas exchange, and ocean circulation that vary over time (e.g., Alves et al., 2018; Alveson et al., 2018). The $^{14}$C content of living terrestrial plants is in equilibrium with the atmosphere via processes of photosynthesis, respiration, and accordingly, the $^{14}$C of terrestrial plant remains in a sediment section directly reflects the amount of radioactive decay, thus the time passed since the plant's death, and the $^{14}$C composition of the atmosphere during the time of plant growth.

Contrariwise, $^{14}$C values of marine waters are cut off from cosmogenic $^{14}$C production in the atmosphere, hence depend on the carbon transfer at the air-sea interface and transport and mixing of carbon in the ocean. For surface waters, the air-sea transfer is relatively fast and effective involving a time span of ten years and less (e.g., Nydal et al., 1998). Yet, vertical and horizontal water mixing results in surface ocean $^{14}$C concentrations differing from those in the contemporaneous atmosphere, expressed as differential $^{14}$C 'reservoir ages' (or 'reservoir effects' *sensu* Alves et al., 2018). These 'ages' reflect the local oceanography and are highly variable through time. Differences may range from near zero up to values of more than 700 yr, in some regions up to 2500 yr, induced, for example, by old waters upwelled from below (e.g., Stuiver and Braziunas, 1993; Grootes and Sarnthein, 2006; Sarnthein et al., 2015). Apart from U/Th dated corals (many papers on their reservoir age since Adkins and Boyle, 1997) the $^{14}$C age of planktic foraminifers is the most common tracer of surface water ages in marine sediments, a rough estimate of the time passed since sediment deposition. Initially, marine geologists were most interested in this 'simple' age value. Soon, however, they were confronted with





age inconsistencies that implied a series of unknowns, in particular the [14]C 'reservoir age'
that finally turned out to be a most valuable tracer of oceanography.

In turn, [14]C records of benthic carbonate particles in deep-sea sediments sum the time of
radioactive decay since their deposition with the apparent 'ventilation age' of the deep
waters in which they lived. Ventilation age is primarily the time span from the moment
when carbon dissolved in the (later) deep waters lost contact with the [14]C level of the
atmosphere and the somewhat reduced level of surface waters until the precipitation of
benthic carbonate. Details on the derivation of ventilation ages are given in Cook and
Keigwin (2015) and Balmer and Sarnthein (2018). In addition, however, ventilation ages
depict hardly quantifiable lateral admixtures of older and/or younger water masses,
moreover, [14]C-enriched organic carbon supplied by the biological pump, thus are called
'apparent'. Today, the apparent transit times of carbon dissolved in the deep ocean range
from a few hundred up to ~1800 [14]C yr found in upper deep waters of the northeastern
North Pacific (Matsumoto, 2007).

Over the last decades, it turned out that both the reservoir ages of surface waters and the
ventilation ages of deep waters present robust and high-resolution tracers essential for
drawing quantitative conclusions on past ocean circulation geometries, marine climate
change, and the processes that drive both past ocean dynamics and carbon budgets,
given the ages rely on a number of robust age tie points. Obtaining such tie points
presents a problem, since any attempt to date a deep-sea sediment record by means of
[14]C encounters a number of intricacies of how to disentangle (i) the effects of atmospheric
[14]C variations due to past changes in cosmogenic [14]C production and carbon cycle, still
hampered by the need for a generally accepted atmospheric reference record for the



period 14–50 ka, from (ii) depositional effects such as sediment hiatuses and winnowing,
differential bioturbational mixing depth, sediment transport by deep burrows, (iii) the
effects of ocean mixing resulting in reservoir and ventilation ages that change through time
and space (e.g., Alves et al. 2018; Grootes and Sarnthein, 2006), and (iv) from the final
target, quantitatively 'pure' $^{14}$C ages due to radioactive decay.

By now, $^{14}$C-based chronologies of deep-sea sediment records, used to constrain and
correlate the age of glacial-to-deglacial changes in ocean dynamics and climate on a
global scale, are often of unsatisfactory quality when they are based on (i) age tie points
spaced far too wide-spaced (e.g., on DO-events 1, 2, and 3 only for the time span 30–14
cal ka), (ii) disregarding atmospheric $^{14}$C plateaus, (iii) the risky assumption of ±constant
planktic $^{14}$C reservoir ages and other speculative stratigraphic correlations/compilations,
and (iv) on ignoring small-scale major differences in low-latitude reservoir age. Likewise,
clear conclusions are precluded by an uncertainty range of 3-4 kyr sometimes accepted
for tie points during the glacial-to-deglacial period (Lisiecki and Stern, 2016), where
significant global climate oscillations occurred on decadal-to-centennial time scales as
widely shown on the basis of speleothem and ice core-based records (Steffensen et al.,
2008; Svensson et al., 2008; Wang et al., 2001).

Thus marine paleoclimate and paleoceanographic studies today focus on the continuing
quest for a high-resolution and global, hence necessarily atmospheric $^{14}$C reference record
that is marked by abundant, narrow-standing tie points on the calibrated (cal.) age scale.
Such pertinent tie points are provided by a suite of reproducible 'plateaus' and 'jumps' that
mark the atmospheric $^{14}$C record (Figs. 1 and S1; Sarnthein et al., 2007 and 2015; Bronk



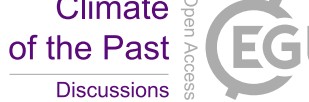

Ramsey et al., 2012 and 2019; Schlolaut et al., 2018; Umling and Thunnell, 2017), hence
form the basis of this synthesis.

*1.2  Review of tie points used to fix calibrated and reservoir ages in marine $^{14}$C records*

The tree ring-based calibration of $^{14}$C ages provides a master record of decadal changes
in atmospheric $^{14}$C concentrations back to ~14 cal. ka (Reimer et al., 2013 and 2019) with
floating sections beyond (from ~12.5–14.5 cal. ka and around 29–31.5 and 43 cal. ka;
Turney et al., 2010, 2017). The evolution of Holocene and late deglacial $^{14}$C ages with time
is not linear but reveals variations with numerous distinct jump (= rapid change) and
(short) plateau-shaped (slow or no change or even inversion) structures indicative of
fluctuations in atmospheric $^{14}$C concentration. Prior to 8500 cal. yr BP, various plateaus
extend over 400–600 cal. yr and beyond (Fig. 2). Given the quality of the tree ring
calibration data, these fluctuations can be considered real, suitable for global correlation
(Sarnthein et al., 2007, 2015; Sarnthein and Werner, 2018). Air-sea gas exchange
transfers the atmospheric $^{14}$C fluctuations into the surface ocean where they can provide
high-resolution tie points to calibrate the marine $^{14}$C record and marine reservoir ages
back to ~14 ka (via the so-called $^{14}$C wiggle match approach). In the near future, however,
it is unlikely that a continuous tree ring-based record will become available to trace such
atmospheric $^{14}$C variations further back, over the period 14–29 cal. ka crucial for the
understanding of last-glacial-to-interglacial changes in climate. Hence various other, less
perfect $^{14}$C archives have been employed for this period to tie past changes in
atmospheric $^{14}$C concentration/age to an 'absolute' or 'calibrated' (e.g., incremental) age
scale and to constrain the widely unknown evolution of $^{14}$C reservoir ages of surface
waters for various regions of the ocean.




Suites of [14]C ages of paired marine and terrestrial plant-borne samples, e.g. paired
planktic foraminifers and wood chunks, provide most effective but rarely realizable
absolute-age markers and reservoir ages of local ocean surface waters (Zhao and
Keigwin, 2018; Rafter et al., 2018; Schroeder et al., 2016; Broecker et al., 2004).
Likewise successful can be the alignment of [14]C-dated variations in downcore sea-
surface temperatures (SST) with changes in hydroclimate as recorded in age-
calibrated sedimentary leaf-wax hydrogen isotope (δD) records from ancient lakes
(Muschitiello et al., 2019), assumed to be synchronous. Further tie points are derived
from volcanic ash layers (Waelbroeck et al., 2001; Siani et al, 2013; Davies et al.,
2014), paired U/Th- and [14]C-based coral ages (e.g., Adkins and Boyle, 1997; Chen et
al., 2015), and the (fairly fragmentary) alignment of major tipping points in [14]C dated
records of marine SST and planktic $\delta^{18}$O to the incremental age scale of climate events
dated in Greenland and Antarctic ice core records (Waelbroeck et al., 2011). Such well-
defined climate-age tie points, however, are wide-spaced in peak glacial-to-early
deglacial ice core records. Finally, various data compilations tentatively rely on the use
of multiple age correlations amongst likewise poorly dated marine sediment records, an
effort necessarily problematic.

In the absence of robust age tie points an increasing number of authors resort to [14]C
reservoir age simulations for various sea regions by ocean GCMs (e.g. Butzin et al., 2017;
Muglia et al., 2018) to quantify the potential difference between marine and atmospheric
[14]C dates during glacial-to-interglacial times. Considering the complexity of the ocean
MOC and the global carbon cycle it is not surprising that the results of a comparison of a



selection of robust empiric vs. simulated $^{14}$C reservoir ages are not that encouraging yet
(as discussed further below).

Accepting a generally close link between $^{14}$C concentrations in the troposphere and in the
surface ocean, the fine structure of planktic $^{14}$C records with centennial-scale-resolution
provides far superior (though costly) evidence, similar to that of tree rings, to furnish a
series of age tie points with semi-millennial-scale time resolution for a global correlation of
glacial-to-deglacial marine sediment sections. These suites of tie structures can link the
marine sediment records to a reference suite of narrow-standing jumps and boundaries of
the apparent plateaus found in the atmospheric $^{14}$C record of Lake Suigetsu (Bronk
Ramsey et al., 2012, 2019) provided that common $^{14}$C variations are robustly identified in
both atmospheric and marine records. Prior to 14 cal. ka, the absolute age of these
atmospheric $^{14}$C structures can be calibrated either by (microscopy-based) varve ages or
by a series of paired U/Th- and $^{14}$C-based model ages correlated from the Hulu Cave
speleothem record (Bronk Ramsey, 2012 and 2019; Southon et al., 2012; Cheng et al.,
2018). The difference between these calibrations (Fig. 3) is discussed below. It is,
however, little important for both the correlation and the derivation of the time-varying
offsets in $^{14}$C concentration of planktic sediments from coeval concentrations in the
atmospheric record, an offset derived from the average $^{14}$C age difference between two
'coeval' planktic and Suigetsu $^{14}$C plateaus correlated.

A basic philosophical controversy exists whether the apparent jump and plateau structures
in the Suigetsu and planktic $^{14}$C records reflect real $^{14}$C fluctuations or statistical noise. In
the 'null hypothesis' the $^{14}$C values shaping plateaus of the calibration curve are regarded
as result of mere statistical scatter. Thus, the record of atmospheric $^{14}$C ages against time



would form a simple continuous rise resulting from radioactive decay and the advance of
time, such as suggested by a fairly straight progression of the highly resolved deglacial
Hulu Cave [14]C record plotted vs. U/Th ages (Southon et al., 2012; Cheng et al., 2018).

This null hypothesis is contradicted by the 'master record' of tree ring data (Fig. 2; Reimer
et al., 2013 /2020). Unequivocally it shows fluctuations in atmospheric [14]C concentration
on the order of 2–3 % over the last 10 kyr (Stuiver and Braziunas, 1993) and even larger
back to ~14 ka (Reimer et al., 2013, 2020). Though not resolved in speleothem data these
plateau/jump structures are real and widely reproducible in marine sediment records.
Under glacial and deglacial low-$CO_2$ conditions beyond 14 ka, when climate and ocean
dynamics were less constant than during the Holocene, atmospheric [14]C fluctuations were,
most likely, even stronger than those reported by Stuiver and Braziunas and [14]C plateaus
and jumps accordingly larger.

Thus, the age-defined plateaus and jumps in the Suigetsu atmospheric [14]C calibration
curve may most likely be regarded as a suite of 'real' structures, extending the tree ring
record for Holocene and B/A-to-Early Holocene times (Fig. 2) into early deglacial and LGM
times. In part the plateau/jump structures may be linked to changes in cosmogenic [14]C
production, as possibly shown in the [10]Be record (Fig. 4; based on data of Adolphi et al.,
2018), and – presumably more dominant – to short-term changes in ocean mixing and the
carbon exchange between ocean and atmosphere, the latter crucial, since the carbon
reservoir of the ocean contains up to 60 (preindustrial) atmospheric carbon units (Berger
and Keir, 1984). The apparent contradiction with the smooth Hulu Cave [14]C record
(Southon et al., 2012; Cheng et al., 2018) may possibly be explained by (i) the Hulu Cave
speleothem precipitation system acting as a low-pass filter for fluctuating atmospheric [14]C



concentrations (statistical tests of Bronk Ramsey et al., pers. comm. 2018), (ii) to a very
limited degree by the obvious scatter in the Suigetsu data that, however, appears
insufficient to feign plateaus in view of the evidence based on tree ring based plateaus
(Fig. 2). The filter for Hulu data possibly led to a sweeping loss especially of short-lived
structures in the preserved atmospheric $^{14}$C record, though some remainders indeed were
preserved in the $^{14}$C records of Hulu Cave (Fig. 1). So we rather trust in the amplitude of
Suigetsu 14C structures, but trust in the timing of Hulu Cave data as discussed below.

Like a 'rung ladder' the age-calibrated suite of $^{14}$C plateau boundaries and jumps is suited
for tracing the calibrated age of numerous plateau boundaries in glacial-to-deglacial
marine $^{14}$C records likewise densely sampled. Moreover, one may record the offset of
planktic $^{14}$C ages from paired atmospheric $^{14}$C ages that is the planktic reservoir age, for
each single $^{14}$C plateau (Sarnthein et al., 2007, 2015). For the first time, this suite of tie
points may facilitate a precise temporal correlation of all sorts of changes in surface and
deep-water composition on a global scale, crucial for a better understanding of past
changes in ocean and climate dynamics.

Over the time span 14–40 ka, the Suigetsu record of optical varve counts (Schlolaut et al.,
2018) presents a rare means of age calibration of terrestrial and marine sediment records
being based on an incremental age scale similar to that of ice cores (e.g. Svensson et al.,
2008). In the crucial sediment sections of the Last Glacial Maximum (LGM) and deglacial
Heinrich Stadial 1 (HS-1), however, the degree of varve quality/perceptibility in the
Suigetsu profile is highly variable (Fig. 5), a problem met by Schlolaut (2018) who
developed a special computer program to derive Suigetsu calendar ages from varve
counts. Nonetheless the interpolated varve counts have limited accuracy and precision. To





further improve the chronology the varve data were combined with Hulu Cave-based U/Th-
based model ages employed in this paper (Bronk Ramsey et al., 2012). The different age
models, however, do not affect our conclusions on planktic reservoir ages.

*1.3  Items to be addressed in this synthesis*

(1) The purely varve-based chronology of $^{14}$C plateau boundaries previously employed for
the Suigetsu record (Sarnthein et al., 2015) may be incomplete in view of broad poorly
laminated sediment sections interpolated in Suigetsu sediment cores, recently recorded by
Schlolaut et al. (2018). As compared to a U/Th-based age model (Bronk Ramsey et al.
2012; Cheng et al., 2018) many hundred years appear to have been missed. Thus, the
value of different age calibrations needs to be re-evaluated.

(2) In view of most recent findings on the quality of varve counts (Fig. 5; Schlolaut et al.,
2018) our suite of tie points is now extended from 23 to 27/29 cal. ka. The calibrated-age
uncertainties of $^{14}$C plateau boundaries and jumps in the redefined Suigetsu record
(Bronk-Ramsey et al., 2012, 2019; Sarnthein et al., 2015) and their correlatives in ocean
sediment cores, values crucial for any accurate correlation of these tie points to possibly
underlying ocean events over the period 10–27/29 cal. ka, are now discussed in
Supplement Text no. 1.

(3) Our set of records of marine $^{14}$C reservoir ages (Sarnthein et al. 2015) has now been
amended by several records from the Southern Hemisphere (Balmer et al., 2016 and
2018; Küssner et al., 2017, and in prep.) and northeast Atlantic (Ausin et al., 2019 and in
prep.). In total, 18 (LGM) / 19 (HS-1) empiric records plus 3 wood chunk-based records





(e.g. Broecker et al., 2004; Zhao et al., 2018) now depict the spatial and temporal variation
of past [14]C reservoir ages of surface waters for different ocean regions.

(4) In the discussion we will compare our local reservoir ages with independent LGM
estimates of surface water [14]C reservoir ages simulated by the GCM of Muglia et al.
(2018). Differences between the results may help to constrain potential caveats in the
boundary conditions and fine structure of model simulations.

(5) We discuss some habitat- and season-specific [14]C reservoir ages characteristic of
different planktic foraminifera species, ages that monitor for past changes in the local
geometry of surface ocean dynamics (Sarnthein and Werner, 2018).

(6) Finally, we refer to [14]C reservoir and ventilation ages of surface and deep waters that
form a robust tracer of circulation geometries and the dissolved inorganic carbon (DIC) in
different basins of the ocean (Sarnthein et al., 2013). The estimates provide crucial
insights into the origin of past changes in the global carbon cycle from glacial to
interglacial times, an important correlative to model simulations.

2.  AGE TIE POINTS BASED ON [14]C PLATEAU BOUNDARIES
*2.1 A slight revision of absolute age control of the Suigetsu [14]C record*

Originally, we based the chronology of [14]C plateau boundaries in the Suigetsu record
(Sarnthein et al., 2015) on a scheme of varve counts by means of light microscopy of thin
sections (Bronk Ramsey et al., 2012; Schlolaut et al., 2018). In parallel, varve-based age
estimates have been derived from counting various elemental peaks in μXRF data,





interpreted as seasonal signals (Marshall et al., 2012). In general, the results obtained
from these two independent counting methods and their interpolations widely support each
other. The microscopy-based counts ultimately formed the backbone of a high-resolution
chronology obtained by tying the Suigetsu [14]C record to the U/Th based time scale of the
Hulu cave [14]C record (Bronk Ramsey et al., 2012). Recently the scheme of varve counts
was revisited by Schlolaut et al. (2018) who showed that Suigetsu varve preservation is
fairly high both over late glacial Termination I and prior to ~32 ky BP. However, it is fairly
poor over large parts of the LGM and HS-1, from ~15 - ~32 cal ka (17.3-28.5 m c.d. in Fig.
5), where less than 20-40 % of the annual layers expected from interpolation between
clearly varved sections are distinguished by microscopy per 20 cm sediment section. The
varve count using µXRF data (Marshall et al., 2012) can distinguish subtle changes in
seasonal element variations, which are not distinguishable in thin section microscopy,
hence results in higher varve numbers, for example, during early deglacial-to deglacial
times (Fig. 3). Yet, some subtle variations are difficult to distinguish from noise, thus also
introduce a degree of uncertainty to the µXRF-based counts. Thus the results deduced
from either counting method are subject to uncertainties, as shown by error estimates that
rise with increased varve age in Fig. 5.

In addition, Bronk Ramsey et al. (2012) established a time scale based on [14]C wiggle
matching to U/Th dated [14]C records of the Hulu Cave and Bahama speleothems. In part,
this calibrated (cal.) age scale was based on Suigetsu varve counts, in part on the
prerequisite of the best-possible fit of a pattern of low-frequency [14]C concentration
changes obtained from Suigetsu und Hulu Cave within the uncertainty envelope of the
Hulu 'Old / Dead Carbon Fraction' (OCF/DCF) of [14]C concentration. The uncertainty of this
model is debatable while the character of the Hulu DCF and thus, its uncertainty back in



time is still incompletely understood. We surmise that the U/Th-based age model of
Suigetsu may suffer from the wiggle matching of atmospheric [14]C ages of Lake Suigetsu
with [14]C ages of the Hulu Cave (Southon et al., 2012) in case of major short-term changes
in the memory effect of soil organic carbon in carbonate-free regions of the cave. These
carbonate-based ages may have been influenced far more strongly by short-term changes
in the local DCF than assumed, as suggested by major variations in a paired $\delta^{13}$C record,
that reach up to 5 ‰, mostly subsequent to short-term changes in past monsoon climate
(Kong et al., 2005). Thus Hulu [14]C ages cannot directly be set equal to atmospheric [14]C
ages under the assumption of a constant OCF/DCF (Southon et al. 2012; Cheng et al.,
2018), a caveat that hampers the age model correlation between Hulu and Suigetsu
records. It turns out, U/Th-based model ages of [14]C plateau boundaries are much higher
than the microscopy-based varve ages over HS-1 and LGM we used thus far, a difference
accumulating from ~200 yr near 15.3 cal. ka to ~600 near 17 ka and 2000 yr near ~29 ka
(Fig. 3) and finally accepted by means of independent evidence shown below.

To calibrate the age of thirty [14]C plateau boundaries with the 'best possible' cal. time
scale we compared the results of the two timescales independently deduced from
varve counts with those of the U/Th-based model age scale using as test case the
base of [14]C Plateau 2b. In contrast to 16.4 cal. ka suggested by optical varve counts,
the XRF-based varve counts suggest an age of ~16.9 cal. ka (Marshall et al., 2012;
Schlolaut et al., 2018). Most important, this estimate matches closely that of 16.93 ka
on the U/Th-based time scale, a robust argument supporting the latter age scale (Fig.
1). Moreover, different from the microscopy-based varve age scale the U/Th model
time scale is further corroborated by a decent match with the ages of Mono Lake at 34



ka and Laschamp at 41 ka independently dated by other methods. We therefore chose
the U/Th model age scale to calibrate the age of $^{14}$C boundaries.

The U/Th-based cal. ages result in reasonable stratigraphic correlations of millennial-scale
events in paleoceanography. Fig. 6 displays potentially correlative events of peak glacial
and deglacial change independently dated by means of annual-layer counts and/or U/Th
ages in ice cores and stalagmites (Table 2). As outlined below, atmospheric $^{14}$C plateaus
may largely result from changes in air-sea gas exchange, and in turn, from changes in
ocean ventilation. A suite of deglacial $^{14}$C plateaus indeed displays a temporal match with
major deglacial events in ocean degassing of $CO_2$ (Marcott et al., 2014) (Table 2 and Fig.
6). Also, short-term North Atlantic warmings matched three $^{14}$C plateaus each during the
peak glacial and earliest deglacial times, similar to that at the end of HS-1.

In view of the recent revision of time scales (Schlolaut et al., 2018; Bronk Ramsey et al,
2019) we extended our plateau tuning and now also defined the boundaries and age
ranges of $^{14}$C plateaus and jumps for the interval ~23–27/29 cal. ka, which results in a total
of ~30 atmospheric age tie points for the time span 10.5–29 cal. ka (Fig. 1; summary in
Table 1; following the rules of Sarnthein et al., 2007 and 2015). Prior to 25 cal. ka, the
definition of $^{14}$C plateaus somewhat suffered from an enhanced scatter of raw $^{14}$C values
of Suigetsu. -- In addition to visual inspection, the $^{14}$C jumps and plateaus were also
defined with higher statistical objectivity by means of the first-derivative of all trends in the
$^{14}$C age-to-calendar age relationship (or –core depth relationship, respectively) by using a
running kernel window (Sarnthein et al., 2015).

Note, any readjustment of the calendar age of a $^{14}$C plateau boundary does not entail any



change in $^{14}$C reservoir ages afore deduced for surface waters by means of the plateau
technique (Sarnthein et al., 2007, 2015), since each reservoir age presents the simple
difference in average $^{14}$C age for one and the same $^{14}$C plateau likewise defined in both
the Suigetsu atmospheric and planktic $^{14}$C records of marine surface waters, independent
of the precise position of this plateau on the calendar age scale.

*2.2  Uncertainties of age control     (Chapter to be presented as Suppl. Text #1)*

Rough estimates of uncertainty and aspects of analytical quality were published by
Sarnthein et al. (2007, 2015). We now focus on uncertainties tied to the calendar age
definition for each $^{14}$C plateau boundary both in the Suigetsu atmospheric and the
various marine sediment records (Table 1). To recap, an age/sediment section is
formally defined as containing a '$^{14}$C plateau', when $^{14}$C ages show almost constant
values with an overall gradient of <0.3 to <0.5 $^{14}$C yr per cal. yr (based on visual
description and/or statistical estimates by means of the 1st derivative of all
downcore changes in the $^{14}$C age – calendar age relationship; Sarnthein et al., 2015)
and a variance of less than ±100 to ±300 $^{14}$C yr, and up to 500 $^{14}$C yr prior to 25 cal. ka.
Here $^{14}$C ages form a plateau-shaped scatter band with up to 10% outliers, that
extends over more than 300 cal. yr in the Suigetsu record and/or equivalent sections of
marine sediment depth (following rules defined by Sarnthein et al., 2007).

On visual inspection a plateau boundary is assigned to the break point between the low
to zero or reversed slope of a $^{14}$C plateau and the normally high regression slope of the
$^{14}$C concentration jump that separates two consecutive plateaus (Figs. 1 and S1). More
precisely, a boundary marks the point, where the $^{14}$C curve exceeds the scatter band of





the plateau either crossing the upper or lower envelope line. Thus the boundary is
chosen about halfway between the last [14]C age within a plateau band and the next
following age outside the scatter band (Figs. 1 and 2). Both on the previously varve-
based and the now U/Th-based model age scale (Bronk Ramsey et al., 2012) most [14]C
dates of the Lake Suigetsu section are spaced at intervals of <10–60 yr from 10 to 15
cal. ka and ones of 20–140 yr between 15 and 29 cal. ka (Fig. 1). Thus the uncertainty
of a plateau boundary age assigned halfway between two [14]C ages nearby inside and
outside a plateau's scatter band would, on average, amount to ±10–±70 cal. yr.

In principle, the calendar age uncertainties of marine [14]C plateau boundaries are
treated likewise: After being tuned to those in the Suigetsu [14]C record, the uncertainties
are deduced for the position of all plateaus of a suite within the uncertainty envelope of
the U/Th model-based age calibration. Hence the estimates of total age uncertainty
present the square root of the squared uncertainty of the calibrated age of each
plateau boundary at Suigetsu plus that of the marine record, where variable depth
spacing of [14]C ages is converted into average time spans.

3. DISCUSSION
*3.1 Origin of short-term structures in the atmospheric [14]C record: Changes in cosmogenic*
*[14]C production versus changes in ocean dynamics*

Besides possible climate influences, variations in [10]Be deposition in ice cores reflect past
changes in [10]Be production as a result of changes in solar activity and the strength of the
Earth's magnetic field (Adolphi et al., 2018). Correspondingly, the changes in [10]Be also
reflect past changes in the cosmogenic production of [14]C. If we accept to omit

462 assumptions on the modulation of past ¹⁴C concentrations by changes in the global carbon

463 cycle over last glacial-to-deglacial times we can calculate the atmospheric ¹⁴C changes

464 with a carbon cycle model and convert it into ¹⁴C ages (Fig. 4), although being aware that

465 carbon cycle changes are prominent and necessarily modify the ¹⁰Be-based ¹⁴C record if

466 included correctly into the modeling. Between 10 and 13.5 cal. ka, the modeled ¹⁴C record

467 displays a number of plateau structures that show a decent match with Suigetsu-based

468 atmospheric ¹⁴C plateaus. Between 15 and 29 cal. ka, however, ¹⁰Be-based ¹⁴C plateaus

469 are far more rare and/or less pronounced, except for a distinct equivalent of Plateau 6a,

470 that is, most plateaus are far shorter than those displayed in the suite of atmospheric ¹⁴C

471 plateaus of Lake Suigetsu (e.g., plateaus near to the top 2a, 2b, and top 5a of Suigetsu

472 plateaus). On the whole, the structures show little coherence, thus indicating that any

473 direct relationship between variations in cosmogenic ¹⁴C production and the Suigetsu

474 plateau record is obscured by the carbon cycle, uncorrected climate effects on the ¹⁰Be

475 deposition and/or noise in the ¹⁴C data.

476

477 On the other hand, three long ¹⁴C plateaus (no. 2a, 1, and Top YD) that dominate the ¹⁴C

478 record during deglacial times (Table 2 and Fig. 6) may be ascribed to coeval brief periods

479 being marked by a short-term major rise in global ocean degassing (Marcott et al. 2014).

480 We thus assume that these events which induced a rapid rise in ¹⁴C depleted atmospheric

481 $CO_2$ may be linked to a variety of fast changes such as that of sea ice cover in the

482 Southern Ocean and/or changes in the salinity and buoyancy of surface waters in high

483 latitudes. These factors control upwelling and meridional overturning of deep waters, in

484 particular, as found in the Southern Ocean (Chen et al., 2015) and/or North Pacific (Rae et

485 al. 2014, Gebhardt et al., 2008). Such events of changes in MOC geometry and intensity

486 may be responsible for ocean degassing and the ¹⁴C plateaus as outlined below.






In an extreme case, ventilation ages in the Southern Ocean near New Zealand (SO213-76
in Fig. S4; Küssner et al., 2019, in prep.) drop from 4000 years (~60 % of the
contemporaneous level of past atmosphere 1.4 'Fraction of Modern Carbon' [FMC] at that
time leading to 1.4 x 0.6 = 0.84 FMC) to 1000 years (equal to 88 % of past atmosphere
FMC) around 18 cal. ka with an otherwise constant atmospheric $^{14}C$ of 1.4 FMC. This
implies an increase to 1.4 x 0.88 = 1.232 FMC of local deep ocean $^{14}C$ at this site. The
concentration difference of ~0.4 FMC means a major $^{14}C$ shift in DIC at that very MOC key
region of the deep Southern Ocean (Rae and Broecker, 2018) over 200 yr. This enhanced
mixing of the Southern Ocean and a similar mixing event in the North Pacific (MD02-2489;
Fig. S4) may have triggered – with phase lag – two trends in parallel, (1) a rise in
atmospheric $CO_2$, in part abrupt (*sensu* Chen et al., 2015; Menviel et al., 2018), and (2) a
gradual enrichment in $^{14}C$ depleted atmospheric carbon, reflected as $^{14}C$ plateau.

By contrast, there is little information for the origin of peak glacial $^{14}C$ plateaus no. 4 to 11.
Some of them may possibly be tied to major short-term warmings / MOC modifications in
the North Atlantic such as that during plateau 'YD', during plateaus no. 3 (at the onset of
Antarctic warming; e.g., Kawamura et al., 2007) and no. 8 (on the U/Th-based age scale;
Table 2). These warmings were probably linked to enhanced overturning and short-term
degassing of $^{14}C$ depleted deep waters in the North Atlantic. However, the causal links of
various further peak glacial plateaus to events in ocean MOC still remain to be uncovered.

*3.2 $^{14}C$ plateau boundaries – A suite of narrow-spaced age tie points to rate short-term*
*changes in marine sediment budgets, chemical inventories, and climate 29–10 cal. ka*

In continuation of previous efforts (Sarnthein et al., 2007 and 2015) the tuning of high-
resolution $^{14}$C records of ocean sediment cores to the new age-calibrated atmospheric $^{14}$C
plateau boundaries now makes it possible to establish a 'rung ladder' of ~30 age tie points
covering the time span 29 – 10.5 cal. ka. On a global scale these tie points show a time
resolution of several hundred to thousand years, now used to constrain the chronology
and potential leads and lags of any kind of event that occurred during peak glacial and
deglacial times (Fig. 1). The locations of the 18(20) cores are shown in Fig. S2. The time
histories of the benthic and planktic reservoir ages are summarized in Figs. S3 and S4 and
the information these provide is discussed below.

In particular, five examples show the power and value of additional information obtained
by means of the $^{14}$C plateau-tuning method. (i) Signals of the onset of northern
hemisphere deglaciation can now be distinguished in detail from the subsequent beginning
of deglaciation in the southern hemisphere (Kawamura et al., 2007; Küssner et al., 2019 in
prep.). (ii) A multicentennial-scale phase lag has been specified for the end of the Antarctic
Cold Reversal (ACR) vs. the onset of the Younger Dryas cold spell (Küssner et al., 2019 in
prep.), a finding important to further constrain the details of 'bipolar see-saw' (Stocker and
Johnsen, 2003). (iii) Signals of deep-water formation in the subpolar North Pacific can now
be separated from signals originating in the North Atlantic (Rae et al. 2014; Sarnthein et
al., 2013). In this way we now can specify and tie major short-lasting reversals in Atlantic
and Pacific MOC on a global scale. (iv) Signals of deglacial meltwater advection can now
be distinguished from short-term interstadial warmings in the northern subtropical Atlantic,
which helps to locate meltwater outbreaks far beyond the well-known Heinrich belt of ice-
rafted debris (Balmer and Sarnthein, 2018). (v) As outlined above, the timing of marine $^{14}$C
plateaus can now be compared in detail with that of deglacial events of the atmospheric



CO$_2$ rise independently dated by means of ice core-based stratigraphy (Table 2; Fig. 6).
These linkages enable a better understanding of deglacial changes in deep-ocean MOC
once the suite of [14]C plateaus has been properly tuned at any particular ocean site.

Furthermore, the refined scale of age tie points reveals unexpected details for changes in
the sea ice cover of high latitudes as reflected by anomalously high [14]C reservoir ages
(e.g. north of Iceland and near to the Azores Islands) and for the evolution of Asian
summer monsoon in the northern and southern hemisphere as reflected by periods of
reduced sea surface salinity (e.g., Sarnthein et al., 2015; Balmer et al., 2018). Finally, the
plateau-based high-resolution chronology has led to a detection of numerous millennial-
scale hiatuses (e.g., Sarnthein et al., 2015; Balmer et al., 2016; Küssner et al., 2019 in
prep.) previously undetected by conventional, e.g., *AnalySerie*-based methods (Paillard et
al. 1996) of stratigraphic correlation (Fig. S4). In turn, the hiatuses give intriguing new
insights into past changes of bottom current dynamics linked to different millennial-scale
geometries of overturning circulation and climate change such as in the South China Sea
(Sarnthein et al., 2013 and 2015), in the South Atlantic (Balmer et al. 2016) and southern
South Pacific (Ronge et al., 2019).

Clearly, the new 'rung ladder' of closely-spaced chronostratigraphic tie points has evolved
to a tool indispensable to uncover functional chains in paleoceanography, that actually
have controlled events of climate change over glacial-to-deglacial times.

*3.3  Definition and origin of Zoophycos burrows: A key foe of high-resolution stratigraphy*
*in Pleistocene sediment records turned into an ally?*

The *Zoophycos* producer displaces planktonic foraminifera tests, each marked by the
[14]C age of its shell formation, down to deep sediment levels, hence may severely bias
the faunal and isotopic composition and in particular, the [14]C age of the ambient host
sediment if (parts of) a Zoophycos burrow is picked in a sample. The well-defined 'rung
ladder' of [14]C plateaus defined in the host sediment now provides a clear yardstick both
for the relative chronostratigraphic displacement of the 'outlier' foraminiferal specimens
downcore in the host sediment and in particular, for the precise age of the source level
of these tests, that is the real time when a burrow was produced.

In continuation of previous studies (Löwemark and Grootes, 2004) Küssner et al.
(2018) demonstrated that *Zoophycos*-based vertical grain transport may reach down to
sediment depths of 150 cm. In particular, they showed that *Zoophycos* burrows formed
during brief episodes of enhanced burrowing activity that coincided with a marked drop
in sedimentation rate, that is, with events of reduced benthic nutrient supply. Thus the
"foe" Zoophycos may help corroborate reconstructed changes in climate and MOC.

*3.4 Empiric vs. model-based [14]C reservoir ages acting as tracer of past changes in*
*surface ocean dynamics and as incentive for further model refinements*

The tuning of [14]C plateau boundaries presents a technique unique to establish a suite
of highly resolved and robust age tie points on short and long time scales in [14]C-dated
marine sediment sections wherever retrieved in the global ocean (Fig. S2a). In
addition, and likewise intriguing, [14]C plateau tuning results in a suite of changing [14]C
reservoir ages over time, prime tracers of past oceanography of local surface waters
and a data set crucial to deduce past apparent deep-water ventilation ages (e.g.,



Muglia et al., 2018; Cook and Keigwin, 2015; Balmer and Sarnthein, 2018). Two
aspects help to sort out short-term climate-driven intra- and inter-plateau changes in
[14]C reservoir age, (i) the evaluation of individual reservoir ages is solely based on
judging a complete suite of plateaus, (ii) our experience shows that different climate
regimes in control of changes in surface ocean dynamics generally occurred on (multi-)
millennial time scales (e.g., YD, B/A, HS-1), whereas atmospheric [14]C plateaus hardly
lasted longer than a few hundred up to ~1000 yr. Thus intra-plateau changes in [14]C
reservoir age are less likely, but indeed may amputate and/or deform a plateau to be
checked in detail for each suite of [14]C plateaus (Sarnthein et al., 2007, 2015).

To recap, the atmospheric [14]C plateaus of Suigetsu provide a suite of up to 18 reference
plateaus over the time span 10 – 29 cal. ka (Fig. 1). In marine sediment cores the [14]C
reservoir age of past surface waters is inferred from the difference between the average
[14]C age of an atmospheric [14]C plateau and that of a coeval [14]C plateau analyzed on
monospecific planktic foraminifera (Sarnthein et al., 2007). In low-to-mid latitudes our [14]C
records are based on *G. bulloides*, *G. ruber,* or *G. sacculifer* with habitat depths of 0–
80/120 (Jonkers and Kucera, 2017). In high latitudes, most [14]C records are derived from
*N. pachyderma* (s) living at 0–200 m depth (Simstich et al., 2003). Averaging of [14]C ages
within a [14]C plateau helps to bypass the analytical noise in [14]C records such as short-term
apparent [14]C age reversals and to deduce the regional evolution of planktic [14]C reservoir
ages with semi-millennial-scale resolution. Nine plateaus are located in the LGM, 18–27
cal. ka (Fig. 1). Here, plankton-based reservoir ages show analytical uncertainties of >200
to >300 yr each. By comparison, short-term temporal variations in reservoir age reach
200–400 yr, occasionally up to 600 yr, in particular, close to the end of the LGM (Table 3).



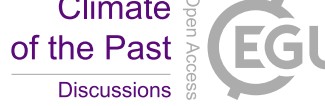

To better decode the informative value of LGM empiric [14]C reservoir ages we compare
them with estimates generated by a General Circulation Model (GCM) of ocean surface
waters (model of Muglia et al., 2018; 0–50 m w.d.; Fig. 7 and Suppl. Fig. S3d), an
approach similar to that of Toggweiler et al. (2019) applied to modern reservoir ages of the
global ocean. Low LGM values (300–750 yr) supposedly document an intensive exchange
of surface waters with atmospheric $CO_2$, most common in model- and foraminifera-based
estimates of the low- and mid-latitude Atlantic. Low empiric values also mark LGM waters
in mid to high latitudes off Norway and off middle Chile, that is, close to sites of potential
deep and/or intermediate water formation. Off Norway and in the northeastern Atlantic,
model-based reservoir ages of Muglia et al. (2018) largely match the empiric range. This is
no proof yet for model quality, since the uncertainty envelopes (±560 yr for data shown in
Fig. 7b; r = 0.59) generally far exceed the spatial differences calculated for the empiric
data. Contrariwise, model-based reservoir ages reproduce only poorly the low plankton-
based estimates off Central Chile and values in the Western Pacific and Southern Ocean.

In part, the differences may be linked to problems like insufficient spatial resolution along
continental margins and/or the estimates of a correct location and extent of seasonal sea
ice cover used as LGM boundary condition such as east off Greenland, in the subpolar
N.W. Pacific, and off Southern Chile, where sea ice hindered the exchange of atmospheric
carbon (per analogy to that of temperature exchange, as recorded by Sessford et al,
2019). In turn, model estimates are compared to [14]C signals of planktic foraminifera that
mostly formed during summer only, when large parts of the Nordic Seas were found ice-
free (Sarnthein et al., 2003). Hence, models may need to better constrain local and
seasonal sealing effects of LGM sea ice cover.





In general, however, the foraminifera-based reservoir age estimates for our sites that
represent various hydrographic key regions in the high-latitude ocean appear much higher
than model-derived values. Deviations reach up to 1400 yr, in particular in the Southern
Ocean. In part, the discrepancies may result from the fact that present models may not yet
be suited to capture values with great small-scale variability. Here, model-based reservoir
ages appear far too low in LGM regions influenced by regional upwelling such as the
South China Sea then governed by an estuarine overturning system (Wang et al., 2005;
Fig. 8), by coastal upwelling off N.W. Australia (Xu et al., 2010; Sarnthein et al., 2011), or
by a melt water lid such as off eastern New Zealand (Bostock et al., 2013; Küssner et al.,
2019, in prep.). Local oceanic features are likely to be missed in model simulation, for
example, by comparison to details in modern current geometry displayed by Yashayaev et
al. (2015) because of a model resolution still too coarse, a lack that suggests directions for
future model refinement. More narrow-spaced empiric data will help to weight more
correctly and develop the skill of models to capture past $^{14}$C reservoir ages.

Various differences amongst plankton- and model-based reservoir ages may result from
differential seasonal habitats of the different planktic species analyzed that, in turn, may
trace different surface and subsurface water currents. Pertinent details are largely
unknown for the modern scenario because of the 'bomb effect', likewise no pertinent data
exist yet for the LGM. However, distinct interspecies differences were found in the
northern Norwegian Sea for the time span of the Preboreal $^{14}$C plateau, 9.6–10.2 cal. ka
(Sarnthein and Werner, 2018). These differences amount up to 600 yr amongst paired $^{14}$C
records of Arctic *Turborotalita quinqueloba* dominantly formed close to the sea surface
during peak summer, Arctic *Neogloboquadrina pachyderma* formed in subsurface waters,
and the subpolar species *N. incompta* mainly advected from the south by Norwegian



Current waters well mixed with the atmosphere during peak winter. This makes closer
specification of model results as product of different seasonal extremes a further target.

*3.5 Plankton-based $^{14}$C reservoir ages – A prime database to estimate past changes in*
*the $^{14}$C ventilation age of deep waters, ocean MOC, and DIC for past states of the ocean*

'Raw' apparent benthic ventilation ages (in $^{14}$C yr; 'raw' *sensu* Balmer et al., 2018) express
the difference between the (coeval) atmospheric and benthic $^{14}$C levels measured at any
site and time of foraminifer deposition. These ages are the sum of (1) the planktic reservoir
age of the $^{14}$C plateau that covers a group of paired benthic and planktic $^{14}$C ages and (2)
the (positive or negative) $^{14}$C age difference between any benthic $^{14}$C age and the average
$^{14}$C age of the paired planktic $^{14}$C plateau. The benthic ventilation ages necessarily rely on
the high quality of $^{14}$C plateau-based chronology, since the atmospheric $^{14}$C level has been
subject to substantial short-term changes over glacial-to-deglacial times. Necessarily, the
ventilation ages include a mixing of different water masses that might originate from
different ocean regions and may contribute differential $^{14}$C ventilation ages, an unknown
justifying the modifier 'apparent'.

In a further step, the $\Delta\Delta^{14}$C equivalent of our 'raw' benthic ventilation age may be adjusted
to changes in atmospheric $^{14}$C that occurred over the (short) time span between deep-
water formation and benthic sediment deposition (e.g., Balmer and Sarnthein, 2018; Cook
and Keigwin, 2015). In most cases, however, this second step is omitted since its
application usually does not imply any major modification of the ventilation age estimates
(Fig. S4a; Skinner et al., 2017; Sarnthein et al., 2013).





On the basis of $^{14}$C plateau tuning we now can rely on 18 precisely dated records of
apparent benthic $^{14}$C ventilation ages (Fig. S4a-c) to reconstruct the global geometry of
LGM and HS-1 deep and intermediate water circulation as summarized in ocean transects
of Figs. 8 and 9. The individual matching of our 20 planktic $^{14}$C plateau sequences with
that of the Suigetsu atmospheric $^{14}$C record is displayed in Sarnthein et al. (2015), Balmer
et al., (2016), Küssner et al. (2019, in prep.), and Ausin et al. (2019, in prep.). In addition,
robust estimates of past reservoir ages are obtained for 4 planktic and benthic $^{14}$C records
from paired atmospheric $^{14}$C ages of wood chunks (Rafter et al., 2018; Zhao and Keigwin,
2018; Broecker et al., 2004).

*3.5.1 — Major features of ocean meridional overturning circulation during LGM (Fig. 9)*

Off Norway and near the Azores Islands very low benthic $^{14}$C ventilation ages of <100–750
yr suggest ongoing deep-water formation in the LGM northern North Atlantic reaching
down to more than 3000–3500 m water depth, with a flow strength possibly similar to
today (and a coeval deep countercurrent of old waters from the Southern Ocean flowing
along the East Atlantic continental margin off Portugal). This pattern clearly corroborates
the assembled benthic $\delta^{13}$C record showing plenty of elevated $\delta^{13}$C values for the
northwestern, eastern and central North Atlantic (Sarnthein et al., 1994; Millo et al., 2006;
Keigwin and Swift, 2017). Irrespective of unspecified potential zonal variations in deep-
water ventilation age at mid latitudes and different from a number of published models
(e.g., Ferrari et al., 2014; Butzin et al., 2017) this 'anti-estuarine' pattern has been
confirmed by MIROC model simulations (Gebbie, 2014; Sherriff-Tadano et al., 2017,
Yamamoto et al., 2019) and, independently, by $\varepsilon_{Nd}$ records (Howe et al., 2016; Lippold et
al., 2016). The latter suggest an overturning of AMOC possibly even stronger than today,



in particular due to a 'thermal stronghold' overlooked in other model simulations. Muglia et
al. (2018) tested in their model also a number of different AMOC flows with a strength of 6,
8, 9, and 13 Sv each, with estimates of 13 Sv appearing somewhat more consistent with
our results.

In contrast to the northern North Atlantic, North Atlantic Deep Waters and old Circumpolar
(CP) deep waters in the subpolar South Atlantic show an LGM $^{14}$C ventilation age of
~3640 yr, finally rising up to 4100 yr (Fig. 9). These waters were upwelled and admixed
from below to surface waters near to the sub-Antarctic Front during terminal LGM (Fig.
S4b; Skinner et al., 2010; Balmer and Sarnthein, 2016; model of Butzin et al., 2012).

In the southwestern South Pacific abyssal, in part possibly Antarctic-sourced waters (Rae
and Broecker, 2018) likewise show high apparent $^{14}$C ventilation ages that rise from 3900
to 4800 yr over the LGM, in particular close to its end (Figs. 9 top and S4c) ($^{14}$C dates of
Ronge et al., 2016, modified by planktic $^{14}$C reservoir ages of Küssner et al., 2019). A
vertical transect of benthic $\delta^{13}$C (McCave et al., 2008) suggests that the abyssal waters
were overlain by CP waters, separated by pronounced stratification near ~3500–4000 m
water depth. In part, the CP waters stemmed from North Atlantic Deep Water. Probably,
their apparent ventilation age came close to 3900–4500 yr, similar to the values found in
the southern South Atlantic. East of New Zealand the CP waters entered the deep western
Pacific and spread up to the subpolar North Pacific, where LGM $^{14}$C ventilation ages
reached 3700 yr.

Similar to today, the MOC of the LGM Pacific was shaped by estuarine geometry, probably
more weakened than today (Du et al., 2018) and more distinct in the far northwest than in





the far northeast. This geometry resulted in an upwelling of old deep waters in the
subarctic Northwest Pacific, here leading to a [14]C reservoir age of ~1700 yr for surface
waters at terminal LGM. On top of the Lower Pacific Deep Waters we may surmise Upper
Pacific Deep Waters that moved toward south (Fig. 9, top panel).

The Pacific deep waters were overlain by Antarctic / Pacific Intermediate Waters (IW) with
LGM [14]C ventilation ages as low as 1400–1600 yr, except for a shelf ice-covered site at
the southern tip of Chile with IW ages of 2460–3760 yr, possibly a result of local upwelling
of CP waters. In general, however, the low values of Pacific IW are similar to those
estimated for South Atlantic IW and likewise reflect a vivid exchange with atmospheric $CO_2$
in their source regions in the Southern Ocean (Skinner et al., 2015).

When entering and crossing the entrance sill to the marginal South China Sea the 'young'
IW were mixed with 'old' CP waters entrained from below, here leading to [14]C ventilation
ages of 2600–3450 yr (Figs. 8 and S4d). The LGM South China Sea was shaped by an
estuarine-style overturning system marked by major upwelling near to its distal end in the
far southwest (Wang L. et al., 1999). This upwelling led to planktic [14]C reservoir ages as
high as 1200–1800 yr, values rarely found elsewhere in surface waters of low latitudes.

Our wide-spaced distribution pattern of 18 [14]C ventilation ages (plus 4 values based on
paired wood chunks) in Fig. 9 agrees only in part with the circulation patterns suggested
by the much larger datasets of [14]C ventilation ages compiled by Skinner et al. (2017) and
Zhao et al. (2018). Several features in Fig. 9 directly deviate, e.g., the ages we derive for
the North Atlantic and mid-depth Pacific. These deviations may be linked to both the
different derivation of our [14]C ventilation age estimates and the details of our calendar-year



chronology now based on the narrow-standing suite of [14]C plateau-boundary ages. The
quality of our [14]C reservoir ages of surface waters also controls the apparent ventilation
age of deep-waters, as it results from direct subtraction of a short-term [14]C average of an
atmospheric [14]C plateau from the paired benthic [14]C value, that is coeval with the planktic
[14]C plateau during the time of benthic foraminifera growth.

*3.5.2 — Major features of meridional overturning circulation during early HS-1 (Fig. 9)*

Near the onset of deglacial Heinrich Stadial 1 (HS-1; ~18–14.7 cal. ka) major shifts in [14]C
ventilation age suggest some short-lasting but fundamental changes in the circulation
geometry of the deep ocean, a central theme of marine paleoclimate research (Fig. 7,
lower panel of Fig. 9, and Figs. S2, S4a and b). Deep waters in the eastern Nordic Seas,
west of the Azores Islands, and off northern Brazil show a rapid rise to high [14]C ventilation
ages of ~2000–2500 yr and up to 4000 yr off Brazil, values that give first proof for a brief
switch from 'anti-estuarine' to 'estuarine' circulation that governed the central North
Atlantic and Norwegian Sea during early HS-1. This geometry continued – except for a
brief but marked and widespread event of recurring NADW formation near 15.2 ka – until
the very end of HS-1 near 14.5 ka (Fig. S4a; Muschitiello et al., 2019). The MOC switch
from LGM to HS-1 is in line with changes depicted in paired benthic $\delta^{13}$C data (Sarnthein
et al., 1994), but not confirmed by the coeval $\varepsilon_{Nd}$ record that suggests a constant source of
'mid-depth waters', with the $\delta^{13}$C drop being simply linked to higher ages (Howe et al.,

783 2018).


Conversely, benthic [14]C ventilation ages in the northeastern North Pacific (Site MD02-
2489) show a coeval and distinct but brief minimum of 1050-1450 yr near 3640 m w.d.




during early HS-1 (~18.1–16.8 ka; Figs. 9, S2, and S4d). This minimum was produced by
extremely small benthic-planktic age differences of 350–650 yr and provides robust
evidence for a short-lasting event of deep-water formation, that has flushed the north-
eastern North Pacific down to more than 3640 m w.d. (Gebhardt et al., 2008; Sarnthein et
al., 2013; Rae et al., 2014). Similar circulation geometries were reported for the Pliocene
(Burls et al., 2017). 'Young' Upper North Pacific Deep Waters (North Pacific Intermediate
Waters *sensu* Gong et al., 2019) then penetrated as 'western boundary current' far south,
up to the northern continental margin of the South China Sea (Fig. 8b and S4d). The short-
lasting North Pacific regime of anti-estuarine overturning was similar to that we find in the
modern and LGM Atlantic and, most interesting, simultaneous with its estuarine episode.

Recent data on benthic-planktic $^{14}$C age differences (Du et al., 2018) precisely recover our
results at ~680 m w.d. off southern Alaska. However, they do not depict the 'young' deep
waters at their Site U1418 at ~3680 m w.d., as corroborated by a paired autigenic $\varepsilon_{Nd}$
maximum suggesting a high local bottom water age nearby. We assume that the amazing
difference in local deep-water ventilation ages is due to small-scale differences in the
effect of Coriolis forcing at high latitudes between a site located directly at the Alaskan
continental margin (U1418; Fig. 9b) and that on the distal Murray Sea Mount in the 'open'
Pacific (MD02-2489; Fig. S4d), which has been washed by a plume of newly formed North
Pacific deep waters probably stemming from the Bering and/or Ochotsk Seas. In contrast,
the incursion of almost 3000 yr old deep waters from the Southern Ocean has continued
along the continental margin all over HS-1. In summary we may conclude that the
geometry of ocean MOC was briefly reversed in the 'open' North Pacific over almost 1500
years during HS-1, far deeper than suggested by previous authors (e.g., Okazaki et al.,





2012; Gong, S., et al. 2019), but similar to changes in geometry first proposed by Broecker
et al. (1985) then, however, for an LGM ocean.

*3.5.3 — Deep-Ocean DIC inventory*

Apart from the changing geometries in ocean MOC, the global set of $^{14}$C plateau-based,
hence refined estimates of apparent $^{14}$C ventilation ages (Fig. 9) has ultimately revealed
new insights into glacial-to-deglacial changes in the ocean DIC inventories (Sarnthein et
al., 2013). On the basis of GLODAP data (Key et al., 2004) any drop in $^{14}$C concentration
(i.e., any rise in average $^{14}$C ventilation age) of modern deep waters is tied linearly to a
rise of carbon (DIC) dissolved in deep ocean waters below ~2000 m, making for 1.22
micromole C / -1 ‰ $^{14}$C. By and large, GCM and box model simulations of Chikamoto and
Abé-Ouchi (2012) and Wallmann et al. (2016) suggest that this ratio may also apply to
LGM deep-water circulation, when apparent $^{14}$C ventilation ages in the Southern Ocean
increased significantly (from 2400 up to ~5000 yr) and accordingly, thermohaline
circulation was more sluggish and transit times of deep waters extended. Accordingly, a
'back-of-the-envelope' calculation of LGM ventilation age averages in the global deep
ocean suggests an additional carbon absorption of 730–980 Gt (Sarnthein et al., 2013).
This estimate can easily accommodate the glacial transfer of ~200 Gt C from the
atmosphere and biosphere, moreover, may also explain 200–450 Gt C then most probably
removed from glacial Atlantic and Pacific intermediate waters. These estimates offer an
independent evaluation of ice core-based data, other proxies, and model-based data on
past changes in the global carbon cycle (e.g., Menviel et al., 2018).

4. SOME CONCLUSIONS



– Regarding the upgraded Plateau-Tuning, despite some analytical scatter, $^{14}$C ages for
the top and base of Lake Suigetsu-based atmospheric $^{14}$C plateaus and coeval planktic
$^{14}$C plateaus do not present statistical 'outliers' but real age estimates that are reproduced
by tree ring-based $^{14}$C ages over the interval 10–13 cal. ka and further back.
– Hulu U/Th model-based ages of $^{14}$C plateau boundaries of the Suigetsu atmospheric $^{14}$C
record appear superior to those derived from microscopy-based varve counts only, since
U/Th model-based ages match far more closely the age deduced from XRF-based varve
counts for a crucial test case of lower plateau boundary 2b in the early deglacial, moreover,
the age assigned to the Laschamp event.
– During deglacial times, several $^{14}$C plateaus paralleled a rise in air-sea gas exchange,
and, in turn, distinct changes in ocean MOC. By contrast, changes in cosmogenic $^{14}$C
production rarely offer a complete explanation for the plateaus identified in the Suigetsu
$^{14}$C data under discussion.
– In total, $^{14}$C plateau boundaries in the range 29–10 cal. ka provide a suite of ~30 tie
points to establish – like chronological ladder rungs – a robust global age control for deep-
sea sediment sections and global stratigraphic correlations of last glacial to deglacial
climate events, 29–10 cal. ka. U/Th model ages confine the cal. age uncertainty of
Suigetsu plateau boundaries assigned halfway between two $^{14}$C ages nearby inside and
outside a plateau's scatter band to less than ±50–±70 yr.
– Regarding oceanographic implications, $^{14}$C ages in a sediment section that form a
separate population of $^{14}$C outliers clearly distinct from the 'normal' $^{14}$C plateau suite help
to trace the reach and origin of Zoophycos burrows, a key 'foe' of high-resolution
stratigraphy in marine sediment cores, and allow for inferences on their origin in a major
reduction in sediment and nutrient supply.



– The difference in $^{14}$C age between coeval atmospheric and planktic $^{14}$C plateaus
presents a robust tracer of planktic $^{14}$C reservoir ages and their temporal and spatial
variability, for the LGM and HS-1 now established for 18/20 sediment sites.
– Paired reservoir ages obtained from different planktic species document the local
distribution patterns of different surface water masses and prevailing foraminiferal habitats
at different seasons.
– A new, more reliable set of deep-water $^{14}$C ventilation ages can be derived on the basis
of our robust planktic $^{14}$C reservoir ages. These ventilation ages reveal geometries of LGM
overturning circulation, the main traits of which are similar to those of today. In contrast,
$^{14}$C ventilation ages of early HS-1 suggest an almost 1500 yr long event of widely reversed
circulation patterns marked by deep-water formation and brief flushing of the northern
North Pacific and estuarine circulation geometry in the northern North Atlantic.
– Increased glacial $^{14}$C ventilation ages and carbon (DIC) inventories of ocean deep
waters suggest an LGM drawdown of about 850 Gt C into the deep ocean and an early
deglacial abrupt carbon release to the atmosphere during HS-1 (Sarnthein et al., 2013).
– Comparison of planktic and model-based reservoir age estimates reveals some major
discrepancies, in particular at sites in middle to high latitudes, and pointe the way to further
model refinements to make the models better reflect the real complex patterns of ocean
circulation, including seasonality.

ACKNOWLEDGMENTS
We owe sincere thanks for plenty of stimulations to the 23$^{rd}$ International Radiocarbon
Conference in Trondheim, in particular to M-J. Nadeau, and to the IPODS–OC3 workshop
in Cambridge U.K, 2018, convened by A. Schmittner and L. Skinner. Moreover, we thank
for most valuable basic discussions with R. Staff, Glasgow, J. Southon, Irvine CA, and M.





Butzin, AWI Bremerhaven, who kindly helped us to discuss the comparison of his model
results, and S. Beil, Kiel, for computer assistance. Over the last three years, G.
Mollenhauer measured with care hundreds of supplementary $^{14}$C ages in her MICADAS
laboratory at AWI Bremerhaven. This study obtained long lasting special support from R.
Tiedemann and his colleagues at the AWI Bremerhaven.

**Author contribution**
All authors contributed data and valuable suggestions to write up this synthesis. MS
and PG designed the outline of this manuscript. KK, BA, TE and MS provided new
marine 14C records in addition to records previously published. GS displayed the
details of Suigetsu varve counts. RM provided a $^{10}$Be-based $^{14}$C record and plots of
raw $^{14}$C data sets of Suigetsu und Hulu Cave. Discussions amongst PG, RM, GS and
MS served to select U/Th-based model ages as best-possible time scale.

**Data availability**
Primary radiocarbon data of most sites are available at PANGAEA de, except for the
$^{14}$C data of 5 marine cores still under publication by Küssner et al. and Ausin et al. (in
prep.; see caption of Fig. S4).

REFERENCES  (99)

905        Adkins, J. F. and Boyle, E. A.: Changing atmospheric Δ$^{14}$C and the record of

paleoventilation ages. Paleoceanography, 12(3), 337–344, 1997.

907        Adolphi, F., Bronk Ramsey, C., Erhard, T., Lawrence Edwards, R., Cheng, H.,

Turney, C.S.M., Cooper, A., Svensson, A., Rasmussen, S.O., Fischer, H., and
Muscheler, R.: Connecting the Greenland ice-core and U/Th timescales via
cosmogenic radionuclides: testing the synchroneity of Dansgaard–Oeschger events.
Clim. Past, 14, 1755–1781. https://doi.org/10.5194/cp-14-1755-2018, 2018.

912        Alves, E.Q., Macario, K., Ascough, P., and Bronk Ramsey, C.: The worldwide

marine radiocarbon reservoir effect: definitions, mechanisms, and prospects. Review of
Geophysics, 56, https://doi.org/10.1002/2017RG000588, 2018.
Alveson, E.Q.: Radiocarbon in the Ocean, EOS, 99,
https://doi.org/10.1029/2018EO095429, 2018.
Ausin, B., Haghipour, N., Wacker, L., Voelker, A. H. L., Hodell, D., Magill, C., et al.:
Radiocarbon age offsets between two surface dwelling planktonic foraminifera species
during abrupt climate events in the SW Iberian margin. Paleoceanography and
Paleoclimatology, 34. https://doi.org/10.1029/2018PA003490, 2019.
Balmer, S., Sarnthein, M., Mudelsee, M., and Grootes, P. M.: Refined modeling and
14C plateau tuning reveal consistent patterns of glacial and deglacial 14C reservoir
ages of surface waters in low-latitude Atlantic. Paleoceanography, 31.
https://doi.org/10.1002/2016PA002953, 2016.
Balmer, S. and Sarnthein, M.: Glacial-to deglacial changes in North Atlantic melt-
water advection and deep-water formation – Centennial-to-millennial-scale $^{14}$C records
from the Azores Plateau. Geochim. Cosmochim. Acta, 236, 399-415,
https://doi.org/10.1016/j.gca.2018.03.001, 2018.
Berger W.H. and Keir, R.S.: Glacial-Holocene changes in atmospheric $CO_2$ and the
deep-sea record. J.E. Hansen, T. Takahashi (Eds.), Geophysical Monograph, 29,
American Geophysical Union, Washington, DC, pp. 337–351, 1984.
Bostock, H.C., Barrows, T.T., Carter, L., Chase, Z., Cortese, G., et al.: A review of
the Australian – New Zealand sector of the Southern Ocean over the last 30 ka (Aus-
INTIMATE project). Quaternary Science Reviews 74, 35-57, 2013.
Broecker W.S, Peteet, D.M., and Rind, D.: Does the ocean-atmosphere system have
more than one stable mode of operation? Nature, **315**, 21-26, doi:10.1038/315021a0,

937    1985

Broecker W.S., Barker, S., Clark, E., Hajdas, I., Bonani, G., and Stott, L.: Ventilation
of the Glacial deep Pacific Ocean, Science, 306, 1169–1172, 2004.
Bronk Ramsey, C., Staff, R. A., Bryant, C. L., Brock, F., Kitagawa, H., van der Plicht,
J., Schlolaut, G., Marshall, M. H., Brauer, A., Lamb, H. F., Payne, R. L., Tarasov, P. E.,
Haraguchi, T., Gotanda, K., Yonenobu, H., Yokoyama, Y., Tada, R., and Nakagawa,
T.: A complete terrestrial radiocarbon record for 11.2 to 52.8 kyr B.P., Science, 338,
370–374, 2012.
Bronk Ramsey, C. et al., Radiocarbon, (under review) 2019.
Burls, N.J., Fedorov, A.V., Sigman, D.M., Jaccard, S.L., Tiedemann, R., and Haug,
G.H.: Active Pacific meridional overturning circulation (PMOC) during the warm
Pliocene. Sci. Adv. 2017;3: e1700156, 2017.





Butzin, M., Prange, M., Lohmann, G.: Radiocarbon simulations for the glacial
ocean: The effects of wind stress, Southern Ocean sea ice and Heinrich events. Earth
Planet. Sci. Lett., 235, 45-61, 2005.
Butzin, M., Prange, M., and Lohmann, G.: Readjustment of glacial radiocarbon
chronologies by self-consistent three-dimensional ocean circulation modeling. Earth
Planet Sci. Lett., 317, 177-184, 2012.
Butzin, M., Köhler, P., and Lohmann, G.: Marine radiocarbon reservoir age
simulations for the past 50,000 years. Geophys. Res. Lett., 44, 8473–8480,
doi:10.1002/2017GL074688, 2017.
Chen, T., Robinson, L.F., Burke, A., Southon, J., Spooner, P., Morris, P.J., and Ng,
H.C.: Synchronous centennial abrupt events in the ocean and atmosphere during the
last deglaciation, Science, 349, 1537-1541, 2015.
Cheng, H., Edwards, R.L., Southon, J., Matsumoto, K., Feinberg, J.M., Sinha, A.,
Zhou, W., Li, H., Li, X., Xu, Y., Chen, S., Tan, M., Wang, Q., Wang, Y., and Ning, Y.:
Atmospheric 14C/12C changes during the last glacial period from Hulu Cave, Science,

964    362, 1293-1297, 2018.

Chikamoto, M.O., Abé-Ouchi, A., Oka, A., Ohgaito, R., and Timmermann, A.:
Quantifying the ocean's role in glacial $CO_2$ reductions, Climate of the Past, 8, 545–563,
doi:10.5194/cp-8-545-2012, 2012.
Cook M.S. and Keigwin L.D.: Radiocarbon profiles of the NW Pacific from the LGM
and deglaciation: Evaluating ventilation metrics and the effect of uncertain surface
reservoir ages, Paleoceanography, 30, 174–195, 2015.
Davies, S.M., Davies, P.M., Abbott, Meara, R.H., et al.: A North Atlantic tephro-
stratigraphical framework for 130-60 ka b2k: New tephra discoveries, marine based
correlations, and future challenges, Quaternary Science Rev., 106, 101-121, 2014.
Du, J., Haley, B.A., Mix, A.C., Walczak, M.H., and Praetorius, S.K.: Flushing of the deep
Pacific Ocean and the deglacial rise of atmospheric $CO_2$ concentrations, Nature geoscience,

976    11, 749-755, 2018.

Ferrari, R., Jansen, M.F., Adkins, J.F., Burke, A., Stewart, A.L., and Thompson,
A.F.: Antarctic sea ice control on ocean circulation in present and glacial climates, Proc.
National Academy Science, 111 (24), 8753–8758, 2014.
Gebbie, G.: How much did Glacial North Atlantic Water shoal? Paleoceanography,
29, 190-209, doi:10.1002/2013PA002557, 2014.



Gebhardt, H., Sarnthein, M., Kiefer, T., Erlenkeuser, H., Schmieder, F., and Röhl, U.:
Paleonutrient and productivity records from the subarctic North Pacific for Pleistocene
glacial terminations I to V. Paleoceanography 23, PA4212, 1-21,
doi:10.1029/2007PA001513, 2008.
Gong, S., Lembke-Jene, L., Lohmann, G., Knorr, G., Tiedemann, R., Zou, J.J and
Shi, X.F.: Enhanced North Pacific deep-ocean stratification by stronger intermediate
water formation during Heinrich Stadial 1. Nature Communications, 10: 656.
https://doi.org/10.1038/s41467-019-08606-2, 2019.
Grootes P.M. and Stuiver, M.: Oxygen 18/16 variability in Greenland snow and ice
with 1000 to 100000 year time resolution, J. Geophys. Res.: Oceans (1978–2012)
102(C12), 26455–26470, 1997.
Grootes, P.M. and Sarnthein, M.: Marine [14]C reservoir ages oscillate, PAGES News,

994    14/3, 18-19, 2006.

Howe, J.N.W., Piotrowski, A.M., Noble, T.L., Mulitza, S., Chiessi, C.M., and Bayon,
G.: North Atlantic deep-water production during the last glacial maximum, Nat.
Commun., 7, 11765, 2016. s
Howe, J.N.W., Huang, K-F., Oppo, D.W., Chiessi, C.M., Mulitza, S., Blusztajn, J.,
and Piotrowski, A.M.: Similar mid-depth Atlantic water mass provenance during the
Last Glacial Maximum and Heinrich Stadial 1, Earth Planetary Science Letters, 490,

1001   51-61, 2018.

Jonkers, L. and Kucera, M.: Quantifying the effect of seasonal and vertical habitat
tracking on planktonic foraminifera proxies, Climate of the Past, 13, 573-586, 2017.
Kawamura, K., Parrenin, F., Lisiecki, L., Uemura, R., Vimeux, F., Severinghaus,
J.P., Hutterli, M.A., Nakazawa, T., Aoki, S., Jouzel, J., Raymo, M.E., Matsumoto, K.,
Nakata, H., Motoyama, H., Fujita, S., Goto-Azuma, K., Fujii, Y., and Watanabe, O.:
Northern Hemisphere forcing of climatic cycles in Antarctica over the past 360,000
years, Nature, 448, 912-916, doi:10.1038/nature06015, 2007.
Keigwin, L.D. and Swift, S.A.: Carbon isotope evidence for a northern source of
deep water in the glacial western North Atlantic, PNAS, 114 (11), 2831-2835, 2017.
Key R. M., Kozyr, A., Sabine, C.L., Lee, K., Wanninkhof, R., Bullister, J.L., Feely,
R.A., Millero, F.J., Mordy, C., and Peng, T.-H. (2004) A global ocean carbon climat-
ology: Results from Global Data Analysis Project (GLODAP), Global Biogeochem. Cy.,
18, GB4031, doi:10.1029/2004GB002247, 2004.
Kong, X., Wang, Y., Wu, J., Cheng, H., Edwards, R.L., and Wang, X.: Complicated





responses of stalagmite δ¹³C to climate change during the last glaciation from Hulu
Cave, Nanjing, China, Science in China Ser. D Earth Sciences, 48, (12), 2174-2181,

1018  2005.

Küssner, K., Sarnthein, M., Lamy, F., and Tiedemann, R.: High-resolution radiocarbon-
based age records trace episodes of *Zoophycos* burrowing, Marine Geology, 403, 48-56,
http://doi:10.1016/j.margeo.2018.04.01, 2018.
Küssner, K., Sarnthein, M., Lamy, F., Michel, E., Mollenhauer, G., Siani G., and
Tiedemann, R.: Glacial-to-deglacial reservoir ages of surface waters in the southern
South Pacific, 2019 (in prep.).
Lippold, J., Gutjahr, M., Blaser, P., Christner, E., de Cavalho-Fereira, M.L., Mulitza, S.
et al.: Deep-water provenance and dynamics of the (de)glacial Atlantic meridional
overturning circulation, Earth Planetary Science Letters, 445, 68-78, 2016.
Lisiecki, L.E. and Stern, J.V.: Regional and global benthic d¹⁸O stacks for the last
glacial cycle. Paleoceanography, 31, doi:10.1002/2016PA003002, 2016.
Löwemark, L. and Grootes, P.M.: Large age differences between planktic
foraminifers caused by abundance variations and Zoophycos bioturbation,
Paleoceanography, 19, PA2001, doi:10.1029/2003PA000949, 2004.
Marcott, S.A., Bauska, T.K., Buizert, C., Steig, E.J., Rosen, J.L., Cuffey, K.M.,
Fudge, T.J., Severinghaus, J.P., Ahn, J., Kalk, M.L., McConnell, J.R., Sowers, T.,
Taylor, K.C., White, J.W.C., and Brook, E.J.: Centennial-scale changes in the global
carbon cycle during the last deglaciation, Nature, 514, 616-619,
doi:10.1038/nature13799, 2014.
Marshall, M., Schlolaut, G., Brauer, A., Nakagawa, T., Staff, R.A., Bronk Ramsey,
C., Lamb, H., Gotanda, K., Haraguchi, T., Yokoyama, Y., Yonenobu, H., Tada, R.,
SG06 project members: A novel approach to varve counting using µXRF and X-
radiography in combination with thin-section microscopy, applied to the Late Glacial
chronology from Lake Suigetsu, Japan, Quaternary Geochronology 13, 70-80, 2012.
McCave, I.N., Carter, I., and Hall, I.R.: Glacial-interglacial changes in water mass
structure and flow in the SW Pacific Ocean, Quaternary Science Rev., 27, 1886–1908,

1045  2008.

Matsumoto, K.: Radiocarbon-based circulation age of the world oceans, J. Geophys.
Res.: Oceans 112(C9), C09004. https://doi.org/10.1029/2007JC004095, 2007.
Menviel, L., Spence, P., Yu, J., Chamberlain, M.A., Matear, R.J., Meissner K.J., and
England, M.H.: Southern Hemisphere westerlies as a driver of the early deglacial





atmospheric CO2 rise, Nature communications, 9:2503, DOI:10.1038/s41467-018-
1051    04876-4, 2018.
Millo, C., Sarnthein, M., and Erlenkeuser, M.: Variability of the Denmark Strait Overflow
during the Last Glacial Maximum, Boreas, 35, 50-60, 2006.
Muglia, J., Skinner, L., and Schmittner, A.: Weak overturning circulation and high
Southern Ocean nutrient utilization maximized glacial ocean carbon, Earth and
Planetary Science Letters 496, 47-56, 2018.
Muschitiello, F., D'Andrea, W.J., Schmittner, A., Heaton, T.J., Balascio, N.L.,
deRoberts, N., Caffee, M.W., Woodruff, T.E., Welten, K.C., Skinner, L.C., Simon, M.H.,
and Dokken T.M.: Deep-water circulation changes lead North Atlantic climate during
deglaciation, Nature Communications 10, 1272, doi.org/10.1038/s41467-019-09237-3,
1061    2019.
Naughton, F., Costas, S., Gomes, S.D., Desprat, S., Rodrigues, T., Sanchez Goñi,
M.F., Renssen, H., Trigo, R., Bronk-Ramsey, C., Oliveira, D., Salgueiro, E., Voelker,
A.H.L., and Abrantes, F.: Coupled ocean and atmospheric changes during Greenland
stadial 1 in southwestern Europe, Quaternary Science Reviews, 212, 108-120, 2019.
Nydal R., Lovseth K., and Skogseth F. H.: Transfer of bomb [14]C to the ocean
surface, Radiocarbon 22(3), 626–635, 1980.
Okazaki, Y., Sagawa, T., Asahi, H., Horikawa, K., and Onodera, J.: Ventilation
changes in the western North Pacific since the last glacial period, Climate of the Past, 8,
17-24, doi:10.5194/cp-8-17-2012, 2012.
Paillard, D., Labeyrie, L., and Yiou, P.: Macintosh program performs time-series
analysis, *Eos Trans, AGU,* **77**: 379, 1996.
Rae, JW.B. and W. Broecker, W.: What fraction of the Pacific and Indian oceans'
deep water is formed in the Southern Ocean? Biogeosciences, 15, 3779-3794, 2018.
Rae, J., Sarnthein, M., Foster, G., Ridgwell, A., Grootes, P.M., and Elliott T.: Deep
water formation in the North Pacific and deglacial $CO_2$ rise*, Paleoceanography, 29,
doi:10.1002/2013PA002570, 645–667, 2014.
Rafter, P.A., Herguera, J.-C., and Southon, J.R.: Extreme lowering of deglacial
seawater radiocarbon recorded by both epifaunal and infaunal benthic foraminifera in a
wood-dated sediment core, Climate of the Past 14, 1977–1989, 2018.
Reimer P.J., Bard, E., Bayliss, A., Beck, J. W., Blackwell, P.G., Bronk Ramsey, C.,
Buck, C.E., Cheng, H., Edwards, R.L., and Friedrich, M.: IntCal13 and Marine13
radiocarbon age calibration curves 0–50,000 years cal. BP, Radiocarbon 55, 1869–



1084   1887, 2013.

Reimer, P.J., et al.: The IntCal19 northern hemisphere radiocarbon calibration curve
(0-55 kcal BP), Radiocarbon, 2019 (under review).
Ronge, T. A., Tiedemann, R., Lamy, F., et al.: Radiocarbon constraints on the extent
and evolution of the South Pacific glacial carbon pool, Nature Communications 7:11487,

1089   2016.

Ronge, T.A., Sarnthein, M., Roberts, J., Lamy, F., and Tiedemann, R.: East Pacific
Core PS75/059-2: Glacial-to-deglacial stratigraphy revisited, Paleoceanography and
Paleoclimatology, 34 (4), 432-435, DOI:10.1029/2019PA003569, 2019.
Sarnthein, M., Winn, K., Jung, S.J., Duplessy, J.C., Labeyrie, L., Erlenkeuser, H.,
and Ganssen, G.: Changes in east Atlantic deepwater circulation over the last 30,000
years: eight time slice reconstructions, Paleoceanography, 9(2), 209–267, 1994.
Sarnthein, M., Pflaumann, U., and Weinelt, M.: Past extent of sea ice in the
northern North Atlantic inferred from foraminiferal paleotemperature estimates,
Paleoceanography, 18(2), 2003.
Sarnthein, M., Grootes, P.M., Kennett, J.P., and Nadeau, M.: $^{14}$C Reservoir ages
show deglacial changes in ocean currents and carbon cycle, Geophys. Monograph –
Am. Geophys. Union, 173, 175–196, 2007.
Sarnthein, M., Grootes, P.M., Holbourn, A., Kuhnt, W., and Kühn, H.: Tropical
warming in the Timor Sea led deglacial Antarctic warming and almost coeval
atmospheric $CO_2$ rise by >500 yr, Earth Planetary Science Letters, 302, 337-348, 2011.
Sarnthein, M., Schneider, B., and Grootes, P.M.: Peak glacial 14C ventilation ages
suggest major draw-down of carbon into the abyssal ocean, Climate of the Past, 9 (1),
925–965, 2013.
Sarnthein, M., Balmer, S., Grootes, P.M., and Mudelsee, M.: Planktic and benthic
14C reservoir ages for three ocean basins, calibrated by a suite of 14C plateaus in the
glacial-to-deglacial Suigetsu atmospheric 14C record, Radiocarbon, 57, 129–151, 2015.
Sarnthein, M. and Werner, K.: Early Holocene planktic foraminifers record species-
specific $^{14}$C reservoir ages in Arctic Gateway, Marine Micropaleontology, 135, 45-55.
DOI:10.1016/j.marmicro.2017.07.002, 2018.
Schlolaut, G.: A unique and easy-to-use-tool to deal with incompletely varved
archives 782 – the Varve Interpolation Program 3.0.0, Quaternary Geochronology,
2019 (in press).



Schlolaut, G., Staff, R.A., Marshall, M.H., Brauer, A., Bronk Ramsey, C., Lamb,
H.F., and Nakagawa, T.: Microfacies analysis of the Lake Suigetsu (Japan) sediments
from ~50 to ~10 ka BP and an extended and revised varve based chronology,
Quaternary Science Reviews, 200, 351-366, 2018.
Schroeder, J., Holbourn, A., Küssner, K., and Kuhnt, W.: Hydrological variability in
the southern Makassar Strait during the last glacial termination, Quaternary Science
Reviews, 154, 143-156, 2016.
Sessford, E.G., Jensen, M.F., Tisserand, A.A., Muschitiello, F., Dokken, T.,
Nisancioglu, K.H., and Jansen, E.: Consistent fluctuations on intermediate water
temperature off the coast off Greenland and Norway suring Dansgaard-Oeschger
events, Quaternary Science Reviews, 223, 105887, 1-17, 2019.
Sherriff-Tadano, S., Abe-Ouchi, A., Yoshimori, M., Oka, A., and Chan, W.-L.:
(Influence of glacial ice sheets on the Atlantic meridional overturning circulation through
surface wind change, Climate Dynamics, 50 (7-8), 2881–2903, 2017.
Siani, G., Michel, E., De Pol-Holz, R., DeVries, T., Lamy, F., Carel, M., Isguder, G.,
Dewilde, F., Lourantou, A.: Carbon isotope records reveal precise timing of enhanced
Southern Ocean upwelling during the last deglaciation, Nature Communications, 4,
1134    2758, 2013.
Simstich, J., Sarnthein, M., and Erlenkeuser, H.: Paired $\delta^{18}O$ signals of
*Neogloboquadrina pachyderma* (s) and *Turborotalita quinqueloba* show thermal
stratification structure in Nordic Seas, Mar. Micropaleontol., 912, 1–19, 2003.
Skinner, L.C., Fallon, S., Waelbroeck, C., Michel, E., and Barker, S.: Ventilation of
the deep Southern Ocean and deglacial CO2 rise, Science, 328, 1147–1151, 2010.
Skinner, L.C., Waelbroeck, C., Scrivner, A.E., and Fallon, S.J.:Radiocarbon
evidence for alternating northern and southern sources of ventilation of the deep
Atlantic carbon pool during the last deglaciation, PNAS, 111, 5480–5484, 2014.
Skinner, L.C. *et al.*: Reduced ventilation and enhanced magnitude of the deep Pacific
carbon pool during the last glacial period, Earth and Planetary Science Letters*, **411**, 45-
1145    52, 2015.
Skinner, L.C., Primeau, F., Freeman, E., de la Fuente, M., Goodwin, P.A.,
Gottschalk, J., Huang, E., McCave, I.N., Noble, T.L., and Scrivner A.E.: Radiocarbon
constraints on the glacial ocean circulation and its impact on atmospheric $CO_2$, Nature
communications*, 8:16010, DOI: 10.1038/ncomms16010, 2017.



Southon, J., Noronha, A.L., Cheng, H, Edwards, R.L., and Wang, Y.: A high-
resolution record of atmospheric $^{14}$C based on Hulu Cave speleothem H82, Quaternary
Science Reviews, 33:32-41, 2012.
Steffensen, J.P., Andersen, K.K., Bigler, M., et al.: High-Resolution Greenland Ice
Core Data Show Abrupt Climate Change Happens in Few Years, Science, 321, 680;
DOI: 10.1126/science.1157707, 2008.
Stocker, T. and Johnsen, S.J.: A minimum thermodynamic model for the bipolar
seesaw, Paleoceanography, 18 (4), 1087, doi:10.1029/2003PA000920, 2003.
Stuiver, M. and Braziunas, T.V.: Modeling atmospheric $^{14}$C influences and $^{14}$C ages
of marine samples to 10,000 B.C., Radiocarbon, 35, 137–189, 1993.
Svensson, A., Andersen, K.K., Bigler, M., Clausen, H.B., Dahl-Jensen, D., Davies,
S.M., Johnsen, S.J., Muscheler, R., Parrenin, F., Rasmussen, S.O., Röthlisberger, R.,
Seierstad, I., Steffensen, J.P., and Vinther, B.M.: A 60 000 year Greenland
stratigraphic ice core chronology, Climate of the Past, 4, 47–57, 2008.
Toggweiler, J.R., Druffel, E.R.M., Key, R.M., and Galbraith, E.D.: Upwelling in the
ocean basins north of the ACC. Part 2: How cool Subantarctic water reaches the surface
in the tropics, J. Geophysical Research, DOI:10.1029/2018JC014795, 2019 (in press).
Turney, C.S.M., Fifield, L.K., Hogg, A.G., et al.: Using New Zealand kauri (*Agathis*
*australis*) to test the synchronicity of abrupt climate change during the Last Glacial
Interval (60,000–11,700 years ago), Quatern. Sci. Rev., 29, 3677–3682, 2010.
Turney, C.S.M., Jones, R.T., Phipps, S.J., et al.: Rapid global ocean-atmosphere
response to Southern Ocean freshening during the last glacial, Nature communications,
8:520, doi:10.1038/s41467-017-00577-6, 2017.
Umling, N.E. and Thunnell, R.C.: Synchronous deglacial thermocline and deep-water
ventilation in the eastern equatorial Pacific, Nature communications, 8, 14203. DOI:
10.1038/ncomms14203, 2017.
Waelbroeck, C., Duplessy, J.-C., Michel, E., Labeyrie, L., Paillard, D., and Duprat,
J.: The timing of the last deglaciation in North Atlantic climate records, Nature, 412,
724–727, 2001.
Waelbroeck, C., Skinner, L.C., Labeyrie, L., Duplessy, J.-C., Michel, E., Riveiros,
N.V., Gherardi, J.-M., and Dewilde, F.: The timing of deglacial circulation changes in
the Atlantic, Paleoceanography, 26, PA3213, https://doi.org/10.1029/2010PA002007,

1182   2011.





Wallmann, K., Schneider, B., and Sarnthein, M.: Effects of eustatic sea-level change,
ocean dynamics, and nutrient utilization on atmospheric $pCO_2$ and seawater composition
over the last 130,000 years – a model study, Climate of the Past, 12, 339-375, doi:
10.5194/cp-12-339-2016, 2016.
Wang Y.C, Cheng, H., Edwards, R.L., An, Z.S., Wu, J.Y., Shen, C.-C., and Dorale,
J.A.: A high-resolution absolute-dated Late Pleistocene monsoon record from Hulu
Cave, China, Science, 294, 2345- 2348. DOI: 10.1126/science.1064618, 2001.
Wang, L.J., Sarnthein, M., Erlenkeuser, H., Grimalt, J., Grootes, P., Heilig, S., Ivanova,
E., Kienast, M., Pelejero, C., and Pflaumann, U.: East Asian monsoon climate during the
late Pleistocene: High-resolution sediment records from the South China Sea, Marine
Geology, 156, 245-284, 1999.
Wang, P., Clemens, S., Beaufort, L., Braconnot, P., Ganssen, G., Jian, Z., Kershaw,
P., and Sarnthein, M.: SCOR/IMAGES Working Group 113 SEAMONS: Evolution and
variability of the Asian Monsoon System: State of the art and outstanding issues,
Quaternary Science Reviews, 24 (5-6), 595-629, 2005.
Xu, J., Kuhnt, W., Holbourn, A., Regenberg, M., and Andersen, N.: Indo-Pacific
Warm Pool variability during the Holocene and Last Glacial Maximum, Paleoceanogr.,

1200    25, 16, 2010.

Yamamoto, A., Abe-Ouchi, A., Ohgaito, R., Ito, A., and Oka, A.: Glacial $CO_2$
decrease and deep-water deoxygenation by iron fertilization from glaciogenic dust,
Climate of the Past, 15, 981-996.
Yashayaev, I., Seidov, D., and Demirov, E.: A new collective view of oceanography
of the Arctic and North Atlantic basins, Progress in Oceanography, 132, 21 pp.,
DOI:http://dx.doi.org/10.1016/j.pocean.2014.12.012, 2015.
Zhao, N. and Keigwin, L.D.: An atmospheric chronology for the glacial-deglacial
Eastern Equatorial Pacific, Nature communications, 9:3077, DOI:10.1038/s41467-018-
05574-x, 2018.
Zhao, N., Marchal, O., Keigwin, L., Amrhein, D., and Gebbie, G.: A synthesis of
deep-sea radiocarbon records and their (in) consistency with modern ocean ventilation,
Paleoceanography and Paleoclimatology, 33, 128-151, 2018.





TABLE CAPTIONS

⊻ **Table 1 a and b.** Summary of varve- and U/Th model-based age estimates (Schlolaut
et al., 2018; Bronk Ramsey et al., 2012) for ~30 plateau (pl.) boundaries in the
atmospheric $^{14}$C record identified in Lake Suigetsu Core SG06$_{2012}$ by means of visual
inspection over the interval 10.5–27 cal. ka (Sarnthein et al., 2015, suppl. and modified).
At the right hand side, three columns give the average (Ø) and uncertainty range of $^{14}$C
ages for each $^{14}$C plateau.

| SUIGETSU SG06_2012 Plateau no. | Plateau Top Varve-based age (yr BP) | U/Th-based age (yr BP) | Depth (cm c.d.) | Plateau Base Varve-based age (yr BP) | U/Th-based age (yr BP) | Depth (cm c.d.) | Ø 14C Age of 14C Plateau (14C yr) | ±Uncertainty (14C yr) | 14C age BP min/max. (1.6 σ range) |
|---|---|---|---|---|---|---|---|---|---|
| 'Preboreal' | 10525 | 10560 | 1325 | 11100 | 11108 | 1383 | 9525 | −170/+110 | 9356/9635 |
| 'Top YD' | 11290 | 11281 | 1402 | 11760 | 11755 | 1453 | 10060 | −100/+35 | 9963/10095 |
| 'YD' | 11950 | 11895 | 1467 | 12490 | 12475 | 1525 | 10380 | −170/ 124 | 10211 10504 |
| 1a | 13580 | 13656 | 1626 | 13980 | 14042 | 1657 | 12006 | 100 | 11857 12050 |
| 1 | 14095 | 14160 | 1666 | 15095 | 15100 | 1740 | 12471 | 185 | 12315 12683 |
| 2a | 15310 | 15420 | 1754 | 16140 | 16520 | 1802 | 13406 | 245 | 13174 13665 |
| 2b | 16075 | 16520 | 1802 | 16400 | 16930 | 1820 | 13850 | 40 | 13808 13885 |
| 3 | 16835 | 17500 | 1847 | 17500 | 18220 | 1888 | 14671 | 105 | 14582 14792 |
| 4 | 17880 | 18650 | 1913 | 18830 | 19590 | 1971 | 15851 | 190 | 15661 16044 |
| 5a | 18960 | 19720 | 1978 | 19305 | 20240 | 2003 | 16670 | 90 | 16570 |




| | | | | | | | | | |
|---|---|---|---|---|---|---|---|---|---|
| | | | | | | | | | 16750 |
| 5b | 19305 | *20240* | 2003 | 20000 | *20900* | 2032 | 17007 | 190 | 16830 |
| | | | | | | | | | 17247 |
| 6a | 20190 | *21000* | 2050 | 20920 | *21890* | 2105 | 17667 | 262 | 17435 |
| | | | | | | | | | 17960 |
| 6b | 20920 | *21890* | 2105 | 21275 | *22300* | 2132 | 18075 | 140 | 17960 |
| | | | | | | | | | 18240 |
| 7 | 21375 | *22400* | 2140 | 21790 | *22870* | 2171 | 18843 | 117 | 18741 |
| | | | | | | | | | 18975 |
| 8 | 21835 | *22940* | 2175 | 22730 | *24250* | 2257 | 19715 | −290 | 19425 |
| | | | | | | | | 325 | 20041 |
| 9 | 22730 | *24250* | 2257 | 23395 | *25150* | 2312 | 20465 | −227 | 20238 |
| | | | | | | | | 263 | 20728 |
| 10a | 23935 | *25880* | 2358 | 25080 | *27000* | 2400 | 22328 | −380 | 21946 |
| | | | | | | | | 270 | 22600 |
| 10b | 25080 | *27000* | 2400 | 25800 | *27600* | 2426 | 22708 | −475 | 22233 |
| | | | | | | | | 440 | 23147 |
| 11 | 26110 | *27770* | 2443 | 27265 | *28730* | 2525 | 24088 | −360 | 23727 |
| | | | | | | | | 505 | 24595 |
















Ⱶ **Table 2**. Temporal match of [14]C plateaus and deglacial periods of major degassing of
the ocean  (AA = Antarctic).

DEGLACIAL  EVENTS  of  pCO$_2$  RISE  vs. age of pla. [14]C PLATEAUS  (in cal. ka)

| pCO$_2$ RISE (~12 ppm) | Plateau no. | Plateau boundaries | |
|---|---|---|---|
| AGE based on annual layers AA ice core | | AGE (cal. ka) based on varve counts | **U/Th model ages** |
| (Marcott et al. 2014) | | (Schlolaut et al. 2018; Bronk Ramsey et al., 2012) | |
| 11.7 – 11.5 | # 'Top YD' | 11.76 – 11.32 | **11.83 – 11.3** |
| 14.8 – 14.53 | # 1 | 15.1 – 14.1 | **15.1 – 14.2** |
| 16.4 – 16.15 | # 2a | 16.14 – 15.3 | **16.52 – 15.5** |
| 17.4 – ~17.1 | # 3 | 17.5 – 16.83 | **18.22  – 17.5** |

*FURTHER POTENTIAL CORRELATIVES*:

| | | | |
|---|---|---|---|
| Progressive N. Atlantic warming during the YD at 12.39 – 12.03 ka * | # 'YD' | 12.49 – 11.87 | **12.46 – 11.98** |
| Onset of Antarctic  ** warming at 18.3–17.6 ka (ice-based time scale) | #3 | 17.5 – 16.85 | **18.22 – 17.5** |
| Onset of North Atlantic  *** warming at 19.3–18.6 ka (U/Th-based time scale) | # 4 | 18.83 – 17.9 | **19.6 – 18.65** |
| Top H2: GIS 2 N. Atlantic warming at 23.4 – 23.3 ka **** | #9 / #8 | 23.4 – 22.73 | **24.25 – 22.95** |

AGE  CONTROL based on

* Naughton et al. (2019)

** Kawamura et al. (2007)

*** Balmer and Sarnthein (2018)

**** Grootes and Stuiver (1997)




⩗ **Table 3** a-c. $^{14}$C reservoir / ventilation ages of surface (top 50-100 m) and bottom
waters vs. U/Th-based model age at 19/22 core sites in the ocean – (a) Spatial and
temporal changes over the LGM (22–20 and 20–18 cal. ka), (b) HS-1, and the B/A.
LGM estimates are compared to model-based estimates of Muglia et al. (2018). (c) Data
sources. For core locations see Fig. S2.
(a)

| Sediment Core U/Th-based model age Plateau (Pl.) no. | Latitude | Longitude | Water depth (m) | LGM pla. res age 24–21 ka Pl. 8 - 7 - 6 | Error (yr) | 21–18.7 ka Pl. 5 - 4/3 | Error (yr) | LGM model res. age strong AMOC (yr) | weak (yr) |
|---|---|---|---|---|---|---|---|---|---|
| **ATLANTIC O.** | | | | | | | | | |
| PS2644 | 67°52.02'N | 21°45.92'W | 777 | 2100 | ±390 | 1920–2200 | ±325 –±125 | 1136 | 1100 |
| GIK 23074 | 66°66.67'N | 4°90'E | 1157 | 620–790 | ±145–±270 | 550–1175 | ±100–±200 | 1054 | 1059 |
| MD08-3180 | 38°N | 31°13.45'W | 3064 | – | | 320–605 | ±125–±405 | 827 | 887 |
| SHAK06-5K | 37°34'N | 10°09'W | 2646 | 700–930 | | 330–650 | | 872 | 855 |
| (= MD99-2334) | (37°48'N | 10°10'W | 3146 | | | | | | |
| ODP 1002 | 10°42.37'N | 65°10.18'W | 893 | 700–210 | ±230–±310 | 25 – -205 | ±205–±215 | 751 | 738 |
| GeoB 3910-1 | 4°15'S | 36°21'W | 2361 | – | | – | | 779 | 796 |
| GeoB 1711-4 | 23°17'S | 12°23'W | 1976 | 1080 | ±290 | 730–840 | ±240–±190 | 711 | 721 |
| KNR 159-5-36GG | 27°31'S | 46°48'W | 1268 | 540 | ±140 | 870 | ±120 | 757 | 777 |
| MD07-3076 | 44°4'S | 4°12'W | 3770 | – | | 2300 | ±200 | 928 | 989 |
| | | | | | | | | | |
| **INDIAN O./TIMOR SEA** | | | | | | | | | |
| MD01-2378 | 13°08.25'S | 121°78.8'E | 1783 | – | | 2000–1700 | ±300–±320 | 885 | 890 |
| **PACIFIC O.** | | | | | | | | | |
| MD02-2489 | 54°39.07'N | 148°92.13'V | 3640 | – | | 1560–1110 | ±310–±335 | 972 | 965 |
| MD01-2416 | 51°26.8'N | 167°72.5'E | 2317 | – | | 1710 | ±440 | 1227 | 1202 |
| ODP 893A | 34°17.25'N | 120°02.33'V | 588 | – | | 1065 | ±280 | 839 | 846 |
| MD02-2503 | 34°16.6'N | 120°01.6'W | 580 | – | | – | | 839 | 846 |
| GIK 17940 | 20°07.0'N | 117°23.0'E | 1727 | 1820–1260 | ±320–±230 | hiatus | | 836 | 838 |
| (= SO50-37) | 18°55'N | 115°55'E | 2655 | 1820–1260 | | | | 836 | 840 |
| PS75/104-1 | 44°46'S | 174°31'E, | 835 | 1650–1280 | | 1500 | | 881 | 895 |
| (= SO213-84) | 45°7.5'S | 174°34,9'E | 972 | 1650–1280 | | 1500 | | 881 | 895 |
| MD07-3088 | 46°S | 75°W | 1536 | 380 | | 200-350 | | 917 | – |
| SO213-76-2 | 46°13'S | 178°1.7'W | 4339 | – | | 1600–1560 | | 915 | 842 |
| PS97/137-1 | 52°39.5'S | 75°33.9'E | 1027 | 2290–2110 | | 2400–1800 | | 1505 | 1419 |













Climate of the Past Discussions — Open Access — EGU

(b)

| Sediment Core U/Th-based model Plateau (Pl.) no. | HS-1 pla. res. age 18 –16.5 ka Pl. 3 - 2b (yr) | Error (yr) | 16.5–15.5 ka Pl. 2a (yr) | Error (yr) | B/A pla. res. age 14.7 –13.6 ka Pl. 1 - 1a | Error (yr) | LGM be. vent age (yr) early | late | LGM b.w. model age strong AMOC (yr) | weak (yr) |
|---|---|---|---|---|---|---|---|---|---|---|
| **ATLANTIC O.** | | | | | | | | | | |
| PS2644 | 1775–1660 | ±105–±160 | 1900 | ±355 | – | | 345 | 2400 | 948 | 918 |
| GIK 23074 | 1730–2000 | ±125–±160 | 670 | ±310 | 140–310 | ±250–±100 | 375 | 375 | 960 | 931 |
| MD08-3180 | 1420–1610 | ±310–±160 | 1460 | ±390 | 630–360 | ±310 | 600 | 600 | 1031 | 1004 |
| SHAK06-5K | 350–420 | | 550 | | 800–1200 | | — | — | — | — |
| (= MD99-2334) | | | | | | | 2200–2700 | 1900 | — | — |
| ODP 1002 | –100 – 20 | ±140 | 90 | ±345 | 355 | ±200 | — | | 1247 | 1175 |
| GeoB 3910-1 | 630–560 | ±160–±180 | 175 | ±475 | 210–230 | ±220–±110 | 2150 | 2150 | — | — |
| GeoB 1711-4 | 660–690 | ±195–±45 | 420 | ±320 | 880 | ±255 | 1500 | 1500 | 1387 | 1714 |
| KNR 159-5-36GGC | 460–340 | ±380–±300 | 170 | ±700 | 180–230 | ±370–±310 | 1470 | 1470 | 1354 | 1563 |
| MD07-3076 | 1650 | ±180 | – | | 920 | ±230 | 3640 | 3640 | 1653 | 2060 |
| **INDIAN O./TIMOR SEA** | | | | | | | | | | |
| MD01-2378 | 740 | ±125 | – | | 200–185 | ±345–±135 | 2720 | — | 1679 | 1881 |
| **PACIFIC O.** | | | | | | | | | | |
| MD02-2489 | 800–550 | ±155–±120 | 550 | ±305 | 440 | ±285 | | 2625 | 2332 | 2595 |
| MD01-2416 | 1480–1140 | ±135–±195 | – | | 720–570 | ±285–±140 | | 3700/510 | 2400 | 2683 |
| ODP 893A | 1065–1490 | ±280–±125 | 1400 | ±370 | 520 | ±185 | | 1430 | 1677 | 1705 |
| MD02-2503 | 965–1365 | ±160–±165 | 1215 | ±325 | 395–535 | ±240–±130 | — | — | — | — |
| GIK 17940 | 1210–1370 | ±200–±470 | 1045 | ±320 | 870–970 | 325–±100 | 3300–1800 | | 1807 | 1897 |
| (= SO50-37) | | | | | | | 3225 | 3225 | 2373 | 2667 |
| PS75/104-1 | 1050 | | 1100 | | 800–250 | | — | — | — | — |
| (= SO213-84) | | | | | | | 1500 | 2400 | 1101 | 1146 |
| MD07-3088 | 800–1090 | | 1010 | | 730–940 | | 1600 | 1600 | 1808 | 1701 |
| SO213-76-2 | 200 | | – | | – | | 4685 | 4685 | 1712 | 2001 |
| PS97/137-1 | 1500–670 | | 435 | | – | | 3300 | 2100 | 1631 | 1871 |


(c)

| Sediment Core | DATA Source |
|---|---|
| **ATLANTIC O.** | |
| PS2644 | Sarnthein et al. 2015 Be.data suppl. |
| GIK 23074 | Sarnthein et al. 2015 |
| MD08-3180 | Balmer et al. 2018 |
| SHAK06-5K | Ausin et al., 2019 |
| (= MD99-2334) | Skinner et al. 2014 |
| ODP 1002 | Sarnthein et al. 2015 |
| GeoB 3910-1 | Balmer et al. 2016 |
| GeoB 1711-4 | Balmer et al. 2016 |
| KNR 159-5-36GGC | Balmer et al. 2016   data suppl. |
| MD07-3076 | Balmer et al. 2016 |
| **INDIAN O./TIMOR SEA** | |
| MD01-2378 | Sarnthein et al. 2015 |
| **PACIFIC O.** | |
| MD02-2489 | Sarnthein et al. 2015 |
| MD01-2416 | Sarnthein et al. 2015   modified |
| ODP 893A | Sarnthein et al. 2015   data suppl. |
| MD02-2503 | Sarnthein et al. 2015 |
| GIK 17940 | Sarnthein et al. 2015 |
| (= SO50-37) | Sarnthein et al. 2015 |
| PS75/104-1 | Küssner et al., 2018 |
| (= SO213-84) | Ronge et al., 2016 |
| MD07-3088 | Küssner et al., 2019 |
| SO213-76-2 | Küssner et al., 2019 |
| PS97/137-1 | Küssner et al., 2019 |





FIGURE  CAPTIONS

– Fig. 1. Atmospheric $^{14}$C ages of Lake Suigetsu plant macrofossils 10–29 cal. ka vs.
U/Th-based model age (blue dots; Bronk Ramsey et al., 2012). Double and triple $^{14}$C
measurements are averaged. (In part large) error bars of single $^{14}$C ages are given in
Suppl. Fig. S1. Suite of labeled horizontal boxes that envelop scatter bands of largely
constant $^{14}$C ages shows $^{14}$C plateaus longer than 250 yr (plateau boundary ages listed
in Table 1). Red and brown dots (core samples from trench and wall) and + signs (off-
axis core samples) depict raw $^{14}$C ages of Hulu stalagmite core H82 (Cheng et al.,
2018; Southon et al., 2012; plot offset by +3000 $^{14}$C yr). Suite of short $^{14}$C plateaus
(black boxes) tentatively assigned to Hulu-based record occupy age ranges slightly
different from those deduced for Suigetsu-based plateaus. The difference possibly
results from short-term changes in the Old / Dead Carbon Fraction (ocf / dcf) that in turn
may reflect major short-term changes in LGM and deglacial monsoon climate (Wang et
al., 2001; Kong et al., 2005).



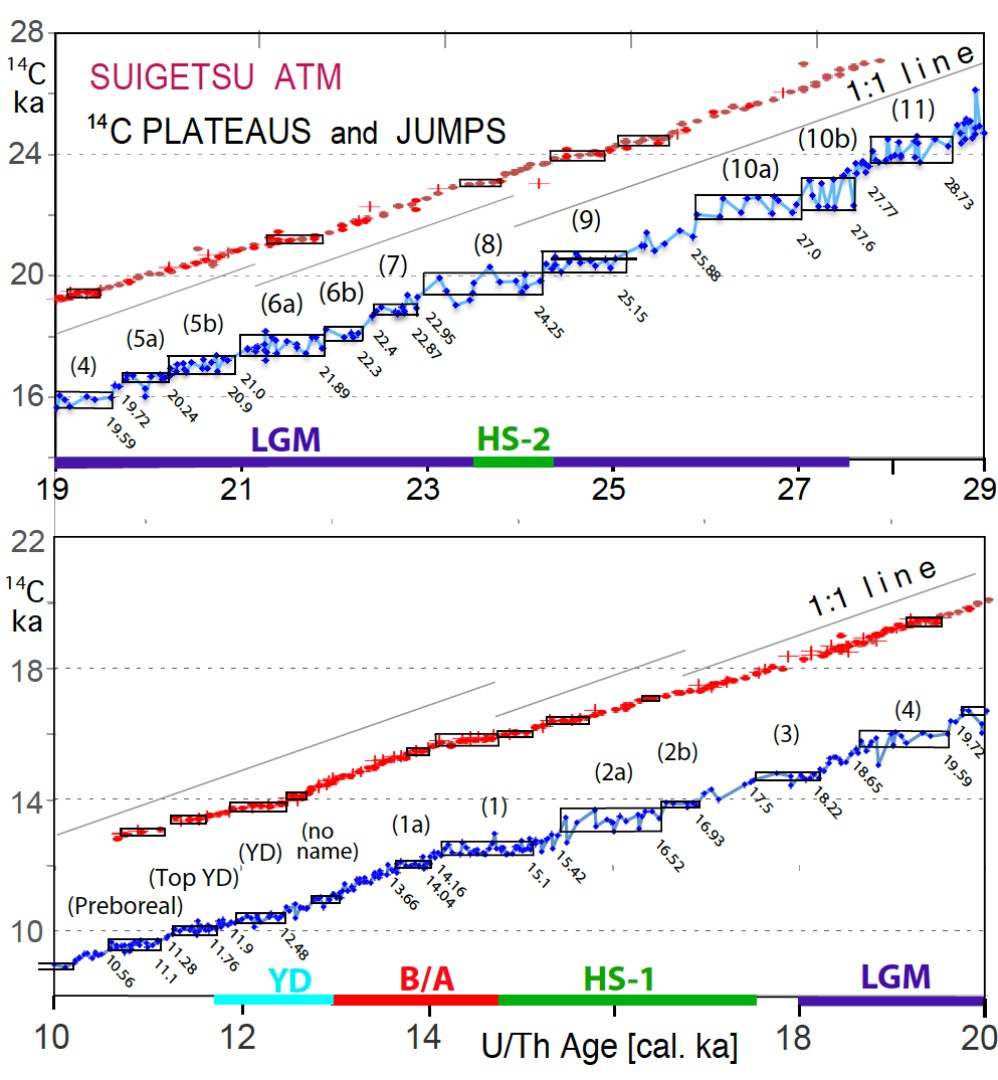





↯ Fig. 2. High-resolution record of atmospheric ¹⁴C jumps and plateaus (= suite of
labeled horizontal boxes that envelop scatter bands of largely constant ¹⁴C ages
extending over >300 cal. yr) in a sediment section of Lake Suigetsu vs. tree ring-based
¹⁴C jumps and plateaus 10–14.5 cal. ka (Reimer et al., 2013). Blue line averages paired
double and triple ¹⁴C ages of Suigetsu plant macrofossils. Age control points (cal. ka)
follow varve counts (Schlolaut et al., 2018) and U/Th model-based ages of Bronk
Ramsey et al. (2012). YD = Younger Dryas, B/A = Bølling-Allerød.

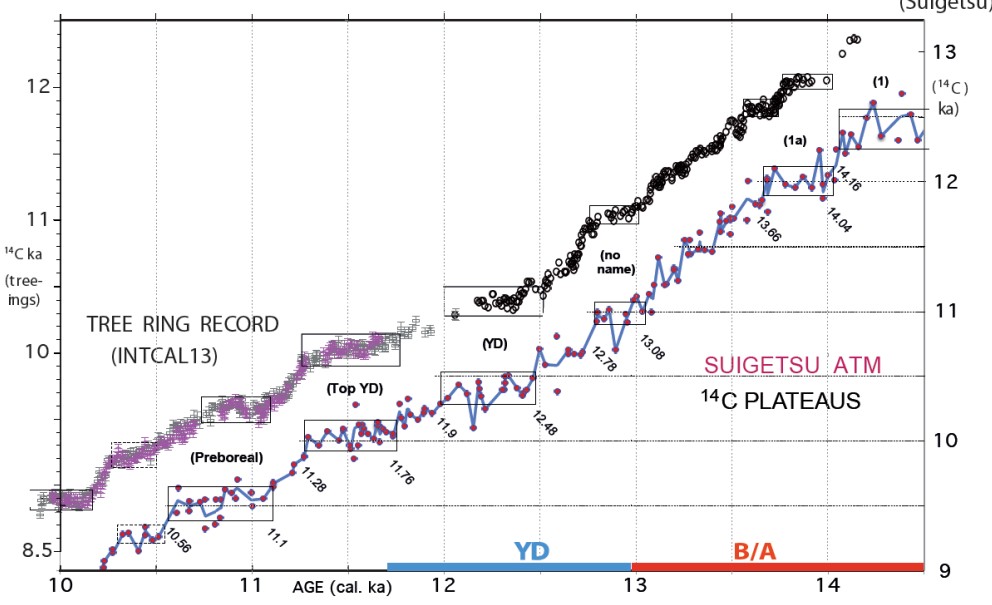






↯ Fig. 3. Difference between Hulu Cave U/Th-based model ages (Southon et al., 2012;
Bronk Ramsey et al., 2012; Cheng et al., 2018) and varve count-based cal. ages for
atmospheric $^{14}$C plateau boundaries in Lake Suigetsu sediment record (Schlolaut et al.,
2018) (Sarnthein et al., 2015, suppl. and revised), displayed on the U/Th-based time
scale 13–27 cal. ka.

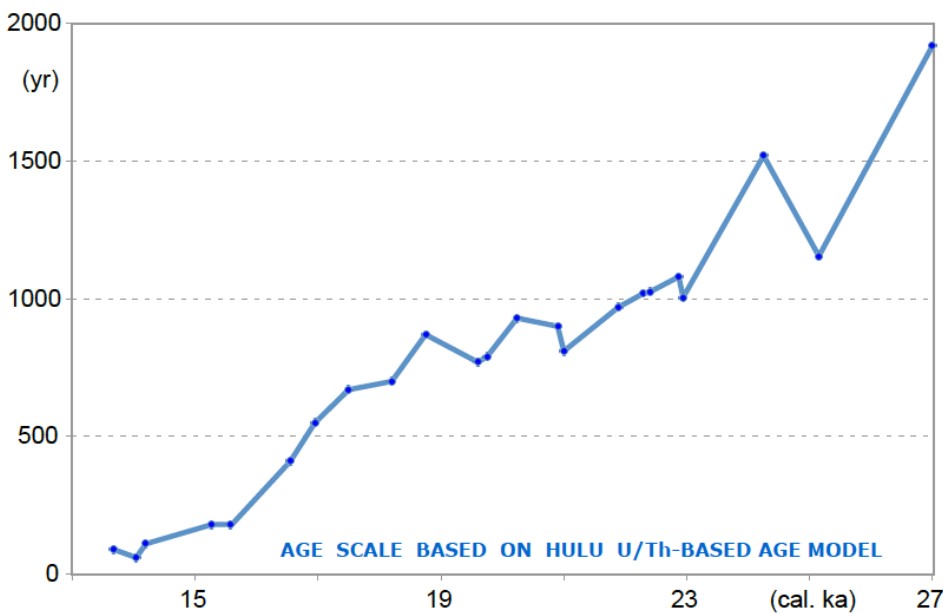












↲ Fig. 4 a and b. Atmospheric $^{14}$C ages and plateaus (horizontal boxes) deduced from
$^{10}$Be production rates vs. GICC05 age scale (Adolphi et al., 2018) compared to the
Suigetsu record of atmospheric $^{14}$C plateaus vs. Hulu U/Th-based model ages (Southon et
al., 2012; Cheng et al., 2018).

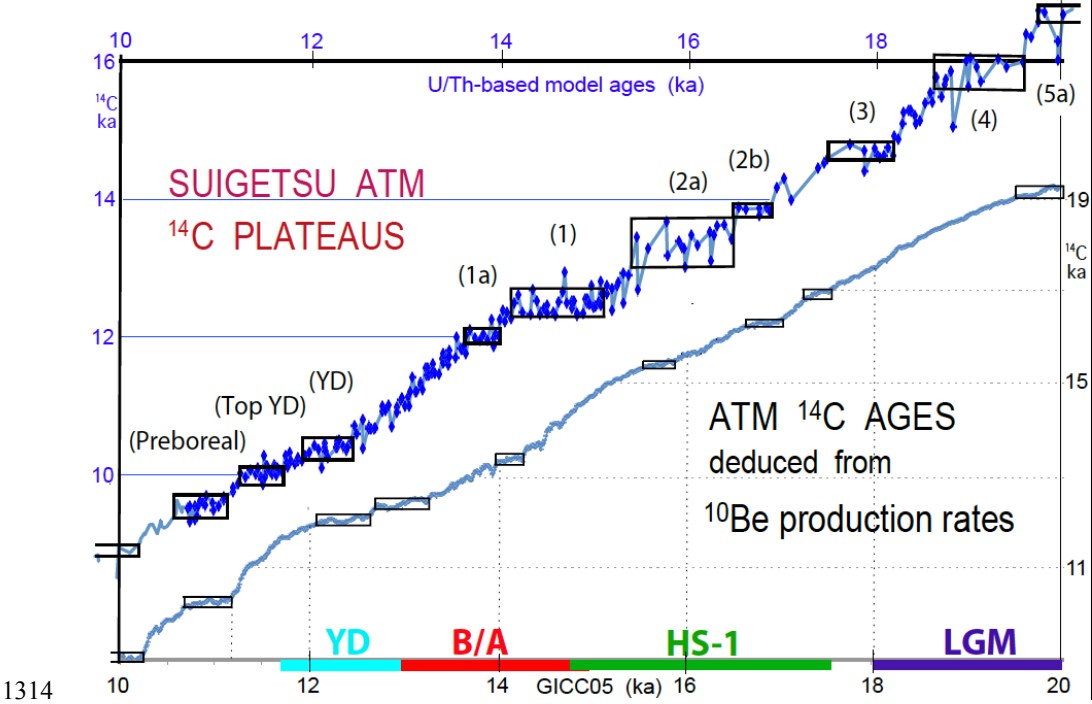



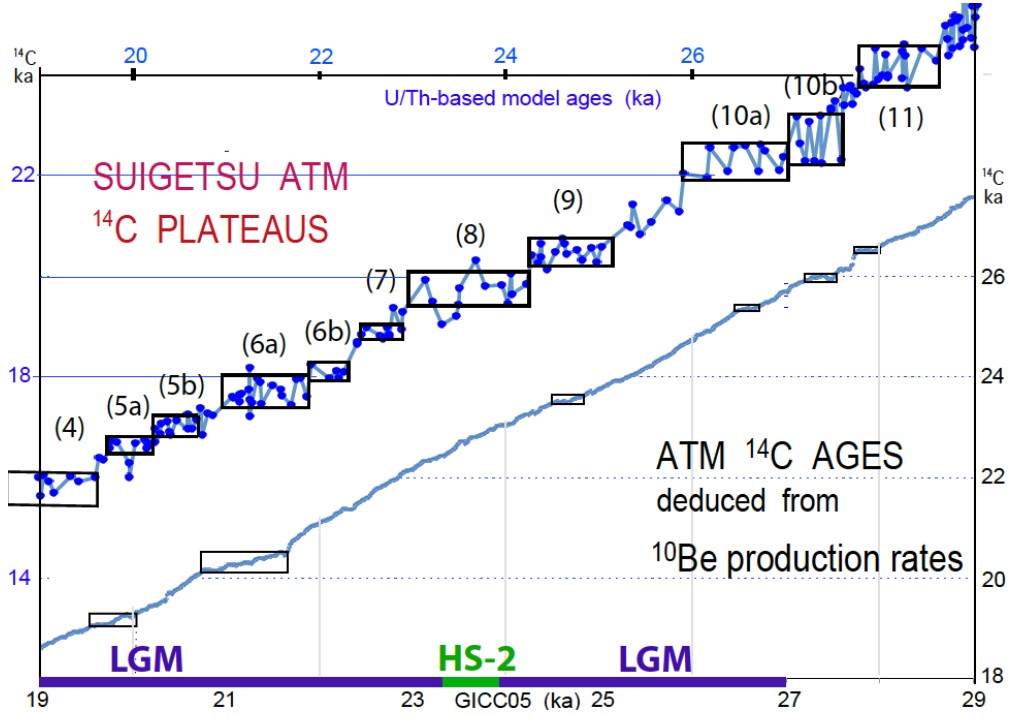
















↓ Fig. 5. Sediment facies and microfacies zones in Lake Suigetsu Core SG06, ~13–32 m
depth (simplified and suppl. from Schlolaut et al., 2018). Microscopy-based frequency of
siderite layers with quality 1–3 in 20-cm sediment sections (= running average of layer
counts for 5 cm thick sections each), a measure of seasonal lamination quality with
gradual transitions between varved and poorly varved sediment sections. Rounded varve
ages are microscopy based and constrain age of major facies and microfacies
boundaries. ANI I to III mark core sections with ultrafine lamination due to sedimentation
rate minima, AT marks tephra layer named AT, 'Event layers' label major thin mud slides
probably earth quake-induced.

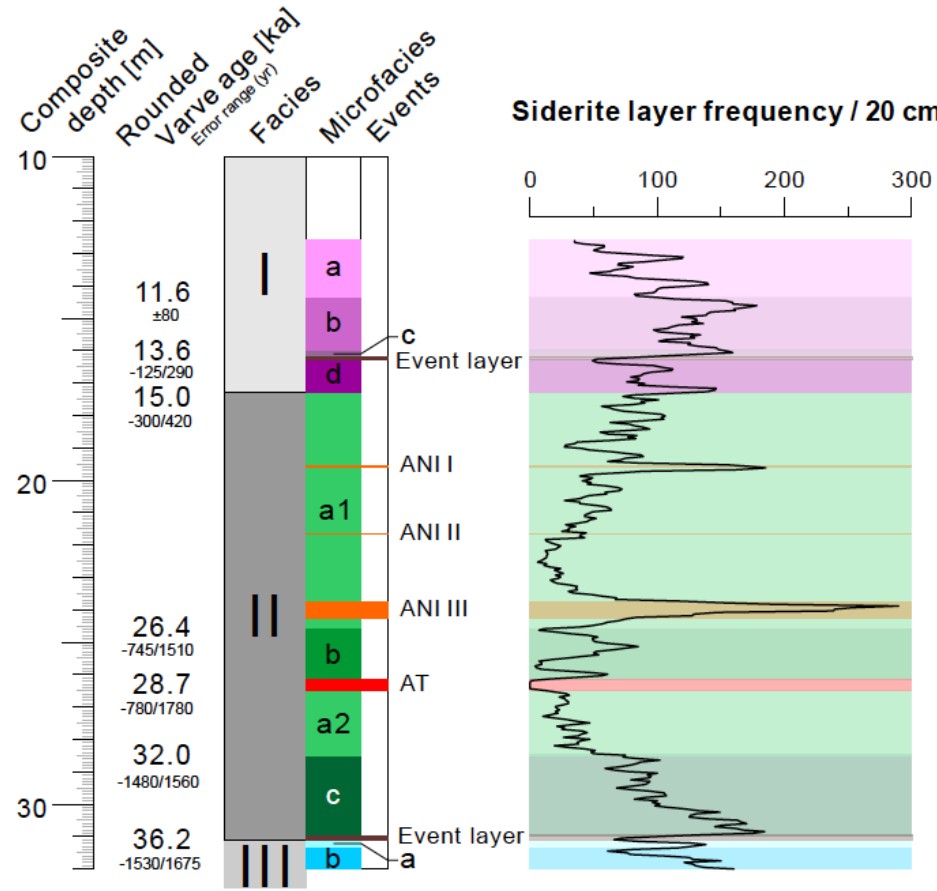






↓ Fig. 6 (a). Four sudden steps (pink bars) in the deglacial atmospheric $CO_2$ rise at West
Antarctic Ice Sheet Divide ice core (WDC) reflect events of fast ocean degassing, that may
have contributed to the origin of deglacial $^{14}C$ plateaus. Age control based on ice cores
(Marcott et al., 2014). (b) The steps are compared to suite of atmospheric $^{14}C$ plateaus dated
by Hulu U/Th-based model ages (Bronk Ramsey et al., 2012). Hol = Holocene; YD = Younger
Dryas; B/A = Bølling-Allerød; HS = Heinrich stadial; LGM = Last Glacial Maximum.

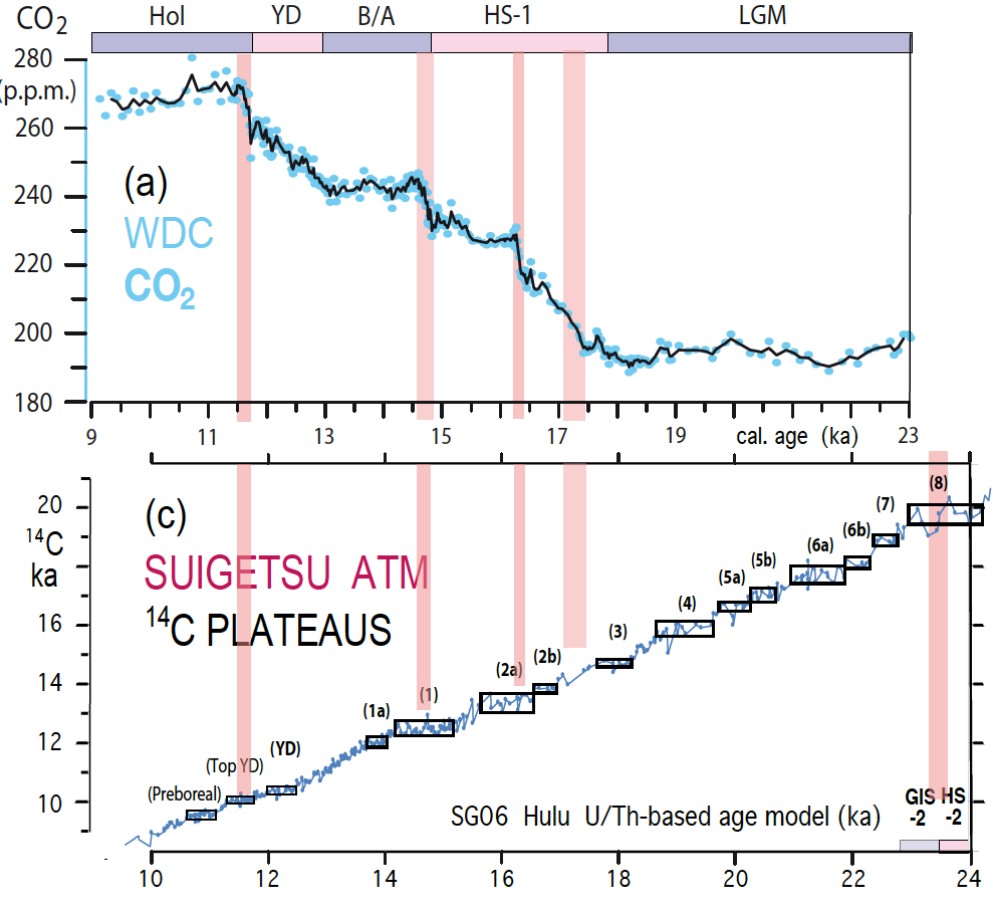







⊻ Fig. 7. Global distribution of $^{14}$C reservoir ages of Late LGM surface waters estimated
by means of planktic $^{14}$C plateau tuning and model-based estimates (general circulation
model of Muglia et al., 2018), assuming an AMOC strength of 13 Sv). X-Y graph (a) and
map (b) show (rounded) differences and intra-LGM trends with minor differences
displayed in magenta, larger differences of >400 yr in red. Planktic habitat depths and
model estimates are largely confined to 0–100 m water depth. Regional distribution
patterns of LGM surface, intermediate, and deep-water ages are given in Table 3 and
Suppl. Fig. S2.

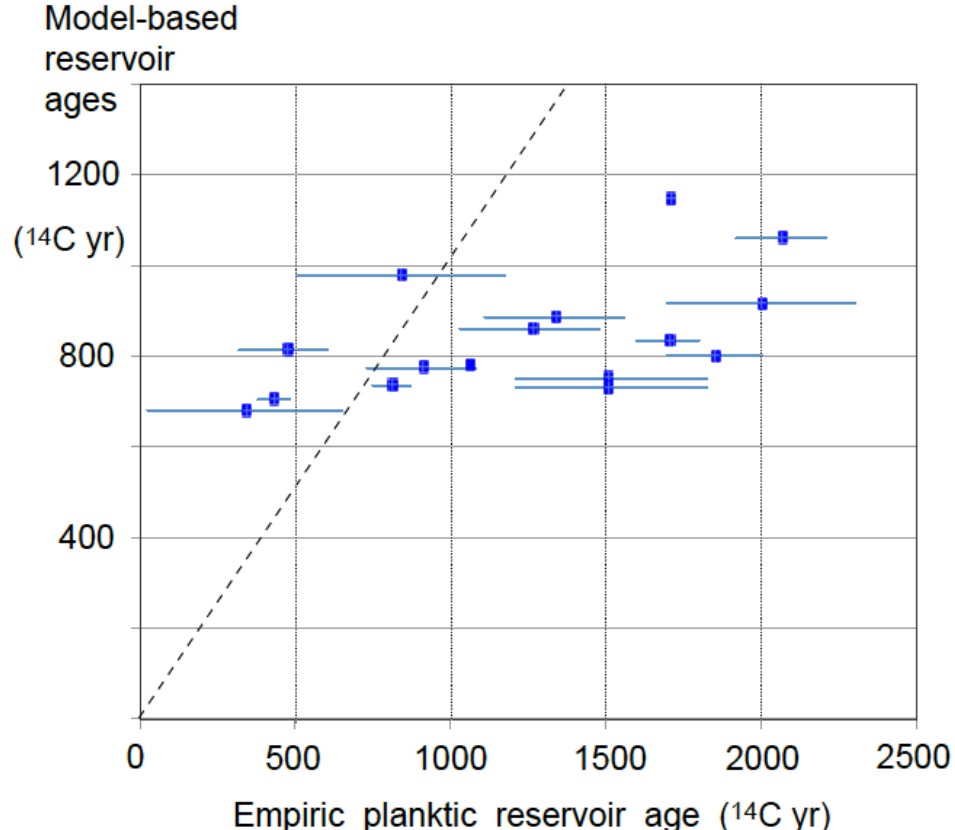






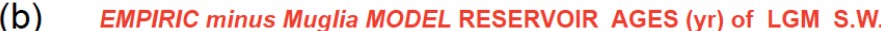

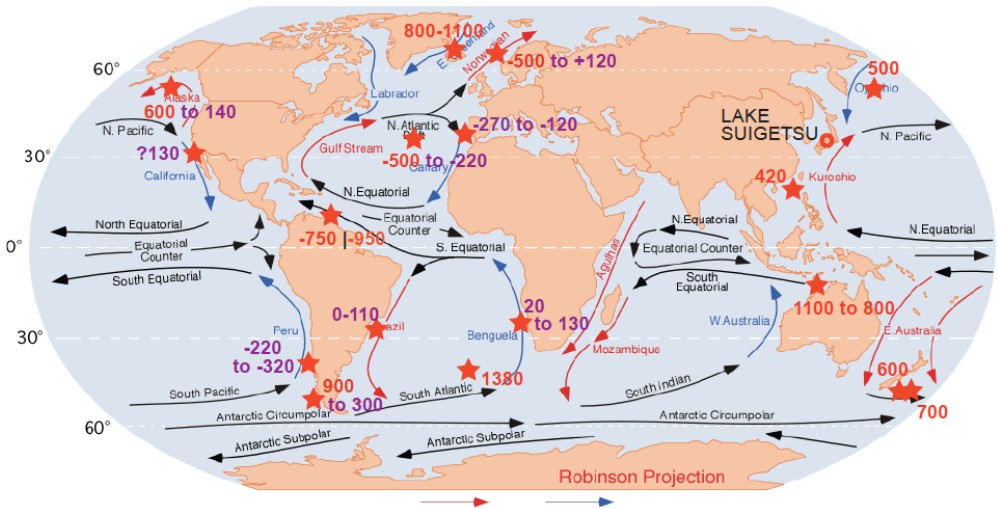




↯ Fig. 8. SW–NE transect of [14]C reservoir and changes in ventilation ages across sites
GIK17940 and SO50-37 in the South China Sea during late LGM ([14]C Plateaus 5 an 4;
upper panel) and HS-1 (lower panel). Insert map shows location of transect. Core
locations are given in Fig. S2. An extreme epibenthic $\delta^{13}C$ minimum in far southwest
(Core GIK17964; Sarnthein et al., 1999) reflects an LGM incursion of Lower/Upper
Pacific Deep Waters (L./ U. PAC DW) with extremely high [14]C ventilation age and DIC
enrichment in contrast to a low ventilation age of North Pacific Deep Water (N. PAC
DW). Arrows reflect direction of potential deep and intermediate-water currents.













⩝ Fig. 9. North Atlantic and North Pacific changes in benthic [14]C ventilation records reflect
seesaw-style reversals in global MOC at the onset and end of early HS-1 (first proposed by
Broecker et al., 1985, however, for LGM times). Arrows within numbers show temporal trends
(a) Late LGM ocean transect reveals global MOC geometry largely similar to today. Blue and
yellow arrows suggest average deep and intermediate-water currents that follow the gradient
from low to high ventilation ages based on paired planktic [14]C reservoir ages derived by
means of the [14]C plateau tuning technique (Sarnthein et al., 2013, Balmer et al., 2018,
Küssner et al., 2019). Note major east-west gradient between LGM eastern and central
Atlantic (off Portugal (PORT) vs. Mid-Atlantic Ridge W of Azores (MAR)). At some Pacific
sites age control is based on paired [14]C ages of planktic foraminifera and wood chunks
(marked by green 'w'; Sarnthein et al., 2015; Zhao and Keigwin, 2018, Rafter et al., 2018).
Zigzag lines mark location of major frontal systems separating counter rotating ocean currents
(e.g., W of Portugal and N of MD07-307: sensu Skinner et al., 2014). (b) HS-1 transect
reveals a short-lasting Atlantic-style overturning in the subpolar North Pacific and a coeval
Pacific-style stratification in the northern North Atlantic. Increased ventilation ages reflect an
enhanced uptake of dissolved carbon in the LGM deep ocean (Sarnthein et al., 2013),
sudden major drops suggest major degassing of $CO_2$ both from the deep Southern Ocean
and North Pacific during early HS-1. SCS = South China Sea. – AABW = Antarctic Bottom
Water; AAIW = Antarctic Intermediate Water. NADW = North Atlantic Deep Water. Blue
arrows = high ventilation, yellow = poor ventilation, red arrows mark poleward warm surface
water currents. Note many arrows are speculative using circumstantial evidence of benthic
$\delta^{13}C$ records and likely local Coriolis forcing at high-latitude sites per analogy to modern
scenarios. Location and names of sediment cores are given in Suppl. Fig. S2, short-term
variations in planktic and benthic [14]C reservoir/ ventilation age in Suppl. Fig. S4 and Table 3.





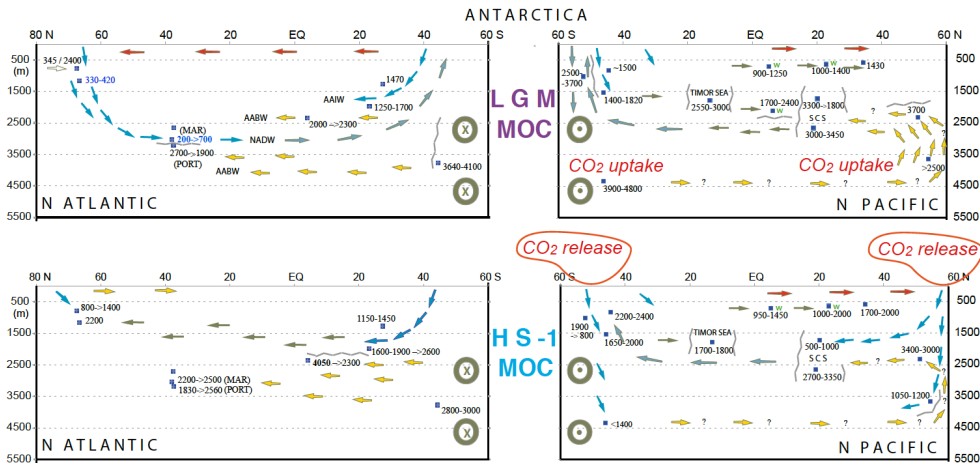