# Peer review of "Plateaus and jumps in the atmospheric radiocarbon record – Potential origin and value as global age markers for glacial-to-deglacial paleoceanography, a synthesis 3 4 Michael Sarnthein1), Kevin Küssner2), Pieter M. Grootes3), Blanca Ausin4)8), Timoth"

_Climate of the Past, 2019_

## Referee Comment (RC1) · Anonymous Referee #1 · 24 Jan 2020

Sarnthein et al. present a revision to the plateau tuning method of deep ocean radiocarbon reservoir age estimation that re-evaluates the age model of the reference Suigetsu 14C record. In addition, the authors present a synthesis of the published research utilizing the plateau tuning method. Because this paper is both a synthesis paper and a presentation of new research, I found it a bit difficult to review. The authors have gone to great lengths to try to organize the paper in a way that helps to wrangle the many topics presented, but I think this paper would benefit from being split into two papers. The first of which could focus on the switch to the U/Th timescale and

the methods and implications that go along with this and the second paper could be a true synthesis paper that summarizes the state of plateau tuning along with its pros and cons. Perhaps these papers could even be presented as companion papers and released as a part 1 and part 2.

Overall, the paper could be simplified to enhance accessibility by deleting some of the unnecessary words (hence, thus, moreover etc.). The authors do a great job of laying out the state of the field and the problems in methodology that exist among current techniques of age model development and the ways plateau tuning can provide a solution to some of these problems. The introduction does a nice job explaining oceanic radiocarbon and the crux of the reservoir age issue. I think it is very important that they have used the introduction to address some of the criticism that has been raised with the plateau tuning method. However, I think this is an area that could be expanded a bit more to address a few more points such as other explanations for marine 14C plateaus, including local shifts in air-sea gas exchange or upwelling, and if so, how can we account for these.

The shift to the U/Th modeled Suigetsu chronology helps address some of the problems associated with the Suigetsu varve counted age model, but it would be helpful to address why using the Suigetsu record for plateau tuning is preferred to Intcal. It seems that correlation directly to Intcal may be more conservative.

If the paper is split into two separate papers, figures S2 and S3 should be included in the main text of the synthesis paper. I found these figures particularly helpful for visualizing the geographic spacing of Plateau tuned records and their findings. More detail included for sections 3.2-3.3 would also be welcome along with a picture/diagram of the Zoophycus burrow.

Specific notes: Figure 1: It isn't clear in the figure caption what the difference is between the top and bottom panels.

Figure 7: Please include the references for the records used in this figure. It would also

be helpful if I could match the data points in the x-y plot to the data points on the map, perhaps using symbols or colors? Also, I'm not sure inclusion of the surface currents in panel b or in figure S3 are helpful, they make it a bit more difficult to see where the cores are located and to read the reservoir age differences- especially because the currents are available in figure S2.

Lines 601-603: Include the unit the corresponds to the foraminifera habitat depths.

Section 3.4, Lines 656-663: Might be good to mention the interspecies 14C differences from Lindsay et al., 2015

Sections 3.5.1 and 3.5.2: In a few spots it would be helpful if the results from plateau tuning studies were more clearly emphasized. This would nicely highlight the important role that this technique has played in our understanding of LGM and HS1 MOC. This is done very nicely in section 3.5.3.

Figure 9: This figure is a bit small and hard to see. The overlapping arrows can also be a bit confusing. Overall, I find that this figure is critical for visualizing the findings from sections 3.51-3.5.2, it might be helpful to also include the modern MOC for comparison.

---

## Short Comment (SC1) · 31 Jan 2020

Dear Editor,

Please note that we have written an extended comment to the paper by Sarnthein et al. CPD (doi: 10.5194/cp-2019-127) which is still in the phase of open discussion.

In addition to commenting on a CPD paper, we believe that our paper includes substantial material of broad interest to the community using radiocarbon in marine sediments for geochronology and paleoceanography.

[Figure]

As recommended on the CPD web site, we have submitted our extended comment as research paper entitled "On the tuning of plateaus in atmospheric and oceanic 14C records to derive deep-sea core chronologies and marine reservoir age changes" by Edouard Bard and Timothy Heaton.

We thank you very much for your attention,

Edouard Bard & Tim Heaton

---

## Referee Comment (RC2) · Anonymous Referee #2 · 10 Feb 2020

Sarnthein et al. provide a review of the plateau tuning technique that hinges on an alignment of atmospheric and oceanic radiocarbon records. The method has not been extensively used outside the first author's group, which might be associated with some sceptisism towards this method. This paper is therefore very welcoming as it presents new findings regarding the age scale of the target atmospheric (Suigetsu) 14C record, uncertainties of this age scale at plateau boundaries, and 14C reservoir age changes in the ocean. Assuming the robustness of the method, the authors suggest that it provides "precise" chronostratigraphic control for marine sediment cores (e.g., in the LGM

[Figure]

Interactive
comment

when other methods are weak), and the opportunity to obtain surface and deep-ocean reservoir ages at high resolution. Additionally, the study covers important aspects such as chronological uncertainties, calendar age controls, bioturbation, ocean ventilation ages, seasonality or habtiat shifts of foraminifera, and the global carbon cycle. In my view, all of these topics overload the paper, as the paper goes off too many tangents, and in fact blurs the main message(s) of the paper. For the sake of its clarity and impact, the paper should re-focus on the advantages and disadvantages of the 14C plateau tuning technique, and resulting surface ocean reservoir ages (with potential comparison to models and existing databases, e.g. Skinner et al. (2017) or Zhao et al., (2018)). This would mean to significantly shorten or remove the bioturbation section and/or repetitions of previous published work e.g. on the carbon cycle.

The paper presents important insights into aspects mentioned above, and emphasizes the benefits and advantages of the 14C plateau tuning technique. I however miss a more nuanced discussion of potential disadvantages of the technique and the underlying assumptions in places. For instance, when is the technique best applied? What are the underlying assumptions, and are there uncertainties associated with these assumptions? What resolution of the tuning record is required?

The paper appears too "crowded", as a number of aspects are discussed: the state of the AMOC and PMOC during the LGM, global circulation changes during the last deglaciation, the global carbon cycle and a model-data comparison. These topics in fact merit their own studies, so I can just recommend once more to streamline the paper and remove redundancy. Some of the findings are presented as "new features of the MOC and carbon cycle" but are in fact based on previous work by the authors (Sarnthein et al., 2013, 2015; Balmer et al., 2016; Balmer and Sarnthein, 2017) and many other studies cited by the authors, so it should be more clearly highlighted what are the new findings. Furthermore, some of the sections appear as a recap of previous papers, e.g., section 3.5.2 (Sarnthein et al., 2015) and section 3.5.3 (Sarnthein et al., 2013), and do not seem to provide new insights. They too might be considered to be
shortened or removed for the sake of clarity.

Below I list major and minor points of criticism that I think should be considered in a revision of the study. I hope that the authors find these comments useful.

Major points: 1) Comparison with latest compilation of surface ocean reservoir age variations of Skinner et al. (2019) and Stern and Lisiecki (2013). The authors have synthesized surface ocean reservoir age records based on the 14C plateau tuning technique that are interpreted for potential driving mechanisms and implications regarding changes in atmospheric CO2. However, it is not clear why other reservoir age estimates have been neglected for instance those based on paired tephra-foraminifera 14C analyses (Skinner et al., 2015; Sikes and Guilderson, 2016) or those resulting from stratigraphic tiepoints (e.g., Waelbroeck et al., 2001). The fact that these estimates are low in resolution, should not diminish their veracity. I strongly recommend that plateau-tuned surface ocean reservoir age estimates are compared with results from other techniques, in particular Skinner et al. (2019).

2) Drivers of 14C plateaus: The causes of 14C plateaus are seemingly not well understand. The plateau tuning technique assumes that oceanic and atmospheric 14C records occur simultaneously, with identical duration and without any temporal offsets, and can unequivocally be identified in the often low(er)-resolution ocean records (see lines 216-217, or line 223). Do all of these conditions always apply?

The authors outline that "air-sea gas exchange transfers the atmospheric 14C fluctuations into the surface ocean" (line 178-179), but it remains unclear how ocean degassing (of 14C-depleted CO2), sea ice and/or wind changes might have affected this one-to-one assumption. In my view, this poses serious challenges to the 14C plateau tuning technique (essentially because these have the potential to cause a phase-shift in atmospheric and oceanic 14C evolution, or skew one independent of the other). These potential caveats are not fully discussed. In light of these issues, I would recommend to address these potential disadvantages of the technique, in particular because

the community has not really embraced this technique yet. It might also be beneficial to tone down some of the overselling language used to describe the technique (e.g., "provides far superior [. . .] evidence" line 218, "sweeping loss" line 266, "clearly, [. . .] a tool indispensable to uncover functional chains in paleoceanography" line 555-556 or the method works "wherever [sediment is] retrieved in the global ocean" line 583). Here I would urge the authors to be more nuanced when addressing the value of the 14C plateau tuning method.

Furthermore, the authors seem to make clear that the assumption "[. . .] these plateau/jump structures are real and widely reproducible in marine sediment records" lines 245-246) remains a speculation. See for instance: "may most likely be regarded as suite of 'real' structures" (line 253). In my view, variations in ocean reservoir ages should be discussed in the light of uncertainties associated with these assumptions, in particular beyond the tree ring chronology.

Line 395-398 and line 477-483: The authors compare the timing of atmospheric 14C plateaus with major changes in the atmospheric CO2 record in order to emphasize the impact of ocean outgassing on the atmospheric CO2 record. Their arguments based on this comparison is inherently weak, as a number of 14C plateaus are not associated with a major shift in atmospheric CO2. This should be acknowledged and discussed in more detail. What causes an temporal agreement between the two, why would the same process not operate at other times of 14C plateaus? What does this tell about the mechanisms driving atmospheric 14C plateaus, and in particular the assumed synchronicity between atmospheric and oceanic 14C changes?

3) Data in preparation by Küssner et al. and Ausin et al. or personal communication 2018 (Line 263): The authors have compiled all existing 14C datasets obtained via the plateau tuning technique (mostly by the group involving the first author), among which are also two new datasets (Küssner et al. and Ausin et al.). I find it hard to follow the findings obtained from these datasets, as crucial metadata is lacking for these cores. It is simply not enough to refer to two in prep. papers, if results are used in the present

manuscript. Full metadata are needed: location, study area, methodology, raw planktic and benthic 14C data and age model strategies/tuning. Given the lack of these pieces of information, statements as in lines 522-529 and in lines 723-733 are difficult to follow, and should be either entirely shifted to the Küssner and Ausin papers in preparation or explained in more detail (with figures and metadata) in the present study.

4) Revision of the Suigetsu age scale: I consider this an important contribution of the paper to the community but I do not find Fig. 5 very informative. What role do siderite layers play (they are not mentioned in the main text)? And how does the figure show age uncertainties, as indicated in lines 352-254. Further elaboration is needed here. Also in lines 385-388, what are the age uncertainties of the Mono Lake and Laschamp paleomagnetic excursions? They should be considered when assessing and comparing different age scales. Why was plateau 2b chosen as test case? How successful is the comparison for any other of 10+ 14C plateaus? I am surprised to see in Table 3 that some plateaus are combined to one long plateau, e.g. 6-7-8. What is the basis for that?

5) Zoophycos burrows (lines 562-576): This whole section is somewhat dubious and not very clearly written. I wonder how useful it is for the review of the 14C plateau tuning technique. The authors seem to suggest that 14C plateaus in host sediments can flag 14C outliers as such, but it is not clear how initial 14C measurements a priori exclude bioturbated material for dating. This should be explained in detail. It is entirely unclear how Zoophyros burrows "help to corroborated changes in MOC and climate". In general, I think that the paper goes off another tangent here. The authors could consider removing this section in my view.

6) Model comparison with Muglia et al. (2018): The authors compare their datasets with the model output of Muglia et al. (2018). Given that other modelling studies exist that studied past ocean reservoir ages variations (e.g., Franke et al., 2008; Butzin et al., 2017), it is unclear why this particular study has been chosen. What characterizes the model run of Muglia et al. (2018), and in particular the modeled AMOC? How were

the simulations forced and what were the LGM boundary conditions? A comparison between the data and model results would be better facilitated by global plots of surface ocean reservoir ages. Is the distribution realistic? It is impossible for the reader to follow statements such as "with estimates of 13 Sv appearing somewhat more consistent with our results." (line 714-715) without any further elaboration or figures. Is the time interval used for comparison the same between 14C data and model data? How was the LGM 14C data obtained? Were several plateaus averaged?

The following long and/or complicated sentences are hard to follow and should be revised: lines 135-144, line 228-232, lines 302-307, line 357-361, line 461-466, and lines 646-649.

Minor comments: Line 110-113: should cite paper(s) on coral reservoir ages here.

Line 116: "that finally turned out to be the most valuable tracer of oceanography" This is in the eye of the beholder, and should be rephrased to "became a valuable tracer for xxx/tool in oceanography"

Line 118: "benthic carbonate particles" should be "benthic foraminifera", also "reflect" might be a better word for "sum" here.

Line 121: Remove "the 14C level of"

Line 135: "provided" instead of "given"

Line 143-144: unclear what iv refers to.

Line 173: jumps

Line 286: remove "developed a special computer program"

Line 442: remove "ones of"

Line 450: consider "propagated error of calendar age uncertainty of a plateau boundary and the uncertainty in its determination"

Line 484: Use (Skinner et al., 2010; Burke and Robinson, 2012) as Southern Ocean reference.

Line 509: "assess" instead of "rate"

Line 639-641: This statement is unclear. Please specify. Do you mean the influence from eddies, AABW formation or the interference of bathymetry with ocean currents?

Line 649-640: Rephrase. "test" instead of "weigh more correctly"?

Fig. 1. It is unclear what the 1:1 line means

Fig. 6. (c) in figure should probably be (b)

References: Balmer, S., Sarnthein, M., 2017. Planktic 14C plateaus: A result of short-term sedimentation pulses? Radiocarbon 59 (1), 3–43. https://doi.org/10.1017/RDC.2016.100 Balmer, S., Sarnthein, M., Mudelsee, M., Grootes, P.M., 2016. Refined modeling and 14C plateau tuning reveal consistent patterns of glacial and deglacial 14C reservoir ages of surface waters in low-latitude Atlantic. Paleoceanography 31, 1–11. https://doi.org/10.1002/2016PA002953 Burke, A., Robinson, L.F., 2012. The Southern Ocean's Role in Carbon Exchange During the Last Deglaciation. Science 335 (6068), 557–561. https://doi.org/10.1126/science.1208163 Butzin, M., Köhler, P., Lohmann, G., 2017. Marine radiocarbon reservoir age simulations for the past 50,000 years. Geophys. Res. Lett. 44, 8473–8480. https://doi.org/10.1002/2017GL074688 Franke, J., Paul, A., Schulz, M., 2008. Modeling variations of marine reservoir ages during the last 45 000 years. Clim. Past 4 (2), 125–136. https://doi.org/10.5194/cp-4-125-2008 Sarnthein, M., Balmer, S., Grootes, P.M., Mudelsee, M., 2015. Planktonic and benthic 14C reservoir ages for three ocean basins, calibrated by a suite of 14C plateaus in the glacial-to-deglacial Suigetsu atmospheric 14C record. Radiocarbon 57 (1), 129–151. https://doi.org/10.2458/azu_rc.57.17916 Sarnthein, M., Schneider, B., Grootes, P.M., 2013. Peak glacial 14C ventilation ages suggest major draw-down of carbon into the

abyssal ocean. Clim. Past 9 (6), 2595–2614. https://doi.org/10.5194/cp-9-2595-2013 Sikes, E.L., Guilderson, T.P., 2016. Southwest Pacific Ocean surface reservoir ages since the last glaciation: Circulation insights from multiple-core studies. Paleoceanography 31 (2), 298–310. https://doi.org/10.1002/2015PA002855 Skinner, L., Mccave, I.N., Carter, L., Fallon, S., Scrivner, A.E., Primeau, F., 2015. Reduced ventilation and enhanced magnitude of the deep Pacific carbon pool during the last glacial period. Earth Planet. Sci. Lett. 411, 45–52. https://doi.org/10.1016/j.epsl.2014.11.024 Skinner, L.C., Fallon, S., Waelbroeck, C., Michel, E., Barker, S., 2010. Ventilation of the deep Southern Ocean and deglacial $CO_2$ rise. Science 328 (5982), 1147–1151. https://doi.org/10.1126/science.1183627 Skinner, L.C., Muschitiello, F., Scrivner, A.E., 2019. Marine reservoir age variability over the last deglaciation: implications for marine carbon cycling and prospects for regional radiocarbon calibrations. Paleoceanogr. Paleoclimatology in press. https://doi.org/10.1029/2019PA003667 Stern, J. V, Lisiecki, L.E., 2013. North Atlantic circulation and reservoir age changes over the past 41,000 years. Geophys. Res. Lett. 40, 3693–3697. https://doi.org/10.1002/grl.50679 Waelbroeck, C., Duplessy, J.C., Michel, E., Labeyrie, L., Paillard, D., Duprat, J., 2001. The timing of the last deglaciation in North Atlantic climate records. Nature 412 (6848), 724–727. https://doi.org/10.1038/35106623

---

## Author Response (AR1)

Dr. Michael Sarnthein, Prof. emer.                    13/26 March 2020
Institut für Geowissenschaften
University of Kiel
Olshausenstr. 40
24098  K i e l,  Germany
<michael.sarnthein@ifg.uni-kiel.de>

Dear Dr. Paul / Dear André,

Below you will find our point-to-point response to the critique raised by two referees, that were constructive and very helpful. Inversely, we regret not having an opportunity yet to include a response to the "extended comment" that Bard & Heaton recently announced to submit to C.P.

Following the advice of Ref. #1 and #2 we have carefully revised and streamlined many sections of our manuscript either following directly the comments of the two referees and/or taking care of their legitimate critique by broadening our empirical evidence and arguments in support of our strategy, interpretations, and conclusions, as outlined in detail below. For a better insight we submit a revised version of both our merged manuscript and the Supplementary Materials, where our recent changes are marked in RED.

We thank you for your great efforts to handle this manuscript that have provided us with the opportunity of manuscript revision, that is, for your help to improve on our manuscript and the quality of our working hypotheses. In turn, we hope and are confident that our manuscript may now satisfy the needs for publication in C.P.

Sincerely,
Michael Sarnthein and Co-authors

........................................................................

In particular we follow your recommendation to consider the following issues:

```
(1) As a highlight in its own right, to present in a dedicated Results section the
"shift to the U/Th modeled Suigetsu chronology" (review #1)/the "new findings
regarding the age scale of the target atmospheric (Suigetsu) 14C record" (review
**2). This would basically correspond to part 1 of the paper (or a hypothetical**
series of two papers) as proposed by referee #1.
```

We now upgrade Section 2 of our manuscript to a "Results section" that further stresses our new findings regarding the age scale of the atmospheric (Suigetsu) 14C reference record.

Here Subsection 2.2 was renamed to "Suigetsu atmospheric $^{14}$C record: Shift to a chronology based on U/Th model ages"

```
(2) ,,,,,, to extend the Discussion section "to expand a bit more to adress a few
more points as other explanations for marine 14C plateaus" and include figures S2
and S3 in the main text (review #1) and "refocus on the advantages and disadvantages
of the 14C plateau tuning technique" and the comparison of the resulting surface
ocean reservoir ages to models, existing databases and results from other techniques
(review #2). This extended Discussion would correspond to part 2 of the paper as
proposed by referee #1.
```

We now start our Results section by a new two-page subsection 2.1 named "Suite of planktic $^{14}$C plateaus: Means to separate global atmospheric from local oceanographic forcings". In this way we try to meet explicitly potential weak points (but disagree with the term "disadvantages") of the $^{14}$C plateau tuning technique. Detailed comparisons to other reconstruction methods of $^{14}$C reservoir ages have already been detailed in subsection 1.2 of the Introduction, a section we do not like to repeat, but slightly enlarge. On the other hand, previous Subsection 3.3 on the definition and origin of Zoophycos burrows is now deleted. Also, Subsection 3.2 (comparison of reconstructed and modeled ocean reservoir ages) is slightly enriched by a reference to results from other reconstruction techniques that – in our view -- have produced less helpful results.

```
Points (1) and (2) would follow up on the suggestions by referee #1 without actually
splitting the paper in two papers.
```

We feel happy to keep basically, though with various necessary modifications, the present organization of our paper.

```
At the same time, I strongly agree with referee #2 that the manuscript would need
some serious streamlining, reduction of redundancy and possibly removal of less
important sections.
```

In harmony with ref. #2 we now streamlined the manuscript, e.g., by deleting a complete subsection (former #3.3). On the other hand, we had to insert a new Subsection 2.1 and many answers to questions of the reviewers, a dilemma with regard to manuscript length. Also, we see a genuine trait of any 'synthesis' paper to include both some recaps of important results already published and a test of the power of the plateau-tuning technique to produce global age markers, that is, to show major implications for our understanding of ocean circulation, climate, and the carbon cycle in the past that make the 'dry' report on a new dating technique more attractive for C.P. readers. Altogether, the length of our text remained constant.

,,,,,,,,,,,,,,,,,,,,,,,,,,,,,,,,,,,,,,,,,,,,,,,,,,,,,,,,,,,,,,,,,,,,,,,,,,,,,,,,,,,,,,,,,,,,,

RESPONSE TO ANONYMOUS REFEREE #1

We thank for a very constructive and helpful review.

**Response to specific comments**

The authors have gone to great lengths to try to organize the paper in a way that helps to wrangle the many topics presented, but I think this paper would benefit from being split into two (companion) papers.

We truly pondered to follow this interesting proposal. For two reasons, however, we see problems in its realization: (1) To be published in a journal named CP any dry report on a new marine dating method also needs to display some major implications in the field of paleoceanography (as now given in Sections 3.1–3.3) to be attractive for CP readers. (2) A subdivision of this manuscript and subsequent review process of two companion papers would be that time consuming that our manuscript might gradually lose timeliness.

,,, the paper could be simplified to enhance accessibility by deleting some of the unnecessary words (hence, thus, moreover etc.).

We are aware that native English readers may suffer from our wording being somewhat biased by a foreign mother tongue, where connectional adverbs appear helpful to generate a 'red thread' of reasoning, though perhaps somewhat colloquial. We went through the manuscript and deleted such words where we felt they did not add to the flow.

I think it is very important that they have used the introduction to address some of the criticism that has been raised with the plateau tuning method. However, I think this is an area that could be expanded a bit more to address a few more points such as other explanations for marine 14C plateaus, including local shifts in air-sea gas exchange or upwelling, and if so, how can we account for these.

This is an important remark. We now broaden our Results section by a two-page long new subsection 2.1 named "*Suite of planktic $^{14}C$ plateaus: Means to separate global atmospheric from local oceanographic forcings*". Based on this resumé we try to meet explicitly potential weak points of the $^{14}C$ plateau tuning technique (points that actually had already been discussed in our previous publications since Sarnthein et al., 2007).

,,, it would be helpful to address why using the Suigetsu record for plateau tuning is preferred to Intcal. It seems that correlation directly to Intcal may be more conservative.

As now added to our text (end of Subsection 1.2), we prefer the Suigetsu record, since it is based on original primary atmospheric data and results in small-scale spatio-temporal changes of reservoir age, whereas IntCal is mixing and smoothing a broad array of different data sources with comparatively coarse time resolution, including carbonate-based marine and speleothem records. Combining these diverse records results in various unsolved problems and assumptions that finally resulted in major differences between the IntCal 13 and 20 records.

(If the paper is split into two separate papers), , , , figures S2 and S3 should be included in the main text of the synthesis paper. I found these figures particularly helpful for visualizing the geographic spacing of Plateau tuned records and their findings.

Figs. S2 and S3 are now included in the main text.

More detail included for sections 3.2-3.3 would also be welcome along with a
picture/diagram of the Zoophycus burrow.

Following the suggestion of Ref.#2 we now have deleted the 'Zoophycos story' of former
Subsection 3.3, in particular, since details are published in detail by Küssner et al., 2018.

**Specific notes**: Figure 1: It isn't clear in the figure caption what the difference
is between the top and bottom panels.

In the figure caption we now specify that the top panel shows the record from 19–29 cal. ka
and the bottom panel that from 10–20 cal. ka.

Figure 7: Please include the references for the records used in this figure.

As cited in the caption of (new number) Fig. 8, all references are given in Table 3 a-c.

It would also be helpful if I could match the data points in the x-y plot to the
data points on the map, perhaps using symbols or colors.

All data now are labeled with the sea region wherefrom a sediment record was obtained.

Also, I'm not sure inclusion of the surface currents in panel b or in figure S3 are
helpful, they make it a bit more difficult to see where the cores are located and to
read the reservoir age differences - especially because the currents are available
in figure S2.

We regard the arrows of surface currents (now Fig. 8) as important for a straight-forward
comparison of different reservoir ages to different sea regions linked to different surface water
currents in east-west direction, important to assess the limits of spatial extrapolation of
reservoir ages.

Lines 601-603: Include the unit the corresponds to the foraminifera habitat depths.
Thanks: (m)

Section 3.4, Lines 656-663: Might be good to mention the interspecies 14C
differences from Lindsay et al., 2015.
Thanks (ref. is now included).

Sections 3.5.1 and 3.5.2: In a few spots it would be helpful if the results from
plateau tuning studies were more clearly emphasized. This would nicely highlight the
important role that this technique has played in our understanding of LGM and HS1
MOC. This is done very nicely in section 3.5.3.

Thanks for an important suggestion. We now highlight some of the results of our new
technique in the text, for instance, the formation of NADW during LGM, the HS-1 reversal of
global MOC, and the reach of NPDW up to the South China Sea during HS-1.

Figure 9: This figure is a bit small and hard to see. The overlapping arrows can
also be a bit confusing.

Indeed, opposed arrows for ocean currents are a bit confusing, since the east-west structures
of the ocean are projected on a simple 2D meridional transect. Many arrows show opposed
directions because of differential Coriolis forcing, especially so in the North Pacific and North
Atlantic (as now specified in figure caption and text).

Overall, I find that this figure is critical for visualizing the findings from sections 3.51-3.5.2,

We hope to display this figure in landscape format spread over a full page of CP. This will require a discussion with the publishing editor.

, , it might be helpful to also include the modern MOC for comparison.

Today the display of a modern MOC may appear too repetitive since it has already been published in that many versions of textbooks.

„„„„„„„„„„„„„„„„„„„„„„„„„„„„„„„„„„„„„„„„„„„„„„„„„„„„„„„„„„„„„„„„„„

**RESPONSE TO ANONYMOUS REFEREE #2**

General statements (paragraphs on page C1 - C2)

-->Thank you for numerous very helpful comments to improve the quality of our manuscript.

The method has not been extensively used outside the first author's group, which might be associated with some sceptisism towards this method.

-->It is common to novel hypotheses and techniques to take some time until their acceptance by a dominantly conservative community that even may try to suppress a new approach. By now, the approach of $^{14}$C plateau tuning has been successfully applied to more than 20 records of marine sediment cores, yielding a consistent output from regions with high sedimentation rates (>10cm/ky).

This paper is therefore very welcoming as it presents new findings regarding the age scale of the target atmospheric (Suigetsu) 14C record, uncertainties of this age scale at plateau boundaries, and 14C reservoir age changes in the ocean.

-->The positive judgement of our efforts is acknowledged.

Assuming the robustness of the method, the authors suggest that it provides "precise" chronostratigraphic control for marine sediment cores

-->We now replace "precise" by "accurate", a term that meets more precisely the quality of our results.

In my view, all of these topics overload the paper, as the paper goes off too many tangents, and in fact blurs the main message(s) of the paper.

-->and:

The paper appears too "crowded", as a number of aspects are discussed: the state of the AMOC and PMOC during the LGM, global circulation changes during the last deglaciation, the global carbon cycle and a model-data comparison. These topics in fact merit their own studies, so I can just recommend once more to streamline the paper and remove redundancy.

-->In our view a 'synthesis paper' on a new, purely 'technical' approach needs to display the wealth of published and unpublished aspects and tangents that result from the technique, an effort that necessarily includes elements of previous publications but does not rank under redundancy. We consider it as genuine trait of a 'synthesis' to include both some recaps of important results already published and various tests of the value of the plateau tuning technique to produce global age markers. A synthesis needs to show major paleoclimatic implications that make the dry report on a new dating technique attractive to readers of CP.

Inversely, we followed the suggestion and deleted the subchapter on Zoophycos burrows. Also, we carved out a clearer separation of the RESULTS from the DISCUSSION AND IMPLICATIONS section.

```
, , , the paper should re-focus on the advantages and disadvantages of the 14C
plateau tuning technique
```

-->and:

```
I however miss a more nuanced discussion of potential disadvantages of the technique
and the underlying assumptions in places. For instance, when is the technique best
applied? What are the underlying assumptions, and are there uncertainties associated
with these assumptions? What resolution of the tuning record is required?
```

-->We follow this helpful suggestion and broaden our Discussion section by a one-page long recap of our previous publications in the new *Subsection 2.1* named "*Suite of planktic $^{14}$C plateaus: Means to separate global atmospheric from local oceanographic forcings* ". In this way we address potential weak points of the $^{14}$C plateau tuning method. We explicitly reject the referee's term "disadvantages". Subsection 2.1 also recaps some technical needs (e.g., minimum sedimentation rates of 10 cm/ky of hemipelagic deposits, minimum $^{14}$C dating resolution of ~100-150 yr, as published in Sarnthein et al., 2007).

```
This would mean to significantly shorten or remove the bioturbation section
and/or repetitions of previous published work e.g. on the carbon cycle.
```

-->The bioturbation section has now been deleted. Other recaps are maintained, as necessary parts of this synthesis paper.

```
The paper appears too "crowded", as a number of aspects are discussed: , , ,
, so I can just recommend once more to streamline the paper and remove
redundancy.
```

-->Our philosophy on the traits of a 'synthesis paper' has been outlined above.

```
, , , so it should be more clearly highlighted what are the new findings
```

-->Thanks for a good suggestion. These are 'new findings' since our last synthesis paper 2015:
• Switch to U/Th-based model ages for $^{14}$C plateaus boundaries in 18/20 sediment cores,
• Inclusion of $^{14}$C records from nine new sediment cores into our synthesis,
• Global data-model intercomparison for LGM and Heinrich-1 surface waters,
• Generation of global deep-water transects.

-->The listing is now incorporated in the text at the end of the Introduction and in Section 3.1.

--> Response to 'major points' and specific questions (page C3-C4-C5)

```
(1) Comparison with latest compilation of surface ocean reservoir age variations of
Skinner et al. (2019) and Stern and Lisiecki (2013). The authors have synthesized
surface ocean reservoir age records based on the 14C plateau tuning technique that
are interpreted for potential driving mechanisms and implications regarding changes
in atmospheric CO2. However, it is not clear why other reservoir age estimates have
been neglected for instance those based on paired tephra-foraminifera 14C analyses
(Skinner et al., 2015; Sikes and Guilderson, 2016) or those resulting from
stratigraphic tie points (e.g., Waelbroeck et al., 2001). The fact that these
estimates are low in resolution, should not diminish their veracity. I strongly
recommend that plateau-tuned surface ocean reservoir age estimates are compared with
results from other techniques, in particular Skinner et al. (2019).
```

-->Detailed comparisons to other methods for tie points to reconstruct past variations in surface ocean reservoir age were given in Subsection 1.2 of the Introduction, in particular in "paragraph 2", that we do not like to repeat in the discussion. Now, however, no problem to enrich the paragraph by additional remarks and references to authors listed by Ref.#2.

-->• To be fair, the paper of Skinner et al. (2019) appeared only two weeks after our manuscript submission. Of course, we appreciate to read that these authors -- like ourselves -- now fully recognize the concept of strongly variable spatiotemporal patterns of surface water reservoir age. We feel happy to include into our discussion the results of this single-core study, that also includes detailed (though in part somewhat debatable) alignment of paleoclimate records.

-->• We agree, each low-resolution age tie point *per se* has full veracity. Being spaced over 5-10 ky, however, these tie points cannot depict the actual dramatic short-term variability of res. ages now revealed for glacial-to-deglacial records by means of $^{14}$C plateau tuning, hence won't provide the accurate timing of paleoclimatic events needed for proper age correlation.

-->• Stern and Lisiecki (2013) indeed had been cited in "§2", but unfortunately deleted in a "streamlining action" prior to our manuscript submission. Based on pers. comm. with L. Lisiecki (Cambridge, 9-2018), however, we (M.S.) learned that the uncertainty range of their age assignments for the LGM amounts to ±1.5–±2.0 kyr, a range problematic for any proper correlation of multi-centennial-scale events in paleoceanography.

-->• Valuable age tie points and reservoir ages were derived from $^{14}$C ages of planktic foraminifera paired with $^{14}$C dated tephra layers as cited in "§2" of Subsection 1.2. Most of these tie points, however, are far too wide-spaced in peak glacial-to-early deglacial sediment records to meet the need to specify millennial-scale LGM-to-deglacial changes in ocean circulation, which may involve wrong age correlations. A rare deglacial suite of four tephra-based reservoir ages off Chile (Siani et al., 2013) was clearly reproduced, its variability, however, was much refined by $^{14}$C plateau-based reservoir ages (Küssner et al. (subm.). Accordingly, we now broaden our discussion of tephra-based ages in "§2". -- Tephra-based results of Waelbroeck et al., 2001 and 2011 had been properly cited in our text.

2) Drivers of 14C plateaus: The causes of 14C plateaus are seemingly not well understand. The plateau tuning technique assumes that oceanic and atmospheric 14C records occur simultaneously, with identical duration and without any temporal offsets, and can unequivocally be identified in the often low(er)-resolution ocean records (see lines 216-217, or line 223). Do all of these conditions always apply?

-->and:

The authors outline that "air-sea gas exchange transfers the atmospheric 14C fluctuations into the surface ocean" (line 178-179), but it remains unclear how ocean degassing (of 14C-depleted CO2), sea ice and/or wind changes might have affected this one-to-one assumption.

-->We now added the new *Subsection 2.1* on the "*Suite of planktic $^{14}$C plateaus: Means to separate global atmospheric* etc." to avoid this misunderstanding: Indeed, only plateaus during which local oceanograpy did not change significantly will correlate and without much delay or change in duration in view of the rapid atmosphere-surface ocean exchange. Thus, the possibility of local ocean effects necessitates the use of a complete suite of $^{14}$C plateaus (and their estimates of reservoir age) to achieve a best-possible tuning of each sediment core. In this case one or two disturbed plateaus (out of up to 19) won't hinder valuable results for the others and, at the same time, provide evidence of local short-term oceanographic changes. Finally, our records show that reservoir ages generally vary with climate changes on time scales of 1000 and more years in contrast to generally much shorter atmospheric plateaus.

In my view, this poses serious challenges to the 14C plateau tuning technique

-->These changes are at least as challenging for IntCal where a close correspondence between $^{14}$C levels in the surface ocean and in the atmosphere is assumed to justify the use of ocean carbonate records for the reconstruction of atmospheric $^{14}$C. A comparison of IntCal13, IntCal20, and MarineCal20 for the time period older than 14 cal. ka shows how much work still needs to be done and how plateau tuning can contribute valuable data. In contrast to Marine Cal20 and IntCal where the timescale of the ocean record is largely based on climate wiggle matching beyond 14 cal ka our plateau tuning is based on atmospheric wiggle matching.

,,,, These potential caveats are not fully discussed.

-->To meet this objection we now added the new *Subsection 2.1.*

recommend to address potential disadvantages , , , ,
-->and:

the community has not really embraced this technique yet.

-->The tuning technique does not have "disadvantages", rather potential weak points that need consideration. At least the technique is better than most other ways to produce R data with high spatial and centennial-to-millennial-scale resolution.

It might also be beneficial to tone down some of the overselling language.

-->We agree, it never is good to oversell. E.g., we may replace "far" by "a bit" superior.

,,, the method works "wherever [sediment is] retrieved in the global ocean"

-->Indeed, we need to specify and say: "wherever cores with high hemipelagic sedimentation rates are sampled"

Furthermore, the authors seem to make clear that the assumption "[. . . ] these plateau/jump structures are real and widely reproducible in marine sediment records" (lines 245-246) remains a speculation.

-->The objection that our assumption remains a speculation is simply not true. Our assumption is, as stated, general and common with those of IntCal. Each new sediment record that, after high-resolution dating, provides a suite of jumps and plateaus that can be correlated with the atmospheric master curve and our planktic records, confirms the practical applicability of the assumption. These findings are no "speculation" but apply to $^{14}$C records of by now 20 (plus one more recent, yet unpublished) sediment cores from the global ocean.

-->Yet, we agree, "one always needs to be cautious". Therefore, we insert a "most likely" because signs of local disturbing influences must always be looked for.

Line 395-398 and line 477-483: The authors compare the timing of atmospheric 14C plateaus with major changes in the atmospheric CO2 record in order to emphasize the impact of ocean outgassing on the atmospheric CO2 record. Their arguments
based on this comparison is inherently weak, as a number of 14C plateaus are not
associated with a major shift in atmospheric CO2. This should be acknowledged and
discussed in more detail.

-->In (former) L. 501-507 we had already acknowledged the weakness of our understanding of several causal links that might have controlled the origin of atmospheric plateaus and jumps. We broached pulsed ocean outgassing as one amongst various other forcings and now terminated (present) Subsection 2.3 with the following sentence (now L. 531):
<<However, there is still little information on the origin of several other peak glacial $^{14}$C plateaus 17.5–29 cal. ka. The actual linkages of these plateaus to events in ocean MOC still remain to be uncovered.>>

What causes a temporal agreement between the two, why would the same process not
operate at other times of 14C plateaus? What does this tell about the mechanisms
driving atmospheric 14C plateaus, and in particular the assumed synchronicity
between atmospheric and oceanic 14C changes?

-->Reformulated *Subsection 2.3* tries to make clear that due to the many factors involved it is often not possible to pinpoint a single dominant forcing of an atmospheric $^{14}$C plateau. In some plateaus we can recognize the role of $^{14}$C production (via $^{10}$Be), while three others coincide with changes in $CO_2$ level shown by ice cores that indicate ocean outgassing and oceanic changes. The difference in length between the plateaus and the outgassing spikes suggests that the spikes were part of longer lasting oceanic processes. Such processes may also be the origin of other plateaus without a recognizable $CO_2$ signal. We here demonstrate that in some cases the origin of atmospheric $^{14}$C changes can be identified. Increased use of plateau tuning may provide data that will allow identification of further determining factors.

-->Also, the fairly fast air-sea transfer of atmospheric $^{14}$C signals that leads to a quasi-synchronicity was displayed at (former) L. 102-110.

3) Data in preparation by Küssner et al. and Ausin et al. or pers. communication
(Line 263): The authors have compiled all existing 14C datasets obtained via
the plateau tuning technique (mostly by the group involving the first author), among
which are also two new datasets (Küssner et al. and Ausin et al.). I find it hard to
follow the findings obtained from these datasets, as crucial metadata is lacking for
these cores.

-->Küssner and Ausin are coauthors of this manuscript. In the meantime, the manuscript of Küssner et al. that we regard as important brick of this synthesis, has now been submitted to P&P. Also, the datasets of Küssner et al. are stored at PANGAEA.de, saved by a password until acceptance of this manuscript for publication.

-->For the $^{14}$C record of Ausin et al., we may refer to their records and results as "unpubl. data". Unfortunately, the manuscript of Ausin et al., perhaps particularly interesting to Ref.#2, will be ready for submission only later this spring.

4) Revision of the Suigetsu age scale: I consider this an important contribution of
the paper to the community but I do not find Fig. 5 very informative. What role do
siderite layers play (they are not mentioned in the main text)? And how does the
figure show age uncertainties, as indicated in lines 352-254. Further elaboration is
needed here.

--> Our text is supplemented in Section 2.2 and the caption of Fig. 5.

Also in lines 385-388, what are the age uncertainties of the Mono Lake and
Laschamp paleomagnetic excursions? They should be considered when assessing
and comparing different age scales.

-->We now refer to data of Lascu et al. (2016).

Why was plateau 2b chosen as test case? How successful is the comparison for any other of 10+ 14C plateaus?

-->Unerring question! In contrast to optical varve counts reaching back to >40 cal. ka, the base of $^{14}$C Plateau 2b marks the oldest tie point captured by XRF-based varve counts. Marshall et al. (2012) showed that the difference between optical and XRF-based varve numbers is modest back to ~15 varve ka. Before, it has risen dramatically back to 17 ka.

I am surprised to see in Table 3 that some plateaus are combined to one long plateau, e.g. 6-7-8. What is the basis for that?

-->All right, we need to mention: Table 3 is grouping the $^{14}$C plateaus of some climate units that can be reproduced by the time slots employed for the model-based res. ages of Muglia et al. Within these slots, the plateau-based res. ages are largely constant.

5) Zoophycos burrows (lines 562-576): This whole section is somewhat dubious and not very clearly written. I wonder how useful it is for the review of the 14C plateau tuning technique. The authors seem to suggest that 14C plateaus in host sediments can flag 14C outliers as such, but it is not clear how initial 14C measurements a priori exclude bioturbated material for dating. This should be explained in detail. It is entirely unclear how Zoophyros burrows "help to corroborated changes in MOC and climate". In general, I think that the paper goes off another tangent here. The authors could consider removing this section in my view.

-->The section on Zoophycos burrows has now been removed. Küssner et al., 2018, already gave all details on the principles how to separate foram tests of Z. burrows from those of the ambient host sediment.

6) Model comparison with Muglia et al. (2018): The authors compare their datasets with the model output of Muglia et al. (2018). Given that other modelling studies exist that studied past ocean reservoir ages variations (e.g., Franke et al., 2008; Butzin et al., 2017), it is unclear why this particular study has been chosen. What characterizes the model run of Muglia et al. (2018), and in particular the modeled AMOC?

-->The choice of model data of Muglia et al. was linked to (1) the simple availability of pertinent recent model data and explanations, kindly provided by Juan Muglia as coauthor. (2) Butzin informed me, when testing his data as alternative dataset and preparing this manuscript in spring 2019, that his dataset (then not particularly successful by comparison with our dataset) was outdated and that his group was preparing a major improvement of their model concept. (3) With regard to res. ages modeled by Franke et al. (2008), we see major age differences found both for high-latitude and upwelling regions. -- In summary, we now introduce three different modelling studies at display and their availability for our test (new L.588-607).

How were the simulations forced and what were the LGM boundary conditions? A comparison between the data and model results would be better facilitated by global plots of surface ocean reservoir ages.

-->To avoid extending the manuscript up to 'textbook' length, we now just add a brief remark on the background of the model simulations.

Is the distribution realistic? It is impossible for the reader to follow statements such as "with estimates of 13 Sv appearing somewhat more consistent with our results." (line 714-715) without any further elaboration or figures.

-->We now give a short explanation for our choice of MOC strength for the LGM: Validation test by Muglia et al. (2018) through a model-data comparison of radiocarbon data compiled by Skinner et al. (2017).

Is the time interval used for comparison the same between 14C data and model data? How was the LGM 14C data obtained? Were several plateaus averaged?
-->Time intervals and average of several plateaus are given in Table 3, as listed in Fig. 8 caption. Both model-based and observed reservoir ages focus on the Late LGM, 21-18.7 cal. ka (14C plateaus no. 4-5).

Response to 'minor comments'

The following long and/or complicated sentences are hard to follow and should be revised: lines 135-144, line 228-232, lines 302-307, line 357-361, line 461-466, and lines 646-649.

-->We now tried to simplify the sentences demurred by the reviewer.

Line 110-113: should cite paper(s) on coral reservoir ages here.
-->Without expanding our manuscript up to 'textbook' length we now add a few refs in addition to Adkins & Boyle, 1997.

Line 116: "that finally turned out to be the most valuable tracer of oceanography" This is in the eye of the beholder, and should be rephrased to "became a valuable tracer for xxx/tool in oceanography"
-->As authors we may regard it legitimate to present these findings in our eye as beholders.

Line 118: "benthic carbonate particles" should be "benthic foraminifera", also "reflect" might be a better word for "sum" here. -- o.k.

Line 121: Remove "the 14C level of" -- o.k.

Line 135: "provided" instead of "given" -- o.k.

Line 143-144: unclear what iv refers to.

-->Item 'iv' is now reworded to a separate sentence that spells out the great need for a generally accepted high-precision atmospheric reference record , , ,

Line 173: jumps --> o.k.

Line 286: remove "developed a special computer program"
--> o.k.(paragraph was deleted)

Line 442: remove "ones of" --> o.k.

Line 450: consider "propagated error of calendar age uncertainty of a plateau boundary and the uncertainty in its determination"

-->We agree but do not follow the idea to include the fishy statement "*and the uncertainty in its determination*".

Line 484: Use (Skinner et al., 2010; Burke and Robinson, 2012) as Southern Ocean reference. --> o.k.

Line 509: "assess" instead of "rate"

-->We prefer "rate".

Line 639-641: This statement is unclear. Please specify. Do you mean the influence from eddies, AABW formation or the interference of bathymetry with ocean currents?  -

-->o.k.: We mean a variability linked to small-scale frontal systems, upwelling cells, and the interference of ocean currents with small-scale bathymetry.

Line 649-640: Rephrase. "test" instead of "weigh more correctly"? --> o.k.

Fig. 1. It is unclear what the 1:1 line means

--> The 1:1 line shows a gradient of one $^{14}$C yr per cal. yr

Fig. 6. (c) in figure should probably be (b) --> o.k., thank you!

xxxxxxxxxxxxxxxxxxxxxxxxxxxxxxxxxxxxxxxxxxxxxxx

**MANUSCRIPT  WITH  AUTHORS' CHANGES ARE MARKED UP in RED**

[revised manuscript text omitted]
 (1) Some means to separate atmospheric and oceanic forcings, that overlap in controlling the structure of a planktic $^{14}$C plateau.

(2) Choice of a U/Th-based reference time scale (Bronk Ramsey et al. 2012; Cheng et al., 2018) instead of the earlier varve-counted version (Schlolaut et al., 2018) to date the short-term structures in the global atmospheric [14]C record of Lake Suigetsu (Sarnthein et al., 2015).

(3) An extension of the suite of age tie points from 23 back to 29 cal. ka, values crucial for an accurate global correlation of ocean events over the period 10–29 cal. ka.

Supplement Text no. 1 is discussing uncertainties in age-calibrated [14]C plateau boundaries and jumps of our 
[revised manuscript text omitted]
. S2; Küssner et al., 2020 subm.) have short-term dropped around 18 cal. ka from 4000 to 1000 yr (Fig. 4c). Over this time, the $^{14}$C level of the atmosphere was constant at 1.4 FMC (Fraction of Modern Carbon). Hence a ventilation age of 4000 yr is equal to  ~60 % of the contemporaneous level of past atmosphere 1.4 FMC, at that time leading to 1.4 x 0.6 = 0.84 FMC. In turn, the ventilation age of 1000 yr was equal to 88

% of past atmosphere FMC. This implies an increase of local deep ocean $^{14}$C to 1.4 x

0.88 = 1.232 FMC at this site. The concentration difference of ~0.4 FMC means a major

$^{14}$C shift in DIC at that very MOC key region of the deep Southern Ocean (Rae and

Broecker, 2018) over 200 yr. This enhanced mixing of the Southern Ocean and a similar, slightly later mixing event in the North Pacific (MD02-2489; Fig. S2d) may have triggered – with phase lag – two trends in parallel, (1) a rise in atmospheric $CO_2$, in part abrupt (*sensu* Chen et al., 2015; Menviel et al., 2018), and (2) a gradual enrichment in

$^{14}$C depleted atmospheric carbon, reflected as $^{14}$C plateau.

Plateau 6a matches a $^{14}$C plateau deduced from atmospheric $^{10}$Be concentrations, thus suggests changes in $^{14}$C production. Other changes in atmospheric $^{14}$C (plateaus 4 and

8) match short-term North Atlantic warmings during peak glacial and earliest deglacial times, similar to that at the end of HS-1 and during plateau 'YD', hence may reflect minor changes in ocean circulation and ocean-atmosphere exchange without major degassing of old $^{14}$C depleted deep waters in the North Atlantic (Table 2, Fig. S2a).

There is still little information, however, on the origin of several other peak glacial $^{14}$C

plateaus 17.5–29 cal. ka. The actual linkages of these plateaus to events in ocean MOC

still remain to be uncovered.

3. DISCUSSION and IMPLICATIONS

*3.1 $^{14}$C plateau boundaries – A suite of narrow-spaced age tie points to rate short-term*

*changes in marine sediment budgets, chemical inventories, and climate 29–10 cal. ka*

In continuation of previous efforts (Sarnthein et al., 2007 and 2015) the tuning of high- resolution planktic $^{14}$C records of ocean sediment cores to the new age-calibrated atmospheric $^{14}$C plateau boundaries now makes it possible to establish a 'rung ladder'

of ~30 age tie points covering the time span 29 – 10.5 cal. ka. These global tie points have a time resolution of several hundred to thousand years, and can be used to constrain the chronology and potential leads and lags of events that occurred during peak glacial and deglacial times (Fig. 1). The locations of the 18 (20) cores are shown in

Fig. 7. The time histories of the benthic and planktic reservoir ages are summarized in

Figs. 8 and S2 and the information these provide is discussed below.

Six prominent examples showing the power and value of additional information obtained by means of the $^{14}$C plateau-tuning method are:

(i) Signals of the onset of northern hemisphere deglaciation can now be distinguished in detail from the subsequent beginning of deglaciation in the southern hemisphere and reveal that changes began ~1400 yr earlier in the north (Fig. S2) (Kawamura et al.,

2007; Küssner et al., 2020 subm.; in harmony with Schmittner and Lund, 2015).

(ii) In southeast Pacific surface waters the end of the Antarctic Cold Reversal (ACR;

WDC Project Members, 2013) was found precisely coeval with the onset of the Younger

Dryas cold spell (Küssner et al., 2020 subm.), a finding important to further constrain the details of 'bipolar see-saw' (Stocker and Johnsen, 2003).

(iii) Signals of deep-water formation in the subpolar North Pacific can now be separated from signals originating in the North Atlantic (Rae et al. 2014; Sarnthein et al.,

2013). In this way we now can specify and tie major short-lasting reversals in Atlantic and Pacific MOC on a global scale.

(iv) Signals of deglacial meltwater advection can now be distinguished from short- term interstadial warmings in the northern subtropical Atlantic, which helps to locate meltwater outbreaks far beyond the well-known Heinrich belt of ice-rafted debris (Balmer and Sarnthein, 2018).

(v) As outlined above, the timing of marine $^{14}$C plateaus can now be compared in detail with that of deglacial events of climate and the atmospheric $CO_2$ rise independ- ently dated by means of ice core-based stratigraphy (Table 2; Fig. 6). These linkages offer a tool to explore details of deglacial changes in deep-ocean MOC once the suite of

$^{14}$C plateaus has been properly tuned at any particular ocean site.

(vi) The refined scale of age tie points also reveals unexpected details for changes in the sea ice cover of high latitudes as reflected by anomalously high $^{14}$C reservoir ages (e.g. north of Iceland and near to the Azores Islands) and for the evolution of Asian summer monsoon in the northern and southern hemisphere as reflected by periods of reduced sea surface salinity (e.g., Sarnthein et al., 2015; Balmer et al., 2018).

Finally, the plateau-based high-resolution chronology has led to the detection of numerous millennial-scale hiatuses (e.g., Sarnthein et al., 2015; Balmer et al., 2016;

Küssner et al., 2020 subm.) overlooked by conventional, e.g., *AnalySerie*-based methods (Paillard et al. 1996) of stratigraphic correlation (Fig. S2). In turn, the hiatuses give intriguing new insights into past changes of bottom current dynamics linked to different millennial-scale geometries of overturning circulation and climate change such as in the South China Sea (Sarnthein et al., 2013 and 2015), in the South Atlantic (Balmer et al. 2016) and southern South Pacific (Ronge et al., 2019).

Clearly, the new atmospheric $^{14}$C 'rung ladder' of closely-spaced chronostratigraphic tie points has evolved to a valuable tool to uncover functional chains in paleoceanography, that actually have controlled events of climate change over glacial-to-deglacial times.

*3.2 Observed vs. model-based $^{14}$C reservoir ages acting as tracer of past changes in*

*surface ocean dynamics and as incentive for model refinements (Fig. 8)*

The atmospheric $^{14}$C plateaus of Suigetsu provide a suite of up to 18 reference plateaus over the time span 10 – 29 cal. ka (Fig. 1). Tuning $^{14}$C plateau boundaries in $^{14}$C-dated marine sediment sections to the Suigetsu $^{14}$C record allows us to establish a suite of highly resolved and robust age tie points on short and long time scales, wherever cores with high hemipelagic sedimentation rates are sampled in the global ocean (Fig. 7). In addition, and likewise intriguing, 
[revised manuscript text omitted]

[Figure]

,,,,,,,,,,,,,,,,,,,,,,,,,,,,,,,,,,,,,,,,,,,,,,,,,,,,,,,,,,,,,,,,,,,,,,,,,,,,,,,,,,,,,,,,,,,,,,,,,,,,,,,,,,

,,,,,,,,,,,,,,,,,,,,,,,,,,,,,,,,,,,,,,,,,,,,,,,,,,,,,,,,,,,,,,,,,,,,,,,,,,,,,,,,,,,,,,,,,,,,,,,,,,,,,,,,,,

---

## Author Response (AR2)

Dr. Michael Sarnthein, Prof. emer.                                    8 June 2020
Institut für Geowissenschaften
University of Kiel
Olshausenstr. 40
24098 K i e l, Germany
<michael.sarnthein@ifg.uni-kiel.de>

Dear Dr. Paul / Dear André,

Thank you for handling with care our manuscript. Below you will find our point-to-point response to the most recent remarks of a referee, that (once more) helped us a lot to further improve the quality of our manuscript. Recent changes of the text are marked **red** in the manuscript we submit.

**Major comments**.  Highlighting the major findings/points:
I welcome the added outline of the manuscript in lines 274-317 that simultaneously emphasizes the highlights of the manuscript. But paragraph 274-317 needs to be significantly shorted to one paragraph in my view. Any introduction of supplementary material is unnecessary here.

The paragraphs of lines 274-317 now were shortened by half a page to one pararaph.

A novel aspect of the study seems to be the extension of the plateau tuning technique to 23-29 kyr BP, which allows to define new chronostratigraphic markers for that time interval. However, reservoir age estimates from this time interval are not discussed at all

All right! We now added three lines of text (new lines 537-539). At least two records show reservoir ages covering HS-2 located prior to 23 cal. ka.

The reassessment of earlier records with the adjusted chronology for the LGM and younger time interval seems to support previous findings (judging from the references given to earlier work of the first authors' group in the discussion). I think the paper would benefit from highlighting the knowledge gained form adjusting the chronology of the records rather than reiterating what these records as a whole show because this was done in numerous previous publications of the first authors' group. at least this could be done in a more concise way. Currently, it is hard to distill the main messages of the paper, in particular in the "Discussion and implications" section.

The conclusions now spell out, paragraph-by-paragraph, which results we regard as main and "new" messages of this paper, that is, all results based on the global relevance of atmospheric $^{14}$C structures for stratigraphy.

When the authors write "These new features of MOC and the carbon cycle" (line 55) and "a new understanding of Ocean MOC during the LGM and its reversal during HS-1" (line 316) it is difficult to say what is exactly the novel piece of information here, also in comparison to the previous knowledge obtained from other proxies. This should be rewritten and/or more clearly carved out.

Our new results have been compared with previous knowledge obtained from other proxies ($\delta^{13}$C, Nd isotopes), e.g., in Section 3.3.1 and 3.3.2 (new line 673-680, 695-696, 753, and 759-764).

In those instances, where new insights can be gained and are introduced, I have to admit that it is really hard to follow the argumentation, in particular when the (raw) data is not discussed or described (as in the case of the Küssner or Ausin data). This applies for instance to line 526: "reveal that changes began ~1400 yr earlier in the north (Fig. S2)", where it is unclear what is meant by "north", how this number of 1400 yr came about, and what is meant by changes.

The first two examples/statements in the delay of Northern Hemisphere deglaciation (Ch. 3.1; new lines 496-501) were now reworded. The '1400-yr delay' was deleted. Also, the 2nd statement was corrected to better specify the role of local warmings and coolings. The full argumentation of the conclusions i – vi, based on the raw data and their summary in Figs. 8, 10, and S2 cannot be given here because of restricted space.

The same applies to lines 529-532 or to line 709-710 "in particular due to a 'thermal threshold' (Abé-Ouchi, pers. comm.) overlooked in other model simulations.".

The 'thermal threshold' includes the idea of a decreasing influence of the temperature influence on differential densities and deep-water formation near to the lower temperature limit of seawater (diagram of Cox, 1969).

**Streamlining**
I think some paragraphs are still long-winded and contain a lot of detail that are likely not necessary for the main message of that paragraph. This applies to the first and second paragraph of section 2.2. Also, I find the exercise performed in lines 440-464 to highlight the ocean carbon cycle influence on atmospheric 14C somewhat redundant. I think the same message (i.e. atmospheric 14C variability cannot be explained by production changes alone) can be conveyed by simply referring to existing literature, e.g. (Hain et al., 2014) or others; hence allowing to shorten the paragraph.

-- The 1st two paragraphs of section 2.2 form a basic portion of our results, a dilemma, since they also need to be digested by 'outsiders', say modelers. To follow the reviewer, however, we shortened them by ~2 lines.

-- Former lines 440-464.

We are happy to add a citation of Hain et al. (2014). Their statements, however, are solely based on model simulation whereas our text is based on actual data. Thus we regard this paragraph essential for our manuscript, however, we now have shortened it by two lines.

In lines 486-491, I find the calculation of F-modern equivalent of a ventilation age change unnecessary. This can be removed in my view.

In view of most recent evidence for improved plateau tuning at Site SO76-136 we now deleted the indicted paragraph.

Also the section on the influence of habitat changes on planktic foraminiferal 14C ages could be summarized in one sentence, to my mind (e.g. "The fact that 14C ages of co-existing planktic foraminifera were suggested to be influenced by habitat depths, in particular when comparing surface and sub-surface dwellers (refs), or seasonality (refs), makes a closer specification of model results as product of different seasonal extremes a further target.").

(Former lines 647-660) The paragraph now was shortened by two lines. However, we regard these results essential to present specific findings (by now published in a micropaleontological journal only) instead of mere arm waiving.

Model-data comparison (Section 3.2.)

I am still confused why the study of Muglia et al., (2018) was chosen for model data comparison, and what the purpose of it is. This is in part because the background on the actual model simulation is missing (how were the reservoir ages estimated? How was the model forced?). Although some of this information is probably given in Muglia et al. (2018), in my view this information should be stand-alone in the paper. The comparison to previous surface reservoir modelling studies (Franke, Butzin) is quite limited and a reiteration of personal communications should be avoided. It seems that the authors have chosen the Muglia study because it seemingly reproduces the best match to their data, but what is the purpose then of the model data
comparison?

– We now slightly modified and abbreviated the title of section 3.2. The aim of our data, generated by plateau tuning, is to provide additional boundary conditions on which to test and further refine ocean circulation models. Our comparison indicates that models using more detailed input and refinement better reproduce the spatial and temporal variability revealed in our data.

– On the whole, section 3.2 is summarizing a number of crucial aspects, figures, and caveats linked to foraminifera-based reservoir ages, that we do not like to delete.

– Six lines were abbreviated in §1, three lines in § 2. New line 575: The personal advise of Butzin was crucial for the generation of this manuscript version, thus needs to be reported, the wording, however, was further reduced.

– On the whole we shortened our manuscript text by ~30 lines.

**Unpublished data**
I still do not support the inclusion of the Ausin et al. unpublished data in the interpretation, and the Küssner et al. submitted, simply because this cannot be assessed by the reader at the moment and likely not at the time of publication. Simply referring to the Pangaea database (in the replies to the reviewers) is not a best practice to discuss the new data, explain how they have been obtained and how surface R were estimated, etc. I expect a better solution here.

**and EDITOR's LETTER**
Not all data are described and not all arguments are explained, there is merely a reference to a supplementary figure (S2) or unpublished data, which I do not find satisfactory.

The foraminifera-based data sets of Ausin and Küssner et al. are stored at PANGAEA databank, under embargo as the papers are still in preparation. As with all other $^{14}$C records, the R values at these five sites were estimated precisely on the basis of the techniques shortly outlined in section 2.1, with more detail in Sarnthein et al. (2015). The caption of Fig. S2 now is explicitly mentioning the data storage at PANGAEA databank.

With regard to Küssner et al. (manuscript largely ready for submission), we feel sorry that for reasons of serious disease, the manuscript has not been further processed yet for submission, thus needs to be downgraded to "in prep.", though containing a rich collection of four crucial records of past changes in R values.

**Minor comments**:
Line 199. Far superior evidence -> is not adjusted although described in the replies to reviewers.

In our view the term 'far superior' appears justified since the plateau-tuning technique

provides more accuracy, less uncertainty and far better time resolution of other R value records. Also important, it provides a coherent set of R values instead of rare and widespread single estimates.

Line 265-269 Move "We prefer the Suigetsu record to IntCal […] including carbonate-based speleothem and marine records." to the preceding paragraph.

Thanks! We agree.

Line 415. Reference your earlier work here.f

A ref. to Sarnthein et al., 2015, is already given in the caption of Fig. 3.

Line 486. There is no Fig. 4c that references the Küssner data subm.

The paragraph was now deleted.

Throughout manuscript: replace plankton-based with planktic foraminifera-based (planktic foraminifera are plankton but not all plankton is planktic foraminifera)

Done.

Again we thank for your handling of our manuscript with that much care. Hoping that the quality of our manuscript may now be sufficient for being accepted for publication,

Sincerely,

Michael Sarnthein and coauthors

---

## Author Response (AR3)

- 07- 16
To the Editor of "CLIMATE OF THE PAST"

Dear Dr. Paul / Dear André,

Thank you for the fast editor's decision of June 24 (copy attached) and his helpful detailed comments. In response, we now like to comment on the editor's three remaining major points, the first one is actually critical for finally accepting your manuscript for publication:

(1) Unpublished data and manuscripts "in prep."

There are two unpublished data sets related to two manuscripts by Ausin et al. and Küssner et al. that are both still "in prep." (by the way, the manuscript by Ausin et al. is not even cited). The data sets are stored at PANGAEA, but "under embargo as the papers are still in preparation", hence they are not yet available in any way, and therefore it is impossible to evaluate them. Both the referee and myself expressed that we are not satisfied with this situation. We understand that it is highly frustrating for you as authors of this manuscript, too. However, in its current form it cannot be published using those data.

With regard to the four options proposed by the editor to cope with the "difficult situation" of our manuscript, we decided to ask for option (a), that is, to put our manuscript "on hold" until the two manuscripts of Küssner et al. and of Ausin et al. will have been submitted to a peer-reviewed journal and the embargo on the related data sets on PANGAEA will be lifted. In both cases the authors are working hard to finalize their manuscripts soon over the upcoming weeks.

(2) Personal communication ("pers. com")

On several occasions, you refer to unpublished, personal communications. As there is no way to check those for correctness, I ask you to remove all personal communications from the text and rephrase the text accordingly. If you want, you can still mention the persons you talked to in the Acknowledgment section.

To cope with this problem of three unpublished personal communications we are happy to follow the editor's suggestion to replace that of Bronk Ramsey (former line 249) by a reference to his paper now published under 2020, that of Butzin (former line 574) by a wording proposed by the editor himself to replace our own wording of former lines 572–577, and that of Abé-Ouchi by a reference to her abstract for the OC3/IPODS Meeting held in Cambridge in September 2018.

(3) Length of manuscript

Another issue is the sheer length of the manuscript, which is 850 lines plus 100 lines and a couple of figures in the supplement. Attempts to shorten the manuscripts as recommended by the referees and myself have only been partly successful. Therefore, I would welcome further attempts to shorten, for example, the abstract and the introduction.

We now shortened the abstract by 10% (by 2 lines) and the introduction by 14% (by 36 lines). A one-page abstract (now 343 w.) may be appropriate for a major synthesis paper. Pertinent text sections are marked-up RED in the revised manuscript. With regard to the overall file size, we'd like to mention that we transferred several figs. from the Suppl. Materials to the main manuscript following the advice of a reviewer.

With sincere thanks for your care in editing and helpful suggestions, and kind regards,

Michael Sarnthein and coauthors

[revised manuscript text omitted]

(a) EMPIRIC LGM SURFACE-WATER RESERVOIR AGES

(b) MODELED RESERVOIR AGES OF LGM SURFACE-WATERS (Muglia)

[Figure]

[Figure]

∀ Fig. 9. SW–NE transect of [14]C reservoir age and changes in ventilation age across sites GIK17940 and SO50-37 in the South China Sea during late LGM ([14]C Plateaus 5

and 4; upper panel) and HS-1 (lower panel). Insert map shows location of transect and core locations. Core locations are given in Fig. 7. An extreme epibenthic $\delta^{13}$C minimum in far southwest (Core GIK17964; Sarnthein et al., 1999) reflects an LGM incursion of

Lower/Upper Pacific Deep Waters (L./ U. PAC DW) with extremely high $^{14}$C ventilation age and DIC enrichment in contrast to a low ventilation age of North Pacific Deep Water (N. PAC DW). Arrows show direction of potential deep and intermediate-water currents.

[Figure]

⊻ Fig. 10. 2D transects of the geometries of global ocean MOC. Arrows (blue = high, yellow = poor ventilation) suggest average deep and intermediate-water currents that follow the gradient from low to high benthic ventilation ages based on paired planktic

$^{14}C$ reservoir ages derived by means of $^{14}C$ plateau tuning technique (Sarnthein et al.,

2013, Balmer et al., 2018, Küssner et al., 2020 subm.). At some Pacific sites reservoir ages are based on paired $^{14}C$ ages of planktic foraminifera and wood chunks (marked by green 'w'; Sarnthein et al., 2015; Zhao and Keigwin, 2018, Rafter et al., 2018). Red arrows suggest poleward warm surface water currents. Zigzag lines indicate major frontal systems separating counter rotating ocean currents (e.g., W of Portugal and N of

MD07-307; after Skinner et al., 2014). (a) Late LGM circulation geometry (21–18.7 cal.

ka), largely similar to today. Note the major east-west gradient of ventilation ages in the central North Atlantic, between Portugal (PORT) and Mid-Atlantic Ridge W of Azores (MAR). (b) HS-1 benthic ventilation ages reveal a short-lasting MOC reversal leading to

Atlantic-style overturning in the subpolar North Pacific and coeval Pacific-style stratific- ation in the northern North Atlantic, with seesaw-style reversals of global MOC at the onset and end of early HS-1 (first proposed by Broecker et al., 1985, however, for LGM

times). Increased ventilation ages reflect enhanced uptake of dissolved carbon in the

LGM deep ocean (Sarnthein et al., 2013), major drops suggest major degassing of $CO_2$

from both the deep Southern Ocean and North Pacific during early HS-1. – SCS =

South China Sea. AABW = Antarctic Bottom Water; AAIW = Antarctic Intermediate

Water. NADW = North Atlantic Deep Water. Small arrows within age numbers reflect temporal trends. Many arrows are speculative using circumstantial evidence of benthic

$\delta^{13}C$ records and local Coriolis forcing at high-latitude sites per analogy to modern scenarios. Location of sediment cores are given in Fig. 7, short-term variations in planktic and benthic $^{14}C$ reservoir/ventilation age in Suppl. Fig. S2 and Table 3.

[Figure]

↓ Fig. 11. Global distribution of ¹⁴C reservoir ages obtained (a) for late LGM

intermediate waters (100–1800 m w.d.) and (b) for LGM deep waters (>1800 m w.d., including Site GIK 23074 at 1157 m in the Norwegian Sea).

---

## Author Response (AR4)

Michael Sarnthein et al.
Dear Editor, Dear André Paul,
With thanks we obtained your helpful recent comments and tried to include them with care into
our manuscript. A detailed response to the points you raised you will find below: They are
marked by arrows "in between your lines".
We now hope that the present version of our manuscript may be satisfactory to be accepted for
publication in C.P.
Yours sincerely,
Michael Sarnthein and coauthors.
„„„„„„„„„„„„„„„„„„„„„„„„„„„„„„„„„„„„„„„„„„„„„„„„„„„„„„„
Editor Decision: Publish subject to minor revisions (review by editor) (30 Jul 2020) by C.P.
editor André Paul: Comments to the Author:
Dear Authors,
Thank you for submitting a revised version of your manuscript. Unfortunately, I think that it is
not quite the final version, because it differs in a few points from your response to the points (1)
and (2) that I had raised before:

**(1) Unpublished data and manuscripts "in prep."**
I accept your decision to put your manuscript on hold until the two manuscripts by Ausin et al.
and of Küssner et al.
become publicly available (which probably means until they are accepted for publication) and
the embargo on the related data sets on PANGAEA will be lifted.
--> Embargo on the crucial data sets of Ausin et al. and Küssner et al. on PANGAEA will be
lifted by September.
As one of our reviewers suggested, readers should not necessarily believe what is written but
rather look at primary data themselves. For this reason, we shall keep you as editor informed,
when the data are publicly available and we shall ask you for a lift of the "on hold" status of our
paper as soon as the embargo of the data at PANGAEA will be released.
In the revised version of the manuscript, the following citations would
need to be corrected:  --> our recent pertinent changes are marked RED
Line 264: Ausin et al., 2020 subm.
--> the present raw draft of this manuscript will be ready for submission to P&P after the
vacations in September.
Line 637: Ausin et al. (in prep.)
--> now converted to '2020 subm.'

Line 171, 263, 476, 481, 503, 591-592, 636, 669, 847, 1338: Küssner et al., 2020 subm.
--> the manuscript is now under review at P & P.
A QUESTION: Would you as editor prefer a change of all citations to "Küssner et al., 2020,
under review"?
In addition, the manuscripts by Ausin et al. would need to be added to the list of references,
while the reference to Küssner et al. in lines 973-975 would need to be completed.
--> Proper refs to Ausin and Küssner are now included!

**57  (2) Personal communication ("pers. com")**

In contrast to your response, I cannot find the reference to the paper by Bronk Ramsey "now
published under 2020"".
Instead, in lines 228-229 of the revised manuscript, there is still a reference to personal
communication.
--> a proper ref. to Bronk Ramsey et al. 2020 has now been included
Similarly, I cannot find the reference to the abstract by Abé-Ouchi "for the OC3/IPODS
Meeting held in Cambridge in September 2018"". Instead, in lines 656-657 of the revised
manuscript, there is still a reference to personal communication. I suppose that in both cases
the modifications to the manuscript were simply lost or forgotten. --> a proper ref. to Abé-
Ouchi was now included.

**70  (3) Length of manuscript**

I am fine with your efforts to slightly shorten the abstract and
the introduction. --> GREAT!
Finally, when reading your manuscript again it occurred to me that the model used by Gebbie
(2014) is not the MIROC model, but a simple "water mass decomposition model" or "global
tracer transport model", as the author himself calls it.
The paragraph leading up to line 653 of the revised version of the manuscript would need to be
rephrased accordingly. --> To meet your argument our text at line 653 now was rephrased.
Yours sincerely,
André Paul
,,,,,,,,,,,,,,,,,,,,,,,,,,,,,,,,,,,,,,,,,,,,,,,,,,,,,,,,,,,,,,,,,,,,,,,

[revised manuscript text omitted]